# MULTI-TASK BEST ARM IDENTIFICATION WITH RISK CONSTRAINT

## ABSTRACT

Best Arm Identification is a very challenging problem in sequential decision-making with many real-world applications. Existing works typically assume that all arms are feasible or/and deal with expectation-based constraints with strong assumptions, loose sample complexity bounds, and non-optimal algorithms. This paper introduces a multi-task best arm identification problem with risk constraint in the fixed-confidence setting, where each arm has multiple performance metrics. The agent aims to optimize one metric while ensuring that the quantiles of other metrics remain below specified thresholds for each task. We first derive a tight, instance-dependent lower bound on sample complexity. Based on this bound, we establish optimality conditions for the static optimal sampling ratio and illustrate how it balances among different tasks and constraints, while addressing the trade-off between optimality and feasibility. We derive a Track-and-Stop strategy with asymptotically optimal sample complexity and a computationally efficient strategy that iteratively solves the optimality conditions. Finally, we extend our results to the linear bandit setting. Numerical experiments show that our algorithm performs relatively well.

## 1 INTRODUCTION

In recent years, Best Arm Identification (BAI) has been attracting considerable attention and has been widely applied in various fields such as chemistry (Bengio & LeCun, 2007; Wang et al., 2024), prompt learning (Shi et al., 2024), recommendation systems (Zhao & Yang, 2024), and A/B/n testing (Russac et al., 2021). In this paper, we consider a multi-task BAI problem with risk constraint. In this problem, each arm has multiple performance metrics and the agent aims to optimize one of the metrics while ensuring that the quantiles of the rest metrics remain below specified thresholds for each task. At each time step $t$, the agent chooses a task, arm, and metric pair to sample, receiving a reward drawn from the corresponding probability distribution. In the fixed-confidence setting, the agent must assess the feasibility and optimality of each arm across all tasks, identifying the best arm for each task with a probability of at least $1 - \delta$, while minimizing the number of samples required.

The multi-task BAI problem with risk constraint has numerous real-world applications. In the context of drug discovery, for example, the experimenter must identify the most appropriate drug for each disease. A drug has multiple performance metrics, including its curative effect and potential side effects. The experimenter aims to identify the drug with the best average curative effect while ensuring that the risk of side effects remains below a certain threshold for each disease. In such an application, some drugs may be infeasible due to high side effects in rare cases, despite having strong average curative effects. The experimenter must consider both optimality and feasibility while balancing the difficulty of identifying the best drug for various diseases. Additional motivating examples are provided in Appendix A.2.

The existing methods in the literature cannot be used directly to solve the multi-task BAI problem with risk constraint. Traditional BAI models assume that all arms are feasible (Garivier & Kaufmann, 2016; Wang et al., 2021) with primary focus on evaluating the optimality of these arms. They cannot simultaneously address both optimality and feasibility. While some works have tried to tackle the BAI problem with constraint, their formulations either impose constraints on the sampling rule (Das & Basu; Tang et al., 2024) rather than the multiple performance settings we consider here, or they focus only on expectation-based constraints with very strong assumptions, such as

linear structure (Shang et al., 2023; Wang et al., 2022) and known objective value (Lindner et al., 2022). Previous work on risk-based BAI either uses a risk-based objective for unconstrained problems (Agrawal et al., 2021) or can only address a single performance metric setting (David et al., 2018; Chang et al., 2020; Hou et al., 2022). Furthermore, the sample complexity lower bounds in these works are often very loose and the optimality remains elusive.

The contributions of this paper are summarized as follows:

- We propose a multi-task BAI problem with risk constraint applicable to various real-world problems. We derive a tight, instance-dependent lower bound on the sample complexity required to guarantee a high probability identification of the feasible and optimal arm for each task.

- Based on the lower bound, we derive the optimality conditions for the static optimal sampling ratio and discuss how this ratio balances the difficulty across different tasks and constraints, illustrating the trade-off between optimality and feasibility. Additionally, we present a closed-form formula for the problem's hardness by analyzing some challenging instances.

- We derive a Track-and-Stop strategy for the multi-task BAI problem with risk constraint, achieving asymptotically optimal sample complexity and a computationally efficient strategy that iteratively solves the optimality conditions. Additionally, we extend our sample complexity results to the linear bandit setting. Numerical experiments show that the proposed algorithm performs well in comparison to several benchmarks.

**Best arm identification.** BAI is a widely studied problem in the bandit community (Even-Dar et al., 2006; Gabillon et al., 2012; Kaufmann & Kalyanakrishnan, 2013). Our work builds on recent research on standard BAI, which aims to derive instance-dependent sample complexity lower bounds and asymptotically optimal strategies (Kaufmann et al., 2016; Garivier & Kaufmann, 2016; Degenne et al., 2019; Degenne & Koolen, 2019; Wang et al., 2021). We extend these methods to address the more challenging multi-task formulations with risk constraint, and the results for standard BAI can be derived as a special case.

**Constrained best arm identification.** There is an extensive literature on the constrained BAI problem. Some works focus on constraints related to the sampling ratio of arms, such as Fair BAI (Russo & Vannella, 2024) and BAI with Knapsacks (Tran-Thanh et al., 2012; Li et al., 2023) and other settings (Das & Basu; Tang et al., 2024), which differ from our formulation. Others address performance-based constraints (Faizal & Nair, 2022), such as in safety linear BAI (Camilleri et al., 2022; Shang et al., 2023). However, many of these works center on linear constraints with stronger assumptions (Lindner et al., 2022) or requires that constraints be satisfied throughout the exploration phase (Shang et al., 2023; Wang et al., 2022). Unlike these works, our problem focuses on pure exploration, where the agent can choose each arm and estimate both its optimality and feasibility simultaneously. Additionally, some works explore BAI with risk constraint, where formulations typically consider the single performance metric of an arm, using the mean as the objective function and the corresponding risk measures, such as variance (Hou et al., 2022) or conditional value at risk (David et al., 2018), as constraints. Our formulation differs from all prior works in that we consider multi-task BAI with risk constraint in the fixed-confidence setting. Our lower bound is both tight and instance-dependent, while the sample complexity upper bound of our algorithm matches the lower bound and is asymptotically optimal.

**Multi-objective best arm identification.** Our work is also related to multi-objective BAI, which considers arms with multiple performance metrics. However, different from the previous works that focus on identifying the Pareto set (Auer et al., 2016; Kone et al., 2023; 2024), we aim to optimize one performance metric while keeping the quantiles of other metrics below known thresholds. These two types of problems have different objectives, and the Pareto set identification algorithm cannot be used to solve our problem.

## 2    PROBLEM FORMULATION

In this section, we present the formulation for the multi-task BAI problem with risk constraint and define the notation used throughout the paper.

The agent is given $M$ BAI tasks, each with $K$ arms. In each task $a \in [M] = \{1, \ldots, M\}$, arm $i \in [K] = \{1, \ldots, K\}$ corresponds to a random vector $(X_i^a, Y_{i1}^a, \ldots, Y_{iS}^a) \in \mathbb{R}^{S+1}$, where each element represents a performance metric of arm $i$. Let $F_i^a(\cdot)$ and $F_{ij}^a(\cdot)$ denote the cumulative distribution function (CDF) of random variable $X_i^a$ and $Y_{ij}^a$, respectively, with $\mu_i^a$ and $\mu_{ij}^a$ being their means. Define the $\phi$-quantile of $Y_{ij}^a$ as

$$q(Y_{ij}^a, \phi) = \inf\{y : F_{ij}^a(y) \geq \phi\}. \tag{1}$$

The agent needs to solve the following optimization problem for each task:

$$\max_{i \in [K]} \mathbb{E}[X_i^a] \quad \text{s.t.} \ \ q(Y_{ij}^a, \phi) \leq b, \ \forall j \in [S]. \tag{2}$$

For notational simplicity, we let $F_{ij}^a$ represent $F_{ij}^a(b)$ and consider a uniform quantile level $\phi$ and constraint threshold $b$, which can be easily extended to multiple quantile levels and thresholds. For each task $a$, let $\boldsymbol{\mu}^a = (\mu_i^a)_{i \in [K]}$, $\boldsymbol{Q}^a = (q(Y_{ij}^a, \phi))_{i \in [K], j \in [S]}$. Then, a multi-task BAI problem instance can be denoted as $\mathcal{P} = (\boldsymbol{\mu}^a, \boldsymbol{Q}^a)_{a \in [M]}$. We adopt the following widely used assumptions.

**Assumption 1.** *Let $\mathcal{S}$ denote the set of problem instances $\tilde{\mathcal{P}}$ where a unique best arm $i^*(a, \tilde{\mathcal{P}})$ exists for each task $a$, and all constraints are active. Assume that $\mathcal{P} \in \mathcal{S}$.*

**Assumption 2.** *The distribution of $X_i^a$ belongs to the single-parameter exponential family. $F_{ij}^a(\cdot)$ has a continuous density function.*

Assumption 1 is widely used in the BAI problem (Garivier & Kaufmann, 2016; Camilleri et al., 2022), as distinguishing between two arms with identical means or determining whether quantiles equal to the threshold would require an infinite number of samples. The single-parameter exponential family (SPEF) in Assumption 2 is general and includes common distributions such as Bernoulli and Gaussian with known variance. More details about the SPEF can be found in Appendix B.3.

In the online setting, at each time step $t$, the agent chooses a task, arm, and metric pair $\pi^t = (a^t, i^t, j^t)$ to sample, where $j^t = S + 1$ indicates sampling from the objective function $X_i^a$. The agent then observes a random sample $Z_t$, drawn independently across tasks, arms, and metrics, corresponding to the chosen random variable. Let $\mathcal{F}_t = \sigma(\pi^1, Z_1, \ldots, \pi^t, Z_t)$ denote the sigma-field generated by the sampling decisions and observations up to time $t$.

**Remark 1.** *In the formulation above, the agent observes only one performance metric for each sample. In contrast, some works in the literature (Hunter & Pasupathy, 2013; Camilleri et al., 2022; Kone et al., 2023) assume the agent observes a vector of all metrics after pulling an arm. We focus on the former, as it is more challenging, and the sample complexity scales with the number of constraints. An extension to the latter case is provided in Theorem 5.*

**Remark 2.** *In this formulation, we adopt the multi-task setting. An alternative is to solve each task individually. However, the mathematical model and analysis of this setting are more general, enabling extension to scenarios with linear structure across tasks (see Appendix D.4 for details).*

A strategy for BAI is defined by three components: the sampling rule $\{\pi^t\}_t$, where $\pi^t$ is $\mathcal{F}_{t-1}$ measurable; the stopping rule $\tau$, which is a stopping time with respect to $\mathcal{F}_t$; and the decision rule $(\hat{i}_\tau(a))_{a \in [M]}$, where $\hat{i}_\tau(a)$ denotes the recommended arm for task $a$ when the algorithm terminates. Let $N_{ij}^a(t), j \in [S]$ and $N_{i(S+1)}^a(t)$ represent the number of samples of constraint $j$ and the objective, respectively, for arm $i$ in task $a$ up to time $t$. $\omega_{ij}^a(t) = N_{ij}^a(t)/t, j \in [S+1]$ denote the corresponding sampling ratio up to time $t$. In the fixed confidence setting, given a confidence level $\delta \in (0, 1)$, the agent aims to identify the best arm $i^*(a, \mathcal{P})$ for all tasks $a \in [M]$ with a probability of at least $1 - \delta$, while minimizing the sample complexity $\mathbb{E}[\tau]$. To simplify notation, we omit the dependence of $i^*(a, \mathcal{P})$ on $\mathcal{P}$ when it does not cause any confusion.

**Additional notation.** Denote by $\Omega = \{\boldsymbol{\omega} \in \mathbb{R}_+^{MK(S+1)} : \sum_{a \in [M], i \in [K], j \in [S+1]} \omega_{ij}^a = 1\}$. $\mathcal{D}_1^a = \{i \in [K] : \mu_i^a < \mu_{i^*(a)}^a, \forall j \in [S], q(Y_{ij}^a, \phi) \leq b\}$ is the set of all suboptimal arms, $\mathcal{D}_2^a = \{i \in [K] : \mu_i^a > \mu_{i^*(a)}^a, \exists j \in [S], q(Y_{ij}^a, \phi) > b\}$ is the set of all infeasible arms with better objective value, and $\mathcal{D}_3^a = \{i \in [K] : \mu_i^a < \mu_{i^*(a)}^a, \exists j \in [S], q(Y_{ij}^a, \phi) > b\}$ is the set of all infeasible arms with worse objective value for task $a$. Let $\mathcal{B}_1^a(i) = \{j \in [S] : q(Y_{ij}^a, \phi) \leq b\}$ represent the indices of feasible constraints for arm $i$, and $\mathcal{B}_2^a(i) = [S] \backslash \mathcal{B}_1^a(i)$ represent the indices of infeasible constraints.

Denote $\mathcal{A}(\mathcal{P}) = \{\mathcal{P}' \in \mathcal{S} : \exists a \in [M], i^*(a, \mathcal{P}) \neq i^*(a, \mathcal{P}')\}$ as the set of problem instances where the best arm for at least one task $a$ differs from that in $\mathcal{P}$. Define $\mathrm{kl}(p, q) = p\log(p/q) + (1 - p)\log((1 - p)/(1 - q))$ for $p, q \in (0, 1)$, and $d(p, q)$ as the Kullback–Leibler divergence between two distributions in a SPEF with means $p$ and $q$, respectively. A table summarizing the notation is provided in Appendix A.1.

## 3 SAMPLE COMPLEXITY

In this section, we establish the lower bound for the sample complexity and derive the optimality conditions for the static optimal sampling ratio. Formal proofs of these results are provided in Appendix B.

### 3.1 LOWER BOUND ON THE SAMPLE COMPLEXITY

In this subsection, we derive an instance-dependent lower bound on the sample complexity $\mathbb{E}[\tau]$. This bound defines the minimum number of samples required to identify the best arm for all tasks with high probability. To exclude trivial cases, we assume that for each task $a$, there are four types of arms: the optimal arm $i^*(a)$, suboptimal arms in $\mathcal{D}_1^a$, infeasible arms with better objective values in $\mathcal{D}_2^a$, and infeasible arms with worse objective values in $\mathcal{D}_3^a$. Theorem 1 provides the lower bound on $\mathbb{E}[\tau]$.

**Theorem 1.** *Given a fixed confidence level $\delta \in (0, 1)$, define $\mu_{i,i^*(a)}^a = \arg\inf_{\tilde{\mu}} \omega_{i(S+1)}^a d(\mu_i^a, \tilde{\mu}) + \omega_{i^*(a)(S+1)}^a d(\mu_{i^*(a)}^a, \tilde{\mu})$. Under Assumptions 1-2, for any problem instance $\mathcal{P} \in \mathcal{S}$ and any strategy satisfying $\mathbb{P}\left(\forall a \in [M], i^*(a) = \hat{i}_\tau(a)\right) \geq 1 - \delta$,*

$$\mathbb{E}[\tau] \geq \mathcal{H}^*(\mathcal{P})\mathrm{kl}(\delta, 1 - \delta), \tag{3}$$

*as $\delta \to 0$, we have*

$$\liminf_{\delta \to 0} \frac{\mathbb{E}[\tau]}{\log(1/\delta)} \geq \mathcal{H}^*(\mathcal{P}), \tag{4}$$

*where $\mathcal{H}^*(\mathcal{P})^{-1} = \max_{\boldsymbol{\omega} \in \Omega} \min_{a \in [M]} \min\left(V_1^a(\boldsymbol{\omega}), V_2^a(\boldsymbol{\omega}), V_3^a(\boldsymbol{\omega}), V_4^a(\boldsymbol{\omega})\right)$, with*

$$V_1^a(\boldsymbol{\omega}) = \min_{j \in [S]} \omega_{i^*(a)j}^a d(F_{i^*(a)j}^a, \phi),$$

$$V_2^a(\boldsymbol{\omega}) = \min_{i \in \mathcal{D}_1^a} \omega_{i(S+1)}^a d(\mu_i^a, \mu_{i,i^*(a)}^a) + \omega_{i^*(a)(S+1)}^a d(\mu_{i^*(a)}^a, \mu_{i,i^*(a)}^a),$$

$$V_3^a(\boldsymbol{\omega}) = \min_{i \in \mathcal{D}_2^a} \sum_{j \in \mathcal{B}_2^a(i)} \omega_{ij}^a d(F_{ij}^a, \phi),$$

$$V_4^a(\boldsymbol{\omega}) = \min_{i \in \mathcal{D}_3^a} \omega_{i(S+1)}^a d(\mu_i^a, \mu_{i,i^*(a)}^a) + \omega_{i^*(a)(S+1)}^a d(\mu_{i^*(a)}^a, \mu_{i,i^*(a)}^a) + \sum_{j \in \mathcal{B}_2^a(i)} \omega_{ij}^a d(F_{ij}^a, \phi).$$

The basic idea behind the analysis of Theorem 1 is that we can construct an alternative, indistinguishable problem instance $\tilde{\mathcal{P}} \in \mathcal{A}(\mathcal{P})$, which has a different best arm for some tasks. Using Lemma 1 in Kaufmann et al. (2016), we can derive the number of samples necessary to distinguish between these two instances. The main insight from Theorem 1 is that arms with varying feasibility and optimality contribute differently to the overall complexity, as explicitly derived in Proposition 1 for certain challenging instances.

**Technical Novelty.** Theorem 1 extends the sample complexity results of Theorem 1 in Garivier & Kaufmann (2016) to the multi-task BAI with risk constraint setting. The primary technical novelty lies in transforming the problem of comparing quantiles with a threshold into one of comparing the CDF value at a given point with the corresponding quantile level, classifying the arms for each task into four categories based on optimality and feasibility, and analyzing the sample complexity for each category. The analysis method is general and can be extended to other formulations, as discussed in Section 6 and E.2.

The lower bound on the sample complexity depends on an $\delta$-related constant $\mathrm{kl}(\delta, 1 - \delta)$ and the statistical complexity $\mathcal{H}^*(\mathcal{P})$. The complexity $\mathcal{H}^*(\mathcal{P})$ is defined by a multi-level optimization problem. The inner problem seeks to find a problem instance indistinguishable from $\mathcal{P}$ in terms of KL

divergence, while the outer problem tries to maximize the difference between these two instances by finding a static optimal sampling ratio

$$\boldsymbol{\omega}^*(\mathcal{P}) = \arg\max_{\boldsymbol{\omega}\in\Omega} \min_{a\in[M]} \min\left(V_1^a(\boldsymbol{\omega}), V_2^a(\boldsymbol{\omega}), V_3^a(\boldsymbol{\omega}), V_4^a(\boldsymbol{\omega})\right). \tag{5}$$

A strategy is optimal when its sampling ratio $\boldsymbol{\omega}(t)$ meets $\boldsymbol{\omega}^*(\mathcal{P})$, we omit the dependence of $\boldsymbol{\omega}^*(\mathcal{P})$ on $\mathcal{P}$ whenever it is unambiguous.

## 3.2 STATIC OPTIMAL SAMPLING RATIO

In this subsection, we derive the optimality conditions for problem (5) and provide some insights into the static optimal sampling ratio $\boldsymbol{\omega}^*$. Denote $j_h(a,i) = \arg\max_{j\in\mathcal{B}_2^a(i)} d(F_{ij}^a, \phi)$ as the most distinguishable infeasible constraint of arm $i$ in task $a$. We omit the dependence of $j_h(a,i)$ on $(a,i)$ whenever it is unambiguous. Theorem 2 establishes the optimality conditions for $\boldsymbol{\omega}^*$.

**Theorem 2.** *Let* $\mathcal{M}_1^a = \{i \in \mathcal{D}_3^a : d(\mu_i^a, \mu_{i,i^*(a)}^a) > d(F_{ij_h}^a, \phi)\}$ *and* $\mathcal{M}_2^a = \{i \in \mathcal{D}_3^a : d(\mu_i^a, \mu_{i,i^*(a)}^a) < d(F_{ij_h}^a, \phi)\}$. *Assume that* $\mathcal{D}_3^a = \mathcal{M}_1^a \cup \mathcal{M}_2^a$ *for each task* $a \in [M]$, *then the static optimal sampling ratio* $\boldsymbol{\omega}^*$ *satisfies:*

$$V_1^a(\boldsymbol{\omega}^*) = V_2^a(\boldsymbol{\omega}^*) = V_3^a(\boldsymbol{\omega}^*) = V_4^a(\boldsymbol{\omega}^*) = z^*, \forall a \in [M]$$

$$(\omega_{i^*(a)j}^a)^* d(F_{i^*(a)j}^a, \phi) = z^*, \forall a \in [M], j \in [S]$$

$$(\omega_{i(S+1)}^a)^* d(\mu_i^a, \mu_{i,i^*(a)}^a) + (\omega_{i^*(a)(S+1)}^a)^* d(\mu_{i^*(a)}^a, \mu_{i,i^*(a)}^a) = z^*, \forall a \in [M], i \in \mathcal{D}_1^a \cup \mathcal{M}_1^a$$

$$\sum_{i\in\mathcal{D}_1^a\cup\mathcal{M}_1^a} \frac{d(\mu_{i^*(a)}^a, \mu_{i,i^*(a)}^a)}{d(\mu_i^a, \mu_{i,i^*(a)}^a)} = 1, \forall a \in [M]$$

$$(\omega_{ij_h}^a)^* d(F_{ij_h}^a, \phi) = z^*, \forall a \in [M], i \in \mathcal{D}_2^a \cup \mathcal{M}_2^a$$

$$(\omega_{ij}^a)^* = 0, \forall a \in [M], i \in \mathcal{D}_1^a \cup \mathcal{M}_1^a, j \in [S]$$

$$(\omega_{ij}^a)^* = 0, \forall a \in [M], i \in \mathcal{D}_2^a \cup \mathcal{M}_2^a, j \neq j_h$$

$$\sum_{a\in[M]} \sum_{i\in[K]} \sum_{j\in[S+1]} (\omega_{ij}^a)^* = 1,$$

$$(\omega_{ij}^a)^* \geq 0, \forall a \in [M], i \in [K], j \in [S+1]. \tag{6}$$

**Remark 3.** $\mathcal{M}_1^a$ *includes the arms in* $\mathcal{D}_3^a$ *that are easier to identify as suboptimal than infeasible, while* $\mathcal{M}_2^a$ *represents the opposite case. For simplicity, we disregard the case where* $d(\mu_i^a, \mu_{i,i^*(a)}^a) = d(F_{ij_h}^a, \phi)$, *since the optimal solution is not unique, and the solution to equations (6) is one of the optimal solutions.*

**Insights and illustrative examples.** Here we provide some insights on the static optimal sampling ratio $\boldsymbol{\omega}^*$. First, it aims to balance difficulty and equalize error probability across tasks, i.e., $V_1^a(\boldsymbol{\omega}^*) = V_2^{a'}(\boldsymbol{\omega}^*)$, for all task $a$ and $a'$. Second, it seeks to equalize the error probability among the four types of arms for each task, i.e., $V_1^a(\boldsymbol{\omega}^*) = \ldots, = V_4^a(\boldsymbol{\omega}^*)$. Third, for the best arm $i^*(a)$, the optimal sampling ratio is proportional to the difficulty of identification, i.e.,

$$(\omega_{i^*(a)j}^a)^* / (\omega_{i^*(a)j'}^a)^* = d(F_{i^*(a)j'}^a, \phi) / d(F_{i^*(a)j}^a, \phi), \forall a \in [M], j, j' \in [S].$$

Finally, for arms in $\mathcal{D}_1^a$ and $\mathcal{M}_1^a$, it focuses on identifying them as suboptimal, while for arms in $\mathcal{D}_2^a$ and $\mathcal{M}_2^a$, it focuses on identifying them as infeasible. Illustrative examples are provided in Appendix D.1.

**Hardness of problem instance.** The complexity $\mathcal{H}^*(\mathcal{P})$ quantifies the difficulty of the problem instance $\mathcal{P}$. However, it is defined through a multi-level optimization problem, making it challenging to identify the specific factors contributing to the problem's hardness. To make it more explicit, we examine some particularly challenging problem instances.

**Lemma 1.** *Define* $\triangle_i^a = \mu_{i^*(a)}^a - \mu_i^a$, $\triangle_{i^*(a)}^a = \min_{l\in\mathcal{D}_1^a\cup\mathcal{M}_1^a}(\mu_{i^*(a)}^a - \mu_l^a)$, $\sigma_i^a = (\int x^2 dF_i^a(x) - (\mu_i^a)^2)^{\frac{1}{2}} \in (0,\infty)$, *then for each task* $a \in [M]$, *and arm* $i \in \mathcal{D}_1^a \cup \mathcal{M}_1^a$, *as* $\triangle_i^a \to 0$,

$$\inf_{\tilde{\mu}_i^a > \tilde{\mu}_{i^*(a)}^a} \omega_{i(S+1)}^a d(\mu_i^a, \tilde{\mu}_i^a) + \omega_{i^*(a)(S+1)}^a d(\mu_{i^*(a)}^a, \tilde{\mu}_{i^*(a)}^a) \to \frac{(\mu_i^a - \mu_{i^*(a)}^a)^2}{2\left(\frac{(\sigma_i^a)^2}{\omega_{i(S+1)}^a} + \frac{(\sigma_{i^*(a)}^a)^2}{\omega_{i^*(a)(S+1)}^a}\right)}. \tag{7}$$

**Lemma 2.** *Define $\triangle_{ij}^a = |F_{ij}^a - \phi|$, $\sigma_{ij}^a = (F_{ij}^a(1 - F_{ij}^a))^{\frac{1}{2}} \in (0, \infty)$, then for each task $a \in [M]$, arm $i \in [K]$ and constraint $j \in [S]$, as $\triangle_{ij}^a \to 0$,*

$$d(F_{ij}^a, \phi) \to \frac{(F_{ij}^a - \phi)^2}{2(\sigma_{ij}^a)^2}. \tag{8}$$

Lemma 1 and Lemma 2 show that as the problem instance becomes increasingly difficult, the KL divergence can be approximated using only the first two moments. Using these two lemmas, we can derive both an upper and lower bound for $\mathcal{H}^*(\mathcal{P})$ in the following proposition.

**Proposition 1.** *Define*

$$\mathcal{H} = \sum_{a \in [M]} \left( \sum_{j \in [S]} \frac{(\sigma_{i^*(a)j}^a)^2}{(\triangle_{i^*(a)j}^a)^2} + \sum_{i \in \mathcal{D}_1^a \cup \mathcal{M}_1^a \cup \{i^*(a)\}} \frac{(\sigma_i^a)^2}{(\triangle_i^a)^2} + \sum_{i \in \mathcal{D}_2^a \cup \mathcal{M}_2^a} \frac{(\sigma_{ij_h}^a)^2}{(\triangle_{ij_h}^a)^2} \right). \tag{9}$$

*For the problem instance $\mathcal{P}$ with $\triangle_i^a \to 0, \forall a \in [M], i \in \mathcal{D}_1^a \cup \mathcal{M}_1^a$, and $\triangle_{ij}^a \to 0, \forall a \in [M], i \in [K], j \in [S]$, we have $2\mathcal{H} \leq \mathcal{H}^*(\mathcal{P}) \leq 4\mathcal{H}$.*

Proposition 1 provides better insight into the magnitude of $\mathcal{H}^*(\mathcal{P})$. For the best arm $i^*(a)$, the problem's hardness depends on the feasibility gap across all constraints. For suboptimal and infeasible arms in $\mathcal{D}_1^a$ and $\mathcal{M}_1^a$, it depends on the variance and optimality gap, while for infeasible arms in $\mathcal{D}_2^a$ and $\mathcal{M}_2^a$, it is determined by the feasibility gap of the most distinguishable constraints. Additionally, Proposition 1 leads to a sampling ratio in Lemma 3, which is proportional to the identification difficulty.

**Lemma 3.** *For the problem instance $\mathcal{P}$ with $\triangle_i^a \to 0, \forall a \in [M], i \in \mathcal{D}_1^a \cup \mathcal{M}_1^a$, and $\triangle_{ij}^a \to 0, \forall a \in [M], i \in [K], j \in [S]$, consider the static sampling ratio $\tilde{\omega}$ with*

$$\tilde{\omega}_{i^*(a)j}^a = \frac{(\sigma_{i^*(a)j}^a)^2}{\mathcal{H}(\triangle_{i^*(a)j}^a)^2}, \ \forall a \in [M], j \in [S]$$

$$\tilde{\omega}_{i(S+1)}^a = \frac{(\sigma_i^a)^2}{\mathcal{H}(\triangle_i^a)^2}, \ \forall a \in [M], i \in \mathcal{D}_1^a \cup \mathcal{M}_1^a \cup \{i^*(a)\} \tag{10}$$

$$\tilde{\omega}_{ij_h}^a = \frac{(\sigma_{ij_h}^a)^2}{\mathcal{H}(\triangle_{ij_h}^a)^2}, \ \forall a \in [M], i \in \mathcal{D}_2^a \cup \mathcal{M}_2^a$$

*$\tilde{\omega}_{ij}^a = 0$ otherwise. Define $\mathcal{H}(\mathcal{P}, \tilde{\omega})^{-1} = \min_{a \in [M]} \min \left( V_1^a(\tilde{\omega}), V_2^a(\tilde{\omega}), V_3^a(\tilde{\omega}), V_4^a(\tilde{\omega}) \right)$, then we have $\mathcal{H}(\mathcal{P}, \tilde{\omega}) = \mathcal{O}(\mathcal{H})$, where $\mathcal{O}(\cdot)$ represents the suppression of certain constants.*

**Comparison to previous work.** The most related work to ours is Garivier & Kaufmann (2016), which considers the unconstrained, single-task BAI problem. Their formulation is a special case of ours with $M = 1$, $\mathcal{D}_2^a = \mathcal{D}_3^a = \emptyset$, and the agent knows that all arms are feasible. In this case, Theorem 2 reduces to $\omega^* \in \Omega$,

$$(\omega_{i(S+1)})^* d(\mu_i, \mu_{i,i^*}) + (\omega_{i^*(S+1)})^* d(\mu_{i^*}, \mu_{i,i^*}) = z^*, \forall i \in [K] \setminus \{i^*\}$$

$$\sum_{i \in [K] \setminus \{i^*\}} \frac{d(\mu_{i^*}, \mu_{i,i^*})}{d(\mu_i, \mu_{i,i^*})} = 1, \tag{11}$$

recovering their main results given in Theorem 5 of Garivier & Kaufmann (2016).

## 4 THE OPTIMAL STRATEGY

In this section, we propose an asymptotically optimal strategy and a computationally efficient strategy for the multi-task BAI problem with risk constraint. Formal proofs for all the results in this section are provided in Appendix C.

## 4.1 TRACK-AND-STOP STRATEGY

We extend the Track-and-Stop strategy proposed in Garivier & Kaufmann (2016) for solving the unconstrained, single-task BAI problem. The main novelty lies in deriving a new sampling rule by solving the equations in (6) and a stopping rule that incorporates the influence of multiple tasks and risk constraints.

**Sampling rule.** From the analysis of Theorem 1, the constraint $q(Y_{ij}^a, \phi) \leq b$ is equivalent to $F_{ij}^a \geq \phi$. Therefore, a problem instance can be represented as $\mathcal{P} = (\boldsymbol{\mu}^a, \boldsymbol{F}^a)_{a \in [M]}$ with $\boldsymbol{F}^a = (F_{ij}^a)_{i \in [K], j \in [S]}$. Define

$$
\begin{aligned}
\hat{\mu}_i^a(t) &= \frac{1}{N_{i(S+1)}^a(t)} \sum_{s \leq t} Z_s \mathbb{I}(a^s = a, i^s = i, j^s = S+1), \\
\hat{F}_{ij}^a(t) &= \frac{1}{N_{ij}^a(t)} \sum_{s \leq t} \mathbb{I}(Z_s \leq b) \mathbb{I}(a^s = a, i^s = i, j^s = j),
\end{aligned}
\tag{12}
$$

and let $\hat{\mathcal{P}}(t) = (\hat{\boldsymbol{\mu}}^a(t), \hat{\boldsymbol{F}}^a(t))_{a \in [M]}$ denote the empirical problem instance. The static optimal sampling rule $\boldsymbol{\omega}^*(\mathcal{P})$ is not implementable because it depends on the unknown parameters $\mathcal{P}$. A natural approach is to plug in the estimate $\hat{\mathcal{P}}(t)$ and track the empirical optimal sampling ratio $\boldsymbol{\omega}^*(\hat{\mathcal{P}}(t))$. The sampling rule can be defined as

$$
\pi^{t+1} = \begin{cases} \arg\min_{(a,i,j) \in U_t} N_{ij}^a(t) \text{ if } U_t \neq \emptyset \\ \arg\max_{(a,i,j) \in [M] \times [K] \times [S+1]} t(\omega_{ij}^a)^*(\hat{\mathcal{P}}(t)) - N_{ij}^a(t) \end{cases},
\tag{13}
$$

with $U_t = \{(a, i, j) : N_{ij}^a(t) < \sqrt{t} - MK(S+1)/2\}$. Since the computation of $\boldsymbol{\omega}^*(\hat{\mathcal{P}}(t))$ depends on the empirical sets $\{i^*(a, \hat{\mathcal{P}}(t))\}$ and $\hat{\mathcal{D}}_1^a(t)$, the algorithm will, for simplicity, apply an equal sampling rule when either of these sets is empty for some task $a$.

The sampling rule chooses each $(a, i, j)$ pair at least $\Omega(\sqrt{t})$ times overall. Asymptotically, $\hat{\mathcal{P}}(t)$ converges to $\mathcal{P}^*$ and $\boldsymbol{\omega}^*(\hat{\mathcal{P}}(t))$ converges to $\boldsymbol{\omega}^*(\mathcal{P})$.

---

**Algorithm 1** Track-and-Stop Strategy

---

1: **Initialization.** Pull each $(a, i, j) \in [M] \times [K] \times [S+1]$ $n_0$ times.
2: Set $t \leftarrow n_0 MK(S+1)$, and update $\hat{\mathcal{P}}(t)$, $\boldsymbol{\omega}^*(\hat{\mathcal{P}}(t))$, $U_t$, $N_{i(S+1)}^a(t)$, $N_{ij}^a(t)$.
3: **while** $\inf_{\tilde{\mathcal{P}} \in \mathcal{A}(\hat{\mathcal{P}}(t))} f(\hat{\mathcal{P}}(t), \tilde{\mathcal{P}}) \leq \beta(t, \delta)$ **do**
4:

$$
\pi^{t+1} = \begin{cases} \arg\min_{(a,i,j)} N_{ij}^a(t) \text{ if } U_t \neq \emptyset \text{ or } \exists a \in [M], \{i^*(a, \hat{\mathcal{P}}(t))\} = \emptyset \text{ or } \hat{\mathcal{D}}_1^a(t) = \emptyset \\ \arg\max_{(a,i,j)} t(\omega_{ij}^a)^*(\hat{\mathcal{P}}(t)) - N_{ij}^a(t) \end{cases}
$$

5:   Sample the $\pi^{t+1}$ and obtain one observation $Z_{t+1}$.
6:   Set $t \leftarrow t + 1$, and update $\hat{\mathcal{P}}(t)$, $\boldsymbol{\omega}^*(\hat{\mathcal{P}}(t))$, $U_t$, $N_{i(S+1)}^a(t)$, $N_{ij}^a(t)$.
7: **end while**
8: **Output.** Select $\hat{i}_\tau(a) = \arg\max_{i \in [K]} \hat{\mu}_i^a(\tau)$ s.t. $\hat{F}_{ij}^a(\tau) \geq \phi$, $\forall j \in [S]$, as the best arm for each task $a \in [M]$.

---

**Stopping rule and decision rule.** We derive the stopping rule using the generalized likelihood ratio test method. Define

$$
f(\hat{\mathcal{P}}(t), \tilde{\mathcal{P}}) = \sum_{a \in [M], i \in [K]} \left( N_{i(S+1)}^a(t) d(\hat{\mu}_i^a(t), \tilde{\mu}_i^a) + \sum_{j \in [S]} N_{ij}^a(t) d(\hat{F}_{ij}^a(t), \tilde{F}_{ij}^a) \right),
\tag{14}
$$

and let $\tau = \inf\{t \in \mathbb{N} : \inf_{\tilde{\mathcal{P}} \in \mathcal{A}(\hat{\mathcal{P}}(t))} f(\hat{\mathcal{P}}(t), \tilde{\mathcal{P}}) > \beta(t, \delta)\}$. The algorithm stops only if the GLS statistic exceeds the threshold $\beta(t, \delta)$. Lemma 4 establishes the statistical validity of the stopping rule. For the decision rule, we use $\hat{\mathcal{P}}(t)$ to determine the best arm for each task $a \in [M]$.

**Lemma 4.** *Given a fixed confidence level $\delta \in (0,1)$ and $\alpha > 1$. There exists a constant $C(\alpha, M, K, S)$ such that for any sampling rule $\{\pi^t\}_t$, using the stopping rule with $\beta(t, \delta) = \log\left(\frac{Ct^\alpha}{\delta}\right)$, then for all problem instance $\mathcal{P} \in \mathcal{S}$, $\mathbb{P}\left(\exists a \in [M], \hat{i}_\tau(a) \neq i^*(a)\right) \leq \delta$.*

**Sample complexity.** The following theorem establishes the sample complexity upper bound for Algorithm 1. Combined with Theorem 1, this allows us to conclude that Algorithm 1 is asymptotically optimal.

**Theorem 3.** *Under Assumptions 1-2, for problem instance $\mathcal{P} \in \mathcal{S}$, Algorithm 1 satisfies*

$$\limsup_{\delta \to 0} \frac{\mathbb{E}[\tau]}{\log(1/\delta)} \leq \mathcal{H}^*(\mathcal{P}). \tag{15}$$

**Remark 4.** *The intuition behind Theorem 3 is that as $\delta \to 0$, the exploration step ensures each metric is sampled infinitely often, while the sample complexity required to eliminate randomness becomes negligible. Meanwhile, the tracking procedure ensures the empirical sampling ratio converges to the static optimal ratio, making the upper bound asymptotically matching the lower bound.*

## 4.2 AN EFFICIENT STRATEGY

While the Track-and-Stop strategy is asymptotically optimal, it incurs high computational costs due to the need to solve an optimization problem at every iteration. To address this, we propose a more computationally efficient strategy based on the optimality conditions outlined in Theorem 2.

**Sampling rule.** For each task $a \in [M]$, define

$$s_{a,i,j}(\mathcal{P}, \boldsymbol{\omega}(t)) = \begin{cases} \omega^a_{i(S+1)}(t) d(\mu^a_i, \mu^a_{i,i^*(a)}) + \omega^a_{i^*(a)(S+1)}(t) d(\mu^a_{i^*(a)}, \mu^a_{i,i^*(a)}), & \text{if } i \in \mathcal{D}^a_1 \cup \mathcal{M}^a_1, j = [S+1] \\ \omega^a_{i^*(a)j}(t) d(F^a_{i^*(a)j}, \phi), & \text{if } i = i^*(a), j \in [S] \\ \omega^a_{ij_h}(t) d(F^a_{ij_h}, \phi), & \text{if } i \in \mathcal{D}^a_2 \cup \mathcal{M}^a_2, j = j_h \\ \infty, & \text{otherwise.} \end{cases} \tag{16}$$

The optimality conditions in Theorem 2 aim to equalize $s_{a,i,j}(\mathcal{P}, t)$, which is non-decreasing with respect to the corresponding sampling ratio. Consequently, a natural strategy can be derived by iteratively solving these conditions. Define

$$(a^t, i^t, j^t) = \underset{(a,i,j) \in [M] \times [K] \times [S+1]}{\arg\min} s_{a,i,j}(\mathcal{P}, \boldsymbol{\omega}(t)). \tag{17}$$

The sampling rule is to choose $(a^t, i^t, j^t)$ if $i^t \in \mathcal{D}^a_2 \cup \mathcal{M}^a_2 \cup \{i^*(a)\}$ or $i^t \in \mathcal{D}^a_1 \cup \mathcal{M}^a_1$ with

$$\frac{N^{a^t}_{i^*(a)j^t}(t)}{N^{a^t}_{i^*(a)j^t}(t) + N^{a^t}_{i^t j^t}(t)} > \frac{(\omega^{a^t}_{i^*(a)j^t})^*(\mathcal{P})}{(\omega^{a^t}_{i^*(a)j^t})^*(\mathcal{P}) + (\omega^{a^t}_{i^t j^t})^*(\mathcal{P})}, \tag{18}$$

and choose $(a^t, i^*(a), j^t)$ otherwise. In its implementation, we can use $\hat{\mathcal{P}}(t)$ to estimate $\mathcal{P}$ and adaptively update the sampling ratio. This strategy is similar to the BestChallenger method in Garivier & Kaufmann (2016), with an extension to the multi-task and risk-constrained setting based on the results in Theorem 2.

## 5 NUMERICAL EXPERIMENT

In this section, we conduct numerical experiments to evaluate the performance of the proposed algorithms. Additional experimental results and further details are provided in Appendix D.

Two experiments with different configurations are considered. The first experiment involves a Gaussian bandit, and the second involves a Bernoulli bandit. Each experiment consists of 2 tasks, 4 arms, and 1 constraint. Arm 1 is the best, while arms 2, 3, and 4 are suboptimal and infeasible with a better mean, and infeasible with a worse mean, respectively. The detailed parameters of the problem instances are summarized in Appendix D.2.

Since no existing methods can directly be applied to solve our problem, we propose the following benchmarks for comparison:

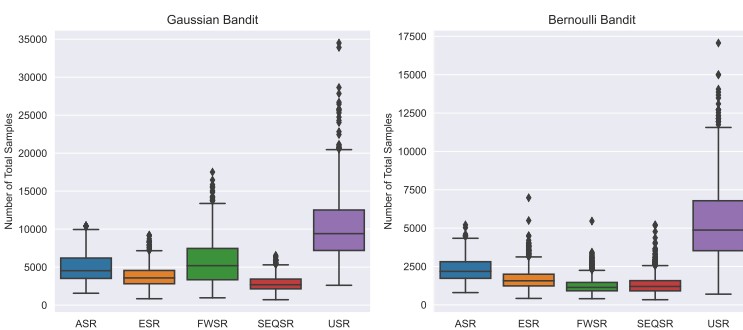

Figure 1: Empirical sample complexity for $1000$ runs with $\delta = 0.1$ and $n_0 = 10$ for Gaussian bandit (left) and Bernoulli bandit (right).

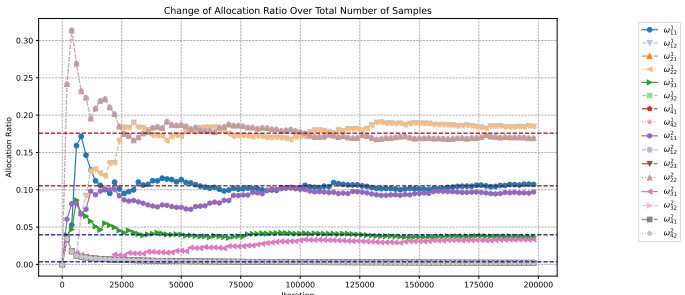

Figure 2: Change in empirical sampling ratio for ESR strategy in Gaussian bandit.

- Equal Sampling Rule (USR): sampling each task, arm, and constraint/objective pair equally.

- Approximate Sampling Rule (ASR): using the sampling ratio proposed in Lemma 3, which is proportional to the difficulty of identification.

- Frank-Wolf Sampling Rule (FWSR): extending the state-of-art strategy for unconstrained, single task BAI (Wang et al., 2021) to the multi-task BAI with risk constraint setting.

The pseudo-code for all strategies, along with the derivation of the Frank-Wolfe strategy, can be found in Appendix D.3. We refer to the Track-and-Stop strategy as the Exact Sampling Rule (ESR) and the efficient strategy in Section 4.2 as the Sequential Sampling Rule (SEQSR).

Figure 1 illustrates the empirical sample complexity for $1000$ independent runs of various strategies across two experiments, with $\delta = 0.1$ and $n_0 = 10$. For all experiments, we use the same stopping rule with $\beta(t, \delta) = \log(\log(t) + 1)/\delta$, which is suggested in Garivier & Kaufmann (2016), and is also used in Degenne et al. (2020) and (Wang et al., 2021). The SEQSR and ESR outperform or are comparable to other benchmarks. Figure 2 shows the change of empirical sampling ratio relative to the total sample size for ESR strategy in Gaussian bandit. As the total sample size increase, the empirical sampling ratio converges to the static optimal ratio, confirming the asymptotic optimality of the ESR sampling ratio. The conclusion remains consistent across different hyperparameters $n_0$ (D.5), confidence levels $\delta$ (D.6), and on a larger example (D.8). The computational efficiency of ASR, FWSR, SEQSR, and USR is comparable and significantly higher than that of ESR (D.7).

## 6 EXTENSION TO LINEAR BANDIT

In this section, we address the multi-task linear BAI problem with constraint and show that Theorem 1 extends naturally to the linear bandit setting.

In the linear bandit setting, arm $i$ corresponds to a vector $x_i \in \mathbb{R}^d$. For each task $a \in [M]$, arm $i$ has multiple performance metrics $(X_i^a, Y_{i1}^a, \ldots, Y_{iS}^a) \in \mathbb{R}^{S+1}$ that exhibit a linear structure. Another formulation that incorporates the linear structural information across tasks is provided in Theorem 6 of Appendix E.2.

**Assumption 3.** *There exist unknown parameters $\theta_1^a, \ldots, \theta_{S+1}^a \in \mathbb{R}^d$ satisfying $X_i^a = x_i^T \theta_{S+1}^a + \varepsilon_i^a$ and $Y_{ij}^a = x_i^T \theta_j^a + \varepsilon_{ij}^a$, where $\varepsilon_i^a \sim N(0, (\sigma_i^a)^2)$, $\varepsilon_{ij}^a \sim N(0, (\sigma_{ij}^a)^2)$, $\|x_i\|_2 \leq 1, \forall i \in [K]$, and $\|\theta_j^a\|_2 \leq 1, \forall a \in [M], j \in [S+1]$.*

The agent needs to solve the following optimization problem for each task

$$\max_{i \in [K]} x_i^T \theta_{S+1}^a \quad \text{s.t.} \ x_i^T \theta_j^a \leq b, \ \forall j \in [S]. \tag{19}$$

Theorem 4 extends Theorem 1 to the linear setting, providing a lower bound on the sample complexity $\mathbb{E}[\tau]$.

**Theorem 4.** *Given a fixed confidence level $\delta \in (0, 1)$. Under Assumption 3, for any linear BAI problem instance $\mathcal{P} \in \mathcal{S}$ and any strategy satisfying $\mathbb{P}\left(\forall a \in [M], i^*(a) = \hat{i}_\tau(a)\right) \geq 1 - \delta$,*

$$\mathbb{E}[\tau] \geq \mathcal{H}^*(\mathcal{P})kl(\delta, 1 - \delta), \tag{20}$$

*where $\Lambda_j^a = \sum_{i \in [K]} \omega_{ij}^a \frac{x_i x_i^T}{(\sigma_{ij}^a)^2}$, $\Lambda_{S+1}^a = \sum_{i \in [K]} \omega_{i(S+1)}^a \frac{x_i x_i^T}{(\sigma_i^a)^2}$,*

$$
\begin{aligned}
\mathcal{H}^*(\mathcal{P})^{-1} = \frac{1}{2} \sup_{\boldsymbol{\omega} \in \Omega} \min_{a \in [M]} \min & \left( \min_{j \in [S]} \frac{(b - x_{i^*(a)}^T \theta_j^a)^2}{x_{i^*(a)}^T (\Lambda_j^a)^{-1} x_{i^*(a)}}, \min_{i \in \mathcal{D}_1^a} \frac{((x_i - x_{i^*(a)})^T \theta_{S+1}^a)^2}{\|x_i - x_{i^*(a)}\|_{(\Lambda_{S+1}^a)^{-1}}^2}, \right. \\
& \left. \min_{i \in \mathcal{D}_2^a} \sum_{j \in \mathcal{B}_2^a(x_i)} \frac{(b - x_i^T \theta_j^a)^2}{x_i^T (\Lambda_j^a)^{-1} x_i}, \min_{i \in \mathcal{D}_3^a} \left[ \frac{((x_i - x_{i^*(a)})^T \theta_{S+1}^a)^2}{\|x_i - x_{i^*(a)}\|_{(\Lambda_{S+1}^a)^{-1}}^2} + \sum_{j \in \mathcal{B}_2^a(x_i)} \frac{(b - x_i^T \theta_j^a)^2}{x_i^T (\Lambda_j^a)^{-1} x_i} \right] \right).
\end{aligned}
\tag{21}
$$

# 7 CONCLUSION

We study the multi-task BAI problem with risk constraint in the fixed-confidence setting. We provide a lower bound on sample complexity, derive the optimality conditions for the optimal sampling ratio, extend the Track-and-Stop strategy with an upper bound asymptotically matching the lower bound, and derive an computationally efficient strategy. Numerical experiments demonstrate that our algorithm outperforms relatively well. Some potential future research directions include: 1) to derive the optimal strategy in the fixed-budget setting or to design efficient algorithms to address the optimization problem associated with the sample complexity lower bound; 2) to consider performance constraints on the arms as well as constraints on the sampling ratio, such as fairness and resource constraints.

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

## A    NOTATION AND MOTIVATING EXAMPLES

### A.1    TABLE OF NOTATION

Table 1 summarizes the notations and their meanings used throughout the paper.

### A.2    MOTIVATING EXAMPLES

In this subsection, we present motivating examples for our problem formulation.

- **Drug discovery.** In drug discovery, each drug (arm) has multiple metrics, like efficacy and side effects. The experimenter can choose one metric to evaluate via clinical experiments in each iteration. The goal is to identify the drug with the highest mean efficacy while ensuring side effect risk (quantile-based) stays below a threshold for each disease (task).
- **Supply chain management.** In supply chain management, each supplier (arm) has multiple metrics, such as total cost, delivery reliability, and product quality. Since monitoring a metric incurs additional costs, companies can evaluate one metric per iteration. The goal is to identify the supplier with the lowest mean total cost while ensuring the risk of significant delays or quality issues (quantile-based) stays below a threshold for each product (task).
- **Financial risk management.** In financial risk management, each investment strategy (arm) has multiple metrics, such as return, volatility, and drawdown. The decision-maker can evaluate one metric per iteration. The goal is to identify the strategy with the highest mean return while ensuring risk metrics (quantile-based) remain below a threshold under each market condition (task).

Table 1: Notation

| Notation | Meaning |
|---|---|
| $M, K, S$ | Number of tasks, arms, and constraints, respectively |
| $X_i^a$ | Random performance in the objective function of arm $i$ under task $a$ |
| $Y_{ij}^a$ | Random performance in the $j$-th constraint of arm $i$ under task $a$ |
| $F^a(\cdot), F_{ij}^a(\cdot)$ | Cumulative distribution function of $X_i^a$ and $Y_{ij}^a$, respectively |
| $\mu_i^a, \mu_{ij}^a$ | Expectation of $X_i^a$ and $Y_{ij}^a$, respectively |
| $q(Y_{ij}^a, \phi)$ | The $\phi$-quantile of $Y_{ij}^a$, with $\phi \in (0,1)$ |
| $\tau$ | Stopping time of the algorithm |
| $\hat{i}_\tau(a)$ | Arm identified as the best for task $a$ upon algorithm termination |
| $N_{ij}^a(t)$ | Number of samples of constraint $j$ for arm $i$ in task $a$ up to time $t$ |
| $N_{i(S+1)}^a(t)$ | Number of samples of the objective for arm $i$ in task $a$ up to time $t$ |
| $\omega_{ij}^a(t)$ | Sampling ratio up to time $t$ |
| $\delta$ | The confidence level $\delta \in (0,1)$ |
| $\mathcal{D}_1^a$ | Set of all suboptimal arms for task $a$ |
| $\mathcal{D}_2^a$ | Set of all infeasible arms with better objective value for task $a$ |
| $\mathcal{D}_3^a$ | Set of all infeasible arms with worse objective value for task $a$ |
| $\mathcal{B}_1^a(i)$ | Indices of feasible constraints for arm $i$ in task $a$ |
| $\mathcal{B}_2^a(i)$ | Indices of infeasible constraints for arm $i$ in task $a$ |
| $d(p,q)$ | KL divergence between two distributions with means $p$ and $q$ |
| $\mathcal{P}$ | A BAI problem instance $\mathcal{P} = (\boldsymbol{\mu}^a, \boldsymbol{Q}^a)_{a \in [M]}$ |
| $i^*(a, \mathcal{P})$ | Best arm under task $a$ in problem instance $\mathcal{P}$, often denoted by $i^*(a)$ |
| $\mathcal{A}(\mathcal{P})$ | Set of alternative problem instances of $\mathcal{P}$ |
| $j_h(a,i)$ | The most distinguishable infeasible constraint of arm $i$ in task $a$ |
| $\boldsymbol{\omega}^*(\mathcal{P})$ | The static optimal sampling rule for problem instance $\mathcal{P}$ |
| $\hat{\boldsymbol{\mu}}^a(t)$ | Empirical estimate of $\boldsymbol{\mu}^a = (\mu_i^a)_{i \in [K]}$ at time $t$ |
| $\hat{\boldsymbol{F}}^a(t)$ | Empirical estimate of $\boldsymbol{F}^a = \{F_{ij}^a\}_{i \in [K], j \in [S]}$ at time $t$ |
| $\hat{\mathcal{P}}(t)$ | Empirical estimate of instance $\mathcal{P}$ at time $t$, $\hat{\mathcal{P}}(t) = (\hat{\boldsymbol{\mu}}^a(t), \hat{\boldsymbol{F}}^a(t)_{a \in [M]}$ |
| $\hat{\mathcal{D}}_1^a(t), \hat{\mathcal{D}}_2^a(t), \hat{\mathcal{D}}_3^a(t)$ | Empirical estimate of sets $\mathcal{D}_1^a, \mathcal{D}_2^a, \mathcal{D}_3^a$ at time $t$ |

# B  SAMPLE COMPLEXITY ANALYSIS

## B.1  PROOF OF THEOREM 1

**Theorem 1.** *Given a fixed confidence level $\delta \in (0,1)$, define $\mu_{i,i^*(a)}^a = \arg\inf_{\tilde{\mu}} \omega_{i(S+1)}^a d(\mu_i^a, \tilde{\mu}) + \omega_{i^*(a)(S+1)}^a d(\mu_{i^*(a)}^a, \tilde{\mu})$. Under Assumptions 1-2, for any problem instance $\mathcal{P} \in \mathcal{S}$ and any strategy satisfying $\mathbb{P}\left(\forall a \in [M], i^*(a) = \hat{i}_\tau(a)\right) \geq 1 - \delta$,*

$$\mathbb{E}[\tau] \geq \mathcal{H}^*(\mathcal{P}) kl(\delta, 1-\delta), \tag{22}$$

*as $\delta \to 0$, we have*

$$\liminf_{\delta \to 0} \frac{\mathbb{E}[\tau]}{\log(1/\delta)} \geq \mathcal{H}^*(\mathcal{P}), \tag{23}$$

*where $\mathcal{H}^*(\mathcal{P})^{-1} = \max_{\boldsymbol{\omega} \in \Omega} \min_{a \in [M]} \min\left(V_1^a(\boldsymbol{\omega}), V_2^a(\boldsymbol{\omega}), V_3^a(\boldsymbol{\omega}), V_4^a(\boldsymbol{\omega})\right)$, with*

$$V_1^a(\boldsymbol{\omega}) = \min_{j \in [S]} \omega_{i^*(a)j}^a d(F_{i^*(a)j}^a, \phi),$$

$$V_2^a(\boldsymbol{\omega}) = \min_{i \in \mathcal{D}_1^a} \omega_{i(S+1)}^a d(\mu_i^a, \mu_{i,i^*(a)}^a) + \omega_{i^*(a)(S+1)}^a d(\mu_{i^*(a)}^a, \mu_{i,i^*(a)}^a),$$

$$V_3^a(\boldsymbol{\omega}) = \min_{i \in \mathcal{D}_2^a} \sum_{j \in \mathcal{B}_2^a(i)} \omega_{ij}^a d(F_{ij}^a, \phi),$$

$$V_4^a(\boldsymbol{\omega}) = \min_{i \in \mathcal{D}_3^a} \omega_{i(S+1)}^a d(\mu_i^a, \mu_{i,i^*(a)}^a) + \omega_{i^*(a)(S+1)}^a d(\mu_{i^*(a)}^a, \mu_{i,i^*(a)}^a) + \sum_{j \in \mathcal{B}_2^a(i)} \omega_{ij}^a d(F_{ij}^a, \phi).$$

*Proof.* Consider the original optimization problem in (2),

$$\max_{i \in [K]} \mathbb{E}[X_i^a] \quad \text{s.t.} \quad q(Y_{ij}^a, \phi) \leq b, \ \forall j \in [S]. \tag{24}$$

According to Assumption 2, the CDF $F_{ij}^a(\cdot)$ has a continuous density function, then by definition, we have $q(Y_{ij}^a, \phi) = (F_{ij}^a)^{-1}(\phi)$.

Therefore, the constraints are equivalent to

$$(F_{ij}^a)^{-1}(\phi) \leq b, \ \forall j \in [S], \tag{25}$$

which is also equivalent to

$$F_{ij}^a(b) \geq \phi, \ \forall j \in [S]. \tag{26}$$

By the definition of CDF, we have $F_{ij}^a(b) = \mathbb{E}[\mathbb{I}(Y_{ij}^a \leq b)]$, and the original optimization problem is equivalent to

$$\max_{i \in [K]} \mathbb{E}[X_i^a] \quad \text{s.t.} \quad \mathbb{E}[\mathbb{I}(Y_{ij}^a \leq b)] \geq \phi, \ \forall j \in [S], \tag{27}$$

where $\mathbb{I}(\cdot)$ is the indicator function, it takes on the value 1 if $\cdot$ is true, and 0 otherwise. For notational simplicity, we let $F_{ij}^a$ represent $F_{ij}^a(b)$.

Let $\mathcal{P} = (\boldsymbol{\mu}^a, \boldsymbol{F}^a)_{a \in [M]}$ represent a problem instance, $\mathcal{A}(\mathcal{P}) = \{\mathcal{P}' \in \mathcal{S} : \exists a \in [M], i^*(a, \mathcal{P}) \neq i^*(a, \mathcal{P}')\}$.

In the fixed confidence setting, for a problem instance $\mathcal{P} \in \mathcal{S}$, the algorithm needs to satisfy that,

$$\mathbb{P}(\exists a \in [M], \hat{i}_\tau(a) \neq i^*(a, \mathcal{P})) \leq \delta, \tag{28}$$

and for any problem instance $\tilde{\mathcal{P}} \in \mathcal{S}$,

$$\mathbb{P}(\exists a \in [M], \hat{i}_\tau(a) \neq i^*(a, \mathcal{P})) \geq 1 - \delta. \tag{29}$$

Define

$$f(\boldsymbol{\omega}, \mathcal{P}, \tilde{\mathcal{P}}) = \sum_{a \in [M]} \sum_{i \in [K]} \left( \omega_{i(S+1)}^a d(\mu_i^a, \tilde{\mu}_i^a) + \sum_{j \in [S]} \omega_{ij}^a d(F_{ij}^a, \tilde{F}_{ij}^a) \right). \tag{30}$$

Then, we can obtain that

$$\begin{aligned}
\sum_{a \in [M]} &\sum_{i \in [K]} \left( \mathbb{E}[N_{i(S+1)}^a(\tau)]d(\mu_i^a, \tilde{\mu}_i^a) + \sum_{j \in [S]} \mathbb{E}[N_{ij}^a(\tau)]d(F_{ij}^a, \tilde{F}_{ij}^a) \right) \\
&= \mathbb{E}[\tau] \sum_{a \in [M]} \sum_{i \in [K]} \left( \omega_{i(S+1)}^a d(\mu_i^a, \tilde{\mu}_i^a) + \sum_{j \in [S]} \omega_{ij}^a d(F_{ij}^a, \tilde{F}_{ij}^a) \right) \\
&= \mathbb{E}[\tau] f(\boldsymbol{\omega}, \mathcal{P}, \tilde{\mathcal{P}}).
\end{aligned} \tag{31}$$

According to the Lemma 1 of Kaufmann et al. (2016), we have

$$\mathbb{E}[\tau] f(\boldsymbol{\omega}, \mathcal{P}, \tilde{\mathcal{P}}) \geq \text{kl}(\delta, 1 - \delta), \forall \tilde{\mathcal{P}} \in \mathcal{A}(\mathcal{P}). \tag{32}$$

Then we can obtain that

$$\text{kl}(\delta, 1 - \delta) \leq \mathbb{E}[\tau] \inf_{\tilde{\mathcal{P}} \in \mathcal{A}(\mathcal{P})} f(\boldsymbol{\omega}, \mathcal{P}, \tilde{\mathcal{P}}) \leq \mathbb{E}[\tau] \sup_{\boldsymbol{\omega} \in \Omega} \inf_{\tilde{\mathcal{P}} \in \mathcal{A}(\mathcal{P})} f(\boldsymbol{\omega}, \mathcal{P}, \tilde{\mathcal{P}}). \tag{33}$$

According to (31), we have

$$\mathbb{E}[\tau] \sup_{\boldsymbol{\omega} \in \Omega} \inf_{\tilde{\mathcal{P}} \in \mathcal{A}(\mathcal{P})} \sum_{a \in [M]} \sum_{i \in [K]} \left( \omega_{i(S+1)}^a d(\mu_i^a, \tilde{\mu}_i^a) + \sum_{j \in [S]} \omega_{ij}^a d(F_{ij}^a, \tilde{F}_{ij}^a) \right) \geq \text{kl}(\delta, 1 - \delta). \tag{34}$$

Therefore, we conclude that

$$\mathbb{E}[\tau] \geq \mathcal{H}^*(\mathcal{P})\text{kl}(\delta, 1 - \delta), \tag{35}$$

where

$$\mathcal{H}^*(\mathcal{P})^{-1} = \sup_{\boldsymbol{\omega} \in \Omega} \inf_{\tilde{\mathcal{P}} \in \mathcal{A}(\mathcal{P})} \sum_{a \in [M]} \sum_{i \in [K]} \left( \omega_{i(S+1)}^a d(\mu_i^a, \tilde{\mu}_i^a) + \sum_{j \in [S]} \omega_{ij}^a d(F_{ij}^a, \tilde{F}_{ij}^a) \right). \tag{36}$$

Next, we provide a detailed analysis on $\mathcal{H}^*(\mathcal{P})$. For each task $a \in [M]$ and constraint $j \in [S]$, define

$$\mathcal{C}_{i^*(a,\mathcal{P})j}^a = \{(\tilde{\boldsymbol{\mu}}, \tilde{\boldsymbol{F}}) : \tilde{F}_{i^*(a,\mathcal{P})j}^a < \phi\}. \tag{37}$$

For each task $a \in [M]$ and arm $i \in [K] \setminus \{i^*(a,\mathcal{P})\}$, define

$$\mathcal{C}_i^a = \{(\tilde{\boldsymbol{\mu}}, \tilde{\boldsymbol{F}}) : \tilde{\mu}_i^a \geq \tilde{\mu}_{i^*(a,\mathcal{P})}^a, \tilde{F}_{ij}^a \geq \phi, \forall j \in [S]\}, \tag{38}$$

where $(\tilde{\boldsymbol{\mu}}, \tilde{\boldsymbol{F}}) = (\tilde{\boldsymbol{\mu}}^a, \tilde{\boldsymbol{F}}^a)_{a \in [M]} = \tilde{\mathcal{P}}$ denote the problem instance $\tilde{\mathcal{P}}$.

Then we can represent $\mathcal{A}(\mathcal{P})$ as the intersection of some subsets.

$$\begin{aligned}
\mathcal{A}(\mathcal{P}) &= \left\{ (\tilde{\boldsymbol{\mu}}, \tilde{\boldsymbol{F}}) : \exists a \in [M], i^*(a,\mathcal{P}) \neq i^*(a,\tilde{\mathcal{P}}) \right\} \\
&= \bigcup_{a \in [M]} \left\{ (\tilde{\boldsymbol{\mu}}, \tilde{\boldsymbol{F}}) : i^*(a,\mathcal{P}) \neq i^*(a,\tilde{\mathcal{P}}) \right\} \\
&= \bigcup_{a \in [M]} \left( \left\{ (\tilde{\boldsymbol{\mu}}, \tilde{\boldsymbol{F}}) : \exists j \in [S], \tilde{F}_{i^*(a,\mathcal{P})j}^a < \phi \right\} \bigcup \left\{ (\tilde{\boldsymbol{\mu}}, \tilde{\boldsymbol{F}}) : \begin{smallmatrix} \exists i \in [K] \setminus \{i^*(a,\mathcal{P})\}, \\ \tilde{\mu}_i^a \geq \tilde{\mu}_{i^*(a,\mathcal{P})}^a, \tilde{F}_{ij}^a \geq \phi, \forall j \in [S] \end{smallmatrix} \right\} \right) \\
&= \bigcup_{a \in [M]} \left( \left( \bigcup_{j \in [S]} \mathcal{C}_{i^*(a,\mathcal{P})j}^a \right) \bigcup \left( \bigcup_{i \in [K] \setminus \{i^*(a,\mathcal{P})\}} \mathcal{C}_i^a \right) \right). \tag{39}
\end{aligned}$$

Define $\mathcal{G}(\boldsymbol{\omega}, \mathcal{P}) = \inf_{\tilde{\mathcal{P}} \in \mathcal{A}(\mathcal{P})} f(\boldsymbol{\omega}, \mathcal{P}, \tilde{\mathcal{P}})$, according to (39), we have

$$\mathcal{G}(\boldsymbol{\omega}, \mathcal{P}) = \min_{a \in [M]} \min \left( \min_{j \in [S]} f_{i^*(a,\mathcal{P})j}^a(\boldsymbol{\omega}, \mathcal{P}), \min_{i \in [K] \setminus \{i^*(a,\mathcal{P})\}} f_i^a(\boldsymbol{\omega}, \mathcal{P}) \right), \tag{40}$$

where $f_{i^*(a,\mathcal{P})j}^a(\boldsymbol{\omega}, \mathcal{P}) = \inf_{\tilde{\mathcal{P}} \in \mathcal{C}_{i^*(a,\mathcal{P})j}^a} f(\boldsymbol{\omega}, \mathcal{P}, \tilde{\mathcal{P}})$, $f_i^a(\boldsymbol{\omega}, \mathcal{P}) = \inf_{\tilde{\mathcal{P}} \in \mathcal{C}_i^a} f(\boldsymbol{\omega}, \mathcal{P}, \tilde{\mathcal{P}})$.

By the definition of $\mathcal{C}_{i^*(a,\mathcal{P})j}^a$ and $\mathcal{C}_i^a$, we have

$$f_{i^*(a,\mathcal{P})j}^a(\boldsymbol{\omega}, \mathcal{P}) = \inf_{\tilde{\mathcal{P}} \in \mathcal{C}_{i^*(a,\mathcal{P})j}^a} f(\boldsymbol{\omega}, \mathcal{P}, \tilde{\mathcal{P}}) = \omega_{i^*(a,\mathcal{P})j}^a d(F_{i^*(a,\mathcal{P})j}^a, \phi), \tag{41}$$

and

$$\min_{i \in [K] \setminus \{i^*(a, \mathcal{P})\}} f_i^a(\boldsymbol{\omega}, \mathcal{P})$$

$$= \min_{i \in [K] \setminus \{i^*(a, \mathcal{P})\}} \inf_{\tilde{\mathcal{P}} \in \mathcal{C}_i^a} f(\boldsymbol{\omega}, \mathcal{P}, \tilde{\mathcal{P}})$$

$$= \min_{i \in [K] \setminus \{i^*(a, \mathcal{P})\}} \inf_{\tilde{\mathcal{P}} \in \mathcal{C}_i^a} \sum_{a \in [M]} \sum_{i \in [K]} \left( \omega_{i(S+1)}^a d(\mu_i^a, \tilde{\mu}_i^a) + \sum_{j \in [S]} \omega_{ij}^a d(F_{ij}^a, \tilde{F}_{ij}^a) \right)$$

$$= \min \left( \min_{i \in \mathcal{D}_1^a} \inf_{\tilde{\mathcal{P}} \in \mathcal{C}_i^a} \sum_{i \in [K]} \omega_{i(S+1)}^a d(\mu_i^a, \tilde{\mu}_i^a), \min_{i \in \mathcal{D}_2^a} \inf_{\tilde{\mathcal{P}} \in \mathcal{C}_i^a} \sum_{i \in [K], j \in [S]} \omega_{ij}^a d(F_{ij}^a, \tilde{F}_{ij}^a), \right.$$

$$\left. \min_{i \in \mathcal{D}_3^a} \inf_{\tilde{\mathcal{P}} \in \mathcal{C}_i^a} \sum_{i \in [K]} \left( \omega_{i(S+1)}^a d(\mu_i^a, \tilde{\mu}_i^a) + \sum_{j \in [S]} \omega_{ij}^a d(F_{ij}^a, \tilde{F}_{ij}^a) \right) \right)$$

$$= \min \left( \min_{i \in \mathcal{D}_1^a} \inf_{\tilde{\mu}_i^a > \tilde{\mu}_{i^*(a)}^a} \omega_{i(S+1)}^a d(\mu_i^a, \tilde{\mu}_i^a) + \omega_{i^*(a)(S+1)}^a d(\mu_{i^*(a)}^a, \tilde{\mu}_{i^*(a)}^a), \right.$$

$$\min_{i \in \mathcal{D}_2^a} \sum_{j \in \mathcal{B}_2^a(i)} \omega_{ij}^a d(F_{ij}^a, \phi),$$

$$\left. \min_{i \in \mathcal{D}_3^a} \inf_{\tilde{\mu}_i^a > \tilde{\mu}_{i^*(a)}^a} \omega_{i(S+1)}^a d(\mu_i^a, \tilde{\mu}_i^a) + \omega_{i^*(a)(S+1)}^a d(\mu_{i^*(a)}^a, \tilde{\mu}_{i^*(a)}^a) + \sum_{j \in \mathcal{B}_2^a(i)} \omega_{ij}^a d(F_{ij}^a, \phi) \right)$$

$$= \min(V_2^a(\boldsymbol{\omega}), V_3^a(\boldsymbol{\omega}), V_4^a(\boldsymbol{\omega})).$$

(42)

, where the last equality is from the Lemma 5.

Then, we conclude that

$$\mathcal{G}(\boldsymbol{\omega}, \mathcal{P}) = \min_{a \in [M]} \min \left( V_1^a(\boldsymbol{\omega}), V_2^a(\boldsymbol{\omega}), V_3^a(\boldsymbol{\omega}), V_4^a(\boldsymbol{\omega}) \right), \tag{43}$$

and

$$\mathcal{H}^*(\mathcal{P})^{-1} = \max_{\boldsymbol{\omega} \in \Omega} \min_{a \in [M]} \min \left( V_1^a(\boldsymbol{\omega}), V_2^a(\boldsymbol{\omega}), V_3^a(\boldsymbol{\omega}), V_4^a(\boldsymbol{\omega}) \right). \tag{44}$$

$\square$

## B.2 PROOF OF THEOREM 2

**Lemma 5.** *Define* $g(\boldsymbol{\omega}) = \inf_{\tilde{\mu}_i^a > \tilde{\mu}_{i^*(a)}^a} \omega_{i(S+1)}^a d(\mu_i^a, \tilde{\mu}_i^a) + \omega_{i^*(a)(S+1)}^a d(\mu_{i^*(a)}^a, \tilde{\mu}_{i^*(a)}^a)$, *and* $\mu_{i,i^*(a)}^a = \arg\inf_{\tilde{\mu}} \omega_{i(S+1)}^a d(\mu_i^a, \tilde{\mu}) + \omega_{i^*(a)(S+1)}^a d(\mu_{i^*(a)}^a, \tilde{\mu})$, *then we have*

$$g(\boldsymbol{\omega}) = \omega_{i(S+1)}^a d(\mu_i^a, \mu_{i,i^*(a)}^a) + \omega_{i^*(a)(S+1)}^a d(\mu_{i^*(a)}^a, \mu_{i,i^*(a)}^a),$$

$$\frac{\partial}{\partial \omega_{i(S+1)}^a} g(\boldsymbol{\omega}) = d(\mu_i^a, \mu_{i,i^*(a)}^a), \quad \frac{\partial}{\partial \omega_{i^*(a)(S+1)}^a} g(\boldsymbol{\omega}) = d(\mu_{i^*(a)}^a, \mu_{i,i^*(a)}^a).$$
(45)

*Proof.* Since both $d(\mu_i^a, \tilde{\mu})$ and $d(\mu_{i^*(a)}^a, \tilde{\mu})$ are decreasing in $(-\infty, \mu_i^a)$, and are increasing in $(\mu_{i^*(a)}^a, +\infty)$, then it is sufficient to search the infimum in the interval $[\mu_i^a, \mu_{i^*(a)}^a]$. Therefore, we have

$$g(\boldsymbol{\omega}) = \inf_{\tilde{\mu}_i^a > \tilde{\mu}_{i^*(a)}^a} \omega_{i(S+1)}^a d(\mu_i^a, \tilde{\mu}_i^a) + \omega_{i^*(a)(S+1)}^a d(\mu_{i^*(a)}^a, \tilde{\mu}_{i^*(a)}^a)$$

$$= \inf_{\tilde{\mu} \in [\mu_i^a, \mu_{i^*(a)}^a]} \omega_{i(S+1)}^a d(\mu_i^a, \tilde{\mu}) + \omega_{i^*(a)(S+1)}^a d(\mu_{i^*(a)}^a, \tilde{\mu})$$

$$= \inf_{\tilde{\mu}} \omega_{i(S+1)}^a d(\mu_i^a, \tilde{\mu}) + \omega_{i^*(a)(S+1)}^a d(\mu_{i^*(a)}^a, \tilde{\mu})$$

$$= \omega_{i(S+1)}^a d(\mu_i^a, \mu_{i,i^*(a)}^a) + \omega_{i^*(a)(S+1)}^a d(\mu_{i^*(a)}^a, \mu_{i,i^*(a)}^a).$$
(46)

In our analysis, we assume that $d(\mu_i^a, \tilde{\mu})$ is strictly convex function of $\tilde{\mu}, \forall a \in [M], i \in [K]$, which is satisfied by many distributions in SPEF such as Bernoulli, Gaussian with known variance and Gamma with known shape parameter (Juneja & Krishnasamy, 2019). Therefore, $\mu_{i,i^*(a)}^a$ is the unique solution of

$$\omega_{i(S+1)}^a \frac{\partial d(\mu_i^a, \mu_{i,i^*(a)}^a)}{\partial \mu_{i,i^*(a)}^a} + \omega_{i^*(a)(S+1)}^a \frac{\partial d(\mu_{i^*(a)}^a, \mu_{i,i^*(a)}^a)}{\partial \mu_{i,i^*(a)}^a} = 0. \tag{47}$$

Then we can obtain

$$\frac{\partial}{\partial \omega_{i(S+1)}^a} g(\boldsymbol{\omega})$$

$$= d(\mu_i^a, \mu_{i,i^*(a)}^a) + (\omega_{i(S+1)}^a \frac{\partial d(\mu_i^a, \mu_{i,i^*(a)}^a)}{\partial \mu_{i,i^*(a)}^a} + \omega_{i^*(a)(S+1)}^a \frac{\partial d(\mu_{i^*(a)}^a, \mu_{i,i^*(a)}^a)}{\partial \mu_{i,i^*(a)}^a}) \frac{\partial \mu_{i,i^*(a)}^a}{\partial \omega_{i(S+1)}^a} \tag{48}$$

$$= d(\mu_i^a, \mu_{i,i^*(a)}^a).$$

We can also obtain $\frac{\partial}{\partial \omega_{i^*(a)(S+1)}^a} g(\boldsymbol{\omega}) = d(\mu_{i^*(a)}^a, \mu_{i,i^*(a)}^a)$, which concludes the proof. $\qquad\square$

**Lemma 6.** *The optimization problem*

$$\max_{\boldsymbol{\omega} \in \Omega} \min_{a \in [M]} \min \left( V_1^a(\boldsymbol{\omega}), V_2^a(\boldsymbol{\omega}), V_3^a(\boldsymbol{\omega}), V_4^a(\boldsymbol{\omega}) \right), \tag{49}$$

*is a convex optimization problem.*

*Proof.* According to the proof of Theorem 1, we have

$$\mathcal{H}^*(\mathcal{P})^{-1} = \max_{\boldsymbol{\omega} \in \Omega} \mathcal{G}(\boldsymbol{\omega}, \mathcal{P})$$

$$= \max_{\boldsymbol{\omega} \in \Omega} \min_{a \in [M]} \min \left( \min_{j \in [S]} f_{i^*(a,\mathcal{P})j}^a(\boldsymbol{\omega}, \mathcal{P}), \min_{i \in [K] \setminus \{i^*(a,\mathcal{P})\}} f_i^a(\boldsymbol{\omega}, \mathcal{P}) \right) \tag{50}$$

$$= \max_{\boldsymbol{\omega} \in \Omega} \min_{a \in [M]} \min \left( \min_{j \in [S]} \inf_{\tilde{\mathcal{P}} \in \mathcal{C}_{i^*(a,\mathcal{P})j}^a} f(\boldsymbol{\omega}, \mathcal{P}, \tilde{\mathcal{P}}), \inf_{\tilde{\mathcal{P}} \in \mathcal{C}_i^a} f(\boldsymbol{\omega}, \mathcal{P}, \tilde{\mathcal{P}}) \right),$$

where

$$\mathcal{C}_{i^*(a,\mathcal{P})j}^a = \{(\tilde{\boldsymbol{\mu}}, \tilde{\boldsymbol{F}}) : \tilde{F}_{i^*(a,\mathcal{P})j}^a < \phi\}, \;\; \mathcal{C}_i^a = \{(\tilde{\boldsymbol{\mu}}, \tilde{\boldsymbol{F}}) : \tilde{\mu}_i^a \geq \tilde{\mu}_{i^*(a,\mathcal{P})}^a, \tilde{F}_{ij}^a \geq \phi, \forall j \in [S]\} \tag{51}$$

and

$$f(\boldsymbol{\omega}, \mathcal{P}, \tilde{\mathcal{P}}) = \sum_{a \in [M]} \sum_{i \in [K]} \left( \omega_{i(S+1)}^a d(\mu_i^a, \tilde{\mu}_i^a) + \sum_{j \in [S]} \omega_{ij}^a d(F_{ij}^a, \tilde{F}_{ij}^a) \right) \tag{52}$$

For each $\tilde{\mathcal{P}}$, $f(\boldsymbol{\omega}, \mathcal{P}, \tilde{\mathcal{P}})$ is a concave function of $\boldsymbol{\omega}$. Since $\mathcal{G}(\boldsymbol{\omega}, \mathcal{P})$ is the infimum of concave functions, it is also concave. $\Omega$ is a convex set, consequently, the corresponding maximization problem is a convex optimization problem. $\qquad\square$

**Theorem 2.** *Let* $\mathcal{M}_1^a = \{i \in \mathcal{D}_3^a : d(\mu_i^a, \mu_{i,i^*(a)}^a) > d(F_{ij_h}^a, \phi)\}$ *and* $\mathcal{M}_2^a = \{i \in \mathcal{D}_3^a : d(\mu_i^a, \mu_{i,i^*(a)}^a) < d(F_{ij_h}^a, \phi)\}$. *Assume that* $\mathcal{D}_3^a = \mathcal{M}_1^a \cup \mathcal{M}_2^a$ *for each task* $a \in [M]$, *then the*

*static optimal sampling ratio $\boldsymbol{\omega}^*$ satisfies:*

$$V_1^a(\boldsymbol{\omega}^*) = V_2^a(\boldsymbol{\omega}^*) = V_3^a(\boldsymbol{\omega}^*) = V_4^a(\boldsymbol{\omega}^*) = z^*, \forall a \in [M]$$

$$(\omega_{i^*(a)j}^a)^* d(F_{i^*(a)j}^a, \phi) = z^*, \forall a \in [M], j \in [S]$$

$$(\omega_{i(S+1)}^a)^* d(\mu_i^a, \mu_{i,i^*(a)}^a) + (\omega_{i^*(a)(S+1)}^a)^* d(\mu_{i^*(a)}^a, \mu_{i,i^*(a)}^a) = z^*, \forall a \in [M], i \in \mathcal{D}_1^a \cup \mathcal{M}_1^a$$

$$\sum_{i \in \mathcal{D}_1^a \cup \mathcal{M}_1^a} \frac{d(\mu_{i^*(a)}^a, \mu_{i,i^*(a)}^a)}{d(\mu_i^a, \mu_{i,i^*(a)}^a)} = 1, \forall a \in [M]$$

$$(\omega_{ij_h}^a)^* d(F_{ij_h}^a, \phi) = z^*, \forall a \in [M], i \in \mathcal{D}_2^a \cup \mathcal{M}_2^a$$

$$(\omega_{ij}^a)^* = 0, \forall a \in [M], i \in \mathcal{D}_1^a \cup \mathcal{M}_1^a, j \in [S]$$

$$(\omega_{ij}^a)^* = 0, \forall a \in [M], i \in \mathcal{D}_2^a \cup \mathcal{M}_2^a, j \neq j_h$$

$$\sum_{a \in [M]} \sum_{i \in [K]} \sum_{j \in [S+1]} (\omega_{ij}^a)^* = 1,$$

$$(\omega_{ij}^a)^* \geq 0, \forall a \in [M], i \in [K], j \in [S+1]. \tag{53}$$

*Proof.* The static optimal ratio can be obtained by solving the following optimization problem

$$\max_{\boldsymbol{\omega} \in \Omega} \min_{a \in [M]} \min \left( V_1^a(\boldsymbol{\omega}), V_2^a(\boldsymbol{\omega}), V_3^a(\boldsymbol{\omega}), V_4^a(\boldsymbol{\omega}) \right), \tag{54}$$

which is equivalent to

$$\max z$$
$$\text{s.t. } \omega_{i^*(a)j}^a d(F_{i^*(a)j}^a, \phi) \geq z, \ \ \forall a \in [M], j \in [S] \ \ (\lambda_{i^*(a)j}^a)$$
$$\omega_{i(S+1)}^a d(\mu_i^a, \mu_{i,i^*(a)}^a) + \omega_{i^*(a)(S+1)}^a d(\mu_{i^*(a)}^a, \mu_{i,i^*(a)}^a) \geq z, \ \ \forall a \in [M], i \in \mathcal{D}_1^a \ \ (\lambda_i^a)$$
$$\sum_{j \in \mathcal{B}_2^a(i)} \omega_{ij}^a d(F_{ij}^a, \phi) \geq z, \ \ \forall a \in [M], i \in \mathcal{D}_2^a \ \ (\lambda_i^a)$$
$$\omega_{i(S+1)}^a d(\mu_i^a, \mu_{i,i^*(a)}^a) + \omega_{i^*(a)(S+1)}^a d(\mu_{i^*(a)}^a, \mu_{i,i^*(a)}^a) + \sum_{j \in \mathcal{B}_2^a(i)} \omega_{ij}^a d(F_{ij}^a, \phi) \geq z, \tag{55}$$
$$\forall a \in [M], i \in \mathcal{D}_3^a \ \ (\lambda_i^a)$$
$$\sum_{a \in [M]} \sum_{i \in [K]} \sum_{j \in [S+1]} \omega_{ij}^a = 1, \ \ (\nu)$$
$$\omega_{ij}^a \geq 0, \forall a \in [M], i \in [K], j \in [S+1] \ \ (\rho_{ij}^a).$$

The KKT condition of this problem includes the following four part:

- The stationary condition:

$$\sum_{a \in [M]} \sum_{j \in [S]} \lambda_{i^*(a)j}^a + \sum_{a \in [M]} \sum_{i \in \mathcal{D}_1^a \cup \mathcal{D}_2^a \cup \mathcal{D}_3^a} \lambda_i^a = 1 \tag{56}$$

$$\lambda_{i^*(a)j}^a d(F_{i^*(a)j}^a, \phi) + \rho_{i^*(a)j}^a = \nu, \quad \forall a \in [M], j \in [S] \tag{57}$$

$$\lambda_i^a d(\mu_i^a, \mu_{i,i^*(a)}^a) + \rho_{i(S+1)}^a = \nu, \quad \forall a \in [M], i \in \mathcal{D}_1^a \cup \mathcal{D}_3^a \tag{58}$$

$$\sum_{i \in \mathcal{D}_1^a \cup \mathcal{D}_3^a} \lambda_i^a d(\mu_{i^*(a)}^a, \mu_{i,i^*(a)}^a) + \rho_{i^*(a)(S+1)}^a = \nu, \quad \forall a \in [M] \tag{59}$$

$$\lambda_i^a d(F_{ij}^a, \phi) + \rho_{ij}^a = \nu, \quad \forall a \in [M], i \in \mathcal{D}_2^a \cup \mathcal{D}_3^a, j \in \mathcal{B}_2^a(i) \tag{60}$$

- The complementary slackness condition:

$$\lambda_{i^*(a)j}^a \left( \omega_{i^*(a)j}^a d(F_{i^*(a)j}^a, \phi) - z \right) = 0, \quad \forall a \in [M], j \in [S] \tag{61}$$

$$\lambda_i^a \left( \omega_{i(S+1)}^a d(\mu_i^a, \mu_{i,i^*(a)}^a) + \omega_{i^*(a)(S+1)}^a d(\mu_{i^*(a)}^a, \mu_{i,i^*(a)}^a) - z \right) = 0, \tag{62}$$

$$\forall a \in [M], i \in \mathcal{D}_1^a \tag{63}$$

$$\lambda_i^a \left( \sum_{j \in \mathcal{B}_2^a(i)} \omega_{ij}^a d(F_{ij}^a, \phi) - z \right) = 0, \quad \forall a \in [M], i \in \mathcal{D}_2^a \tag{64}$$

$$\lambda_i^a \left( \omega_{i(S+1)}^a d(\mu_i^a, \mu_{i,i^*(a)}^a) + \omega_{i^*(a)(S+1)}^a d(\mu_{i^*(a)}^a, \mu_{i,i^*(a)}^a) \right.$$
$$\left. + \sum_{j \in \mathcal{B}_2^a(i)} \omega_{ij}^a d(F_{ij}^a, \phi) - z \right) = 0, \quad \forall a \in [M], i \in \mathcal{D}_3^a \tag{65}$$

$$\rho_{ij}^a \omega_{ij}^a = 0, \quad \forall a \in [M], i \in [K], j \in [S+1] \tag{66}$$

- The feasibility of dual problem:

$$\lambda_{i^*(a)j}^a \geq 0, \quad \forall a \in [M], j \in [S] \tag{67}$$

$$\lambda_i^a \geq 0, \quad \forall a \in [M], i \in \mathcal{D}_1^a \cup \mathcal{D}_2^a \cup \mathcal{D}_3^a \tag{68}$$

$$\rho_{ij}^a \geq 0, \quad \forall a \in [M], i \in [K], j \in [S+1]. \tag{69}$$

- The feasibility of the original problem (55).

We can derive the optimality equations in Theorem 2 by solving the KKT condition.

First, for each task $a \in [M]$, it is easy to observe that

$$(\omega_{ij}^a)^* = 0, \forall i \in \mathcal{D}_1^a, j \in [S]$$
$$(\omega_{ij}^a)^* = 0, \forall i \in \mathcal{D}_2^a \cup \mathcal{D}_3^a, j \in \mathcal{B}_1^a(i) \tag{70}$$
$$(\omega_{i(S+1))}^a)^* = 0, \forall i \in \mathcal{D}_2^a.$$

Since we can find a feasible solution $\omega_{ij}^a = 1/MK(S+1)$ and the corresponding objective value $z > 0$. Then any solution with $z = 0$ is not optimal.

According to the constraints in problem (55), we have $z = 0$ if $\exists a \in [M], j \in [S], \omega_{i^*(a)j}^a = 0$, or $\exists a \in [M], i \in \mathcal{D}_1^a, \omega_{i(S+1)}^a = 0$ or $\omega_{i^*(a)(S+1)}^a = 0$. Therefore, we have $(\omega_{i^*(a)j}^a)^* > 0, \forall a \in [M], j \in [S]$ and $(\omega_{i(S+1)}^a)^* > 0, (\omega_{i^*(a)(S+1)}^a)^* > 0, \forall a \in [M], i \in \mathcal{D}_1^a$. According to (66), we have $\rho_{i^*(a)j}^a = 0, \forall a \in [M], j \in [S]$ and $\rho_{i(S+1)}^a = 0, \rho_{i^*(a)(S+1)}^a = 0, \forall a \in [M], i \in \mathcal{D}_1^a$.

Consider the arm $i \in \mathcal{D}_2^a \cup \mathcal{D}_3^a$, according to (60), we have

$$\lambda_i^a d(F_{ij}^a, \phi) + \rho_{ij}^a = \nu, \quad \forall a \in [M], i \in \mathcal{D}_2^a \cup \mathcal{D}_3^a, j \in \mathcal{B}_2^a(i). \tag{71}$$

Since $j_h = \arg\max_{j \in \mathcal{B}_2^a(i)} d(F_{ij}^a, \phi)$, then in order to satisfy (60), we have

$$\rho_{ij}^a > 0, \forall a \in [M], i \in \mathcal{D}_2^a \cup \mathcal{D}_3^a, j \in \mathcal{B}_2^a(i), j \neq j_h. \tag{72}$$

According to (66), we have

$$(\omega_{ij}^a)^* = 0, \forall a \in [M], i \in \mathcal{D}_2^a \cup \mathcal{D}_3^a, j \in \mathcal{B}_2^a(i) \setminus \{j_h\}. \tag{73}$$

By combining (70) and (73), we can obtain that

$$(\omega_{ij}^a)^* = 0, \forall i \in \mathcal{D}_2^a, j \neq j_h. \tag{74}$$

Now, we prove that $\lambda_{i^*(a)j}^a > 0, \forall a \in [M], j \in [S]$, and $\lambda_i^a > 0, \forall a \in [M], i \in \mathcal{D}_1^a \cup \mathcal{D}_2^a \cup \mathcal{D}_3^a$.

- $\lambda^a_{i^*(a)j} > 0, \forall a \in [M], j \in [S]$:

  Since we know that $(\omega^a_{i^*(a)j})^* > 0$ and $\rho^a_{i^*(a)j} = 0$, assume that $\exists a \in [M], j \in [S], \lambda^a_{i^*(a)j} = 0$, then according to (57), we have $\nu = 0$, which will lead to $\lambda^a_{i^*(a)j} = 0, \forall a \in [M], j \in [S]$ and $\lambda^a_i = 0, \forall a \in [M], i \in \mathcal{D}^a_1 \cup \mathcal{D}^a_2 \cup \mathcal{D}^a_3$ in order to satisfy (57)-(60). This contradicts (56), and we conclude that $\lambda^a_{i^*(a)j} > 0, \forall a \in [M], j \in [S]$.

- $\lambda^a_i > 0, \forall a \in [M], i \in \mathcal{D}^a_1 \cup \mathcal{D}^a_2$:

  For $i \in \mathcal{D}^a_1$, since we know that $(\omega^a_{i(S+1)})^* > 0$ and $\rho^a_{i(S+1)} = 0$, assume that $\exists a \in [M], i \in \mathcal{D}^a_1, \lambda^a_i = 0$, then according to (58), we have $\nu = 0$, which will lead to $\lambda^a_{i^*(a)j} = 0, \forall a \in [M], j \in [S]$ and $\lambda^a_i = 0, \forall a \in [M], i \in \mathcal{D}^a_1 \cup \mathcal{D}^a_2 \cup \mathcal{D}^a_3$ in order to satisfy (57)-(60). This contradicts (56), and we conclude that $\lambda^a_i > 0, \forall a \in [M], i \in \mathcal{D}^a_1$. For $i \in \mathcal{D}^a_2$, the analysis is the same, hence we omit it.

- $\lambda^a_i > 0, \forall a \in [M], i \in \mathcal{D}^a_3$:

  It is easy to see that $(\omega^a_{i(S+1)})^* > 0, (\omega^a_{i^*(a)(S+1)})^* > 0$ or $(\omega^a_{ij_h})^* > 0$, otherwise, we will have $z = 0$, which will not be the optimal solution. Assume that $\exists a \in [M], i \in \mathcal{D}^a_3, \lambda^a_i = 0$, then for the first case $(\omega^a_{i(S+1)})^* > 0, (\omega^a_{i^*(a)(S+1)})^* > 0$, according to (66), we have $\rho^a_{i(S+1)} = 0$. According to (58), we have $\nu = 0$, which will lead to $\lambda^a_{i^*(a)j} = 0, \forall a \in [M], j \in [S]$ and $\lambda^a_i = 0, \forall a \in [M], i \in \mathcal{D}^a_1 \cup \mathcal{D}^a_2 \cup \mathcal{D}^a_3$ in order to satisfy (57)-(60). This contradicts (56). For the second case $(\omega^a_{ij_h})^* > 0$, according to (66), we have $\rho^a_{ij_h} = 0$. Then using (60), we can obtain $v = 0$, which will lead to $\lambda^a_{i^*(a)j} = 0, \forall a \in [M], j \in [S]$ and $\lambda^a_i = 0, \forall a \in [M], i \in \mathcal{D}^a_1 \cup \mathcal{D}^a_2 \cup \mathcal{D}^a_3$ in order to satisfy (57)-(60). This contradicts (56). And we conclude that $\lambda^a_i > 0, \forall a \in [M], i \in \mathcal{D}^a_3$.

Since $\lambda^a_{i^*(a)j} > 0, \forall a \in [M], j \in [S]$, and $\lambda^a_i > 0, \forall a \in [M], i \in \mathcal{D}^a_1 \cup \mathcal{D}^a_2 \cup \mathcal{D}^a_3$, according to the complementary slackness condition, we have

$$\omega^a_{i^*(a)j} d(F^a_{i^*(a)j}, \phi) = z, \quad \forall a \in [M], j \in [S] \tag{75}$$

$$\omega^a_{i(S+1)} d(\mu^a_i, \mu^a_{i,i^*(a)}) + \omega^a_{i^*(a)(S+1)} d(\mu^a_{i^*(a)}, \mu^a_{i,i^*(a)}) = z, \tag{76}$$

$$\forall a \in [M], i \in \mathcal{D}^a_1 \tag{77}$$

$$\omega^a_{ij_h} d(F^a_{ij_h}, \phi) = z, \quad \forall a \in [M], i \in \mathcal{D}^a_2 \tag{78}$$

$$\omega^a_{i(S+1)} d(\mu^a_i, \mu^a_{i,i^*(a)}) + \omega^a_{i^*(a)(S+1)} d(\mu^a_{i^*(a)}, \mu^a_{i,i^*(a)})$$
$$+ \omega^a_{ij_h} d(F^a_{ij_h}, \phi) = z, \quad \forall a \in [M], i \in \mathcal{D}^a_3 \tag{79}$$

For each task $a \in [M]$, consider arm $i \in \mathcal{D}^a_3$, they need to satisfy

$$\lambda^a_i d(\mu^a_i, \mu^a_{i,i^*(a)}) + \rho^a_{i(S+1)} = \nu, \quad \forall a \in [M], i \in \mathcal{D}^a_3$$
$$\lambda^a_i d(F^a_{ij_h}, \phi) + \rho^a_{ij_h} = \nu, \quad \forall a \in [M], i \in \mathcal{D}^a_3 \tag{80}$$

For $i \in \mathcal{M}^a_1$ with $d(\mu^a_i, \mu^a_{i,i^*(a)}) > d(F^a_{ij_h}, \phi)$, then using (80), we have $\rho^a_{ij_h} > 0$. According to (66), we have $(\omega^a_{ij_h})^* = 0$. Then $(\omega^a_{i(S+1)})^* > 0$, and $(\omega^a_{i^*(a)(S+1)})^* > 0$. In this case, (79) is equivalent to

$$\omega^a_{i(S+1)} d(\mu^a_i, \mu^a_{i,i^*(a)}) + \omega^a_{i^*(a)(S+1)} d(\mu^a_{i^*(a)}, \mu^a_{i,i^*(a)}) = z, \forall a \in [M], i \in \mathcal{M}^a_1. \tag{81}$$

In this case, we have

$$\lambda^a_i d(\mu^a_i, \mu^a_{i,i^*(a)}) = \nu, \quad \forall a \in [M], i \in \mathcal{D}^a_1 \cup \mathcal{M}^a_1$$
$$\sum_{i \in \mathcal{D}^a_1 \cup \mathcal{M}^a_1} \lambda^a_i d(\mu^a_{i^*(a)}, \mu^a_{i,i^*(a)}) = \nu, \quad \forall a \in [M] \tag{82}$$

which can be simplified to

$$\sum_{i \in \mathcal{D}^a_1 \cup \mathcal{M}^a_1} \frac{d(\mu^a_{i^*(a)}, \mu^a_{i,i^*(a)})}{d(\mu^a_i, \mu^a_{i,i^*(a)})} = 1. \tag{83}$$

For $i \in \mathcal{M}_2^a$ with $d(\mu_i^a, \mu_{i,i^*(a)}^a) < d(F_{ij_h}^a, \phi)$, then using (80), we have $\rho_{i(S+1)}^a > 0$. According to (66), we have $(\omega_{i(S+1)}^a)^* = 0$. Then $(\omega_{ij_h}^a)^* > 0$, and (79) is equivalent to

$$\omega_{ij_h}^a d(F_{ij_h}^a, \phi) = z, \forall a \in [M], i \in \mathcal{M}_2^a. \tag{84}$$

According to Lemma 6 and the fact that the KKT conditions are sufficient for a global optimal solution of a convex optimization problem, we conclude the proof of Theorem 2. □

### B.3 SINGLE-PARAMETER EXPONENTIAL FAMILY

*The single-parameter exponential family can be defined as*

$$\Gamma = \left\{ (p_\eta)_{\eta \in \mathcal{K}} : dp_\eta(x) = \exp(\eta x - \Lambda(\eta)) d\rho(x) \right\}, \tag{85}$$

*where $\rho$ denote the reference measure on $\mathbb{R}$, $\Lambda(\eta) = \log \int_{x \in \mathbb{R}} \exp(\eta x) d\rho(x)$, $\mathcal{K} = \{\eta : \Lambda(\eta) < \infty\} \subset \mathbb{R}$. According to Dembo (2009), $\Lambda(\cdot) : \mathcal{K} \to \mathbb{R}$ is convex and $\mathcal{C}^\infty$ in $\mathcal{K}^\circ$. Proposition 2 provides some useful properties:*

**Proposition 2.** *Define $\Lambda^*(\theta) = \sup_{\eta \in \mathcal{K}} (\eta\theta - \Lambda(\eta))$, and let $\mu = \int x dp_\eta, \nu = \int x dp_\gamma$ denote the mean of distribution $p_\eta$ and $p_\gamma$, respectively, $KL(\cdot, \cdot)$ denote the Kullback-Leibler divergence between two distributions, then*

*(1) $\mu = \Lambda'(\eta), \forall \eta \in \mathcal{K}^\circ$.*

*(2) $\eta = \Lambda^{*\prime}(\mu)$ and $\Lambda^*(\mu) + \Lambda(\eta) = \mu\eta$.*

*(3) $\forall \eta, \gamma \in \mathcal{K}^\circ$,*

$$d(\mu, \nu) = KL(p_\eta, p_\gamma) = \Lambda(\gamma) - \Lambda(\eta) - \mu(\gamma - \eta) = \Lambda^*(\mu) - \Lambda^*(\nu) - \gamma(\mu - \nu). \tag{86}$$

*(4) $\left.\frac{\partial d(\mu, \nu)}{\partial \nu}\right|_{\nu=\mu} = 0, \left.\frac{\partial^2 d(\mu, \nu)}{\partial^2 \nu}\right|_{\nu=\mu} = \frac{1}{\sigma^2}$, where $\sigma^2$ is the variance of distribution $p_\eta$.*

*Proof.* (1) By definition, we have

$$\Lambda'(\eta) = \frac{\int_{x \in \mathbb{R}} x \exp(\eta x) \rho(x) dx}{\int_{x \in \mathbb{R}} \exp(\eta x) \rho(x) dx} = \int_{x \in \mathbb{R}} x \exp(\eta x - \Lambda(\eta)) \rho(x) dx = \mu. \tag{87}$$

(2) Since

$$\Lambda^*(\theta) = \sup_{\eta \in \mathcal{K}} (\eta\theta - \Lambda(\eta)) = \eta^*\theta - \Lambda(\eta^*), \tag{88}$$

where $\eta^*$ satisfy that $\theta = \Lambda'(\eta^*)$. Let $\theta = \mu$, we have $\mu = \Lambda'(\eta^*) = \Lambda'(\eta)$ according to (1). Then $\Lambda^{*\prime}(\mu) = \eta$ and $\Lambda^*(\mu) + \Lambda(\eta) = \mu\eta$.

(3) By definition of the KL divergence, we have

$$\begin{aligned}
d(\mu, \nu) = KL(p_\eta, p_\gamma) &= \mathbb{E}_{p_\eta} \left[ \log \left( \frac{dp_\eta(x)}{dp_\gamma(x)} \right) \right] \\
&= \mathbb{E}_{p_\eta} \left[ \log \left( \frac{\exp(\eta x - \Lambda(\eta))}{\exp(\gamma x - \Lambda(\gamma))} \right) \right] \\
&= \mathbb{E}_{p_\eta} \left[ (\eta - \gamma)x + \Lambda(\gamma) - \Lambda(\eta) \right] \\
&= \Lambda(\gamma) - \Lambda(\eta) - \mu(\gamma - \eta).
\end{aligned} \tag{89}$$

Notice that, $\mu = \Lambda'(\eta), \nu = \Lambda'(\gamma), \eta = \Lambda^{*\prime}(\mu), \gamma = \Lambda^{*\prime}(\nu)$, then it is straight to see that $d(\mu, \nu) = \Lambda^*(\mu) - \Lambda^*(\nu) - \gamma(\mu - \nu)$.

(4) According to (3), we have $d(\mu, \nu) = \Lambda^*(\mu) - \Lambda^*(\nu) - \gamma(\mu - \nu)$, then by definition

$$
\begin{aligned}
\frac{\partial d(\mu, \nu)}{\partial \nu} &= -\Lambda^{*\prime}(\nu) - \frac{\partial \gamma}{\partial \nu}(\mu - \nu) + \gamma \\
&= -\Lambda^{*\prime}(\nu) - \frac{\partial \gamma}{\partial \nu}(\mu - \nu) + \Lambda^{*\prime}(\nu) \\
&= \frac{\partial \gamma}{\partial \nu}(\nu - \mu),
\end{aligned}
\tag{90}
$$

and we have $\left.\frac{\partial d(\mu, \nu)}{\partial \nu}\right|_{\nu = \mu} = 0$. Furthermore,

$$
\frac{\partial^2 d(\mu, \nu)}{\partial^2 \nu} = \frac{\partial^2 \gamma}{\partial^2 \nu}(\nu - \mu) + \frac{\partial \gamma}{\partial \nu}.
\tag{91}
$$

Since $\gamma = \Lambda^{*\prime}(\nu)$, using the Danskin's Theorem, we have

$$
\frac{\partial \gamma}{\partial \nu} = \Lambda^{*\prime\prime}(\nu).
\tag{92}
$$

Therefore, we have

$$
\begin{aligned}
\left.\frac{\partial^2 d(\mu, \nu)}{\partial^2 \nu}\right|_{\nu = \mu} &= \Lambda^{*\prime\prime}(\mu) \\
&= \int_{x \in \mathbb{R}} x \exp(\eta x - \Lambda(\eta))(x - \Lambda'(\eta))\rho(x) dx \\
&= \int_{x \in \mathbb{R}} x^2 \exp(\eta x - \Lambda(\eta))\rho(x) dx - \mu \int_{x \in \mathbb{R}} x \exp(\eta x - \Lambda(\eta))\rho(x) dx \\
&= \mathbb{E}_{p_\eta}[X^2] - \mu^2 \\
&= \frac{1}{\sigma^2}.
\end{aligned}
\tag{93}
$$

$\square$

## B.4 PROOF OF LEMMA 1

**Lemma 1.** *Define* $\triangle_i^a = \mu_{i^*(a)}^a - \mu_i^a$, $\triangle_{i^*(a)}^a = \min_{l \in \mathcal{D}_1^a \cup \mathcal{M}_1^a}(\mu_{i^*(a)}^a - \mu_l^a)$, $\sigma_i^a = (\int x^2 dF_i^a(x) - (\mu_i^a)^2)^{\frac{1}{2}} \in (0, \infty)$, *then for each task* $a \in [M]$, *and arm* $i \in \mathcal{D}_1^a \cup \mathcal{M}_1^a$, *as* $\triangle_i^a \to 0$,

$$
\inf_{\tilde{\mu}_i^a > \tilde{\mu}_{i^*(a)}^a} \omega_{i(S+1)}^a d(\mu_i^a, \tilde{\mu}_i^a) + \omega_{i^*(a)(S+1)}^a d(\mu_{i^*(a)}^a, \tilde{\mu}_{i^*(a)}^a) \to \frac{(\mu_i^a - \mu_{i^*(a)}^a)^2}{2\left(\frac{(\sigma_i^a)^2}{\omega_{i(S+1)}^a} + \frac{(\sigma_{i^*(a)}^a)^2}{\omega_{i^*(a)(S+1)}^a}\right)}.
\tag{94}
$$

*Proof.* The proof is motivated by the Theorem 2 of Shin et al. (2018). Define

$$
g(\boldsymbol{\omega}) = \inf_{\tilde{\mu}_i^a > \tilde{\mu}_{i^*(a)}^a} \omega_{i(S+1)}^a d(\mu_i^a, \tilde{\mu}_i^a) + \omega_{i^*(a)(S+1)}^a d(\mu_{i^*(a)}^a, \tilde{\mu}_{i^*(a)}^a),
\tag{95}
$$

According to Lemma 5, we have

$$
g(\boldsymbol{\omega}) = \inf_{\tilde{\mu} \in [\mu_i^a, \mu_{i^*(a)}^a]} \omega_{i(S+1)}^a d(\mu_i^a, \tilde{\mu}) + \omega_{i^*(a)(S+1)}^a d(\mu_{i^*(a)}^a, \tilde{\mu})
\tag{96}
$$

Applying the second order Taylor Extension for $d(\mu_i^a, \tilde{\mu})$ at $\tilde{\mu} = \mu_i^a$, we can obtain that

$$
d(\mu_i^a, \tilde{\mu}) = d(\mu_i^a, \mu_i^a) + (\tilde{\mu} - \mu_i^a)\left.\frac{\partial d(\mu_i^a, \tilde{\mu})}{\partial \tilde{\mu}}\right|_{\tilde{\mu} = \mu_i^a} + \frac{(\tilde{\mu} - \mu_i^a)^2}{2}\left.\frac{\partial^2 d(\mu_i^a, \tilde{\mu})}{\partial^2 \tilde{\mu}}\right|_{\tilde{\mu} = \mu_i^a} + o((\tilde{\mu} - \mu_i^a)^2)
\tag{97}
$$

According to Proposition 2, we have

$$d(\mu_i^a, \tilde{\mu}) = \frac{(\tilde{\mu} - \mu_i^a)^2}{2(\sigma_i^a)^2} + o((\tilde{\mu} - \mu_i^a)^2), \tag{98}$$

By the same argument, we can obtain

$$d(\mu_{i^*(a)}^a, \tilde{\mu}) = \frac{(\tilde{\mu} - \mu_{i^*(a)}^a)^2}{2(\sigma_{i^*(a)}^a)^2} + o((\tilde{\mu} - \mu_{i^*(a)}^a)^2), \tag{99}$$

Define

$$f(\boldsymbol{\omega}, \tilde{\mu}) = \omega_{i(S+1)}^a d(\mu_i^a, \tilde{\mu}) + \omega_{i^*(a)(S+1)}^a d(\mu_{i^*(a)}^a, \tilde{\mu}),$$

$$\zeta(\boldsymbol{\omega}, \tilde{\mu}) = \omega_{i(S+1)}^a \frac{(\tilde{\mu} - \mu_i^a)^2}{2(\sigma_i^a)^2} + \omega_{i^*(a)(S+1)}^a \frac{(\tilde{\mu} - \mu_{i^*(a)}^a)^2}{2(\sigma_{i^*(a)}^a)^2},$$

$$h(\boldsymbol{\omega}) = \inf_{\tilde{\mu} \in [\mu_i^a, \mu_{i^*(a)}^a]} \zeta(\boldsymbol{\omega}, \tilde{\mu}) = \frac{(\mu_i^a - \mu_{i^*(a)}^a)^2}{2(\frac{(\sigma_i^a)^2}{\omega_{i(S+1)}^a} + \frac{(\sigma_{i^*(a)}^a)^2}{\omega_{i^*(a)(S+1)}^a})}. \tag{100}$$

Since $\tilde{\mu} \in [\mu_i^a, \mu_{i^*(a)}^a]$, we can obtain that

$$f(\boldsymbol{\omega}, \tilde{\mu}) = \zeta(\boldsymbol{\omega}, \tilde{\mu}) + o((\mu_{i^*(a)}^a - \mu_i^a)^2) \tag{101}$$

Let $\tilde{\mu}_1 = \arg\inf_{\tilde{\mu} \in [\mu_i^a, \mu_{i^*(a)}^a]} f(\boldsymbol{\omega}, \tilde{\mu})$, $\tilde{\mu}_2 = \arg\inf_{\tilde{\mu} \in [\mu_i^a, \mu_{i^*(a)}^a]} \zeta(\boldsymbol{\omega}, \tilde{\mu})$. Then, we have $g(\boldsymbol{\omega}) = f(\boldsymbol{\omega}, \tilde{\mu}_1)$, $h(\boldsymbol{\omega}) = \zeta(\boldsymbol{\omega}, \tilde{\mu}_2)$. Therefore, we can obtain

$$\begin{aligned}
|g(\boldsymbol{\omega}) - h(\boldsymbol{\omega})| &= |f(\boldsymbol{\omega}, \tilde{\mu}_1) - \zeta(\boldsymbol{\omega}, \tilde{\mu}_2)| \\
&\leq |f(\boldsymbol{\omega}, \tilde{\mu}_1) - \zeta(\boldsymbol{\omega}, \tilde{\mu}_1)| + |\zeta(\boldsymbol{\omega}, \tilde{\mu}_1) - \zeta(\boldsymbol{\omega}, \tilde{\mu}_2)| \\
&= o((\mu_{i^*(a)}^a - \mu_i^a)^2) + \left| (\tilde{\mu}_1 - \tilde{\mu}_2) \frac{\partial \zeta(\boldsymbol{\omega}, \tilde{\mu}_1)}{\partial \tilde{\mu}_1} \right|_{\tilde{\mu}_1 = \tilde{\mu}_2} + o((\tilde{\mu}_1 - \tilde{\mu}_2)) \right| \\
&= o((\mu_{i^*(a)}^a - \mu_i^a)^2).
\end{aligned} \tag{102}$$

Therefore, we conclude that, as $\triangle_i^a \to 0$,

$$g(\boldsymbol{\omega}) \to h(\boldsymbol{\omega}), \tag{103}$$

which means

$$\inf_{\tilde{\mu}_i^a > \tilde{\mu}_{i^*(a)}^a} \omega_{i(S+1)}^a d(\mu_i^a, \tilde{\mu}_i^a) + \omega_{i^*(a)(S+1)}^a d(\mu_{i^*(a)}^a, \tilde{\mu}_{i^*(a)}^a) \to \frac{(\mu_i^a - \mu_{i^*(a)}^a)^2}{2(\frac{(\sigma_i^a)^2}{\omega_{i(S+1)}^a} + \frac{(\sigma_{i^*(a)}^a)^2}{\omega_{i^*(a)(S+1)}^a})}. \tag{104}$$

$\square$

## B.5 PROOF OF LEMMA 2

**Lemma 2.** *Define* $\triangle_{ij}^a = |F_{ij}^a - \phi|$, $\sigma_{ij}^a = (F_{ij}^a(1 - F_{ij}^a))^{\frac{1}{2}} \in (0, \infty)$, *then for each task* $a \in [M]$, *arm* $i \in [K]$ *and constraint* $j \in [S]$, *as* $\triangle_{ij}^a \to 0$,

$$d(F_{ij}^a, \phi) \to \frac{(F_{ij}^a - \phi)^2}{2(\sigma_{ij}^a)^2}. \tag{105}$$

*Proof.* Applying the second order Taylor Extension for $d(F_{ij}^a, \phi)$ at $\phi = F_{ij}^a$, we can obtain that

$$\begin{aligned}
d(F_{ij}^a, \phi) &= d(F_{ij}^a, F_{ij}^a) + (\phi - F_{ij}^a) \frac{\partial(F_{ij}^a, \phi)}{\partial \phi} \bigg|_{\phi = F_{ij}^a} \\
&\quad + \frac{(\phi - F_{ij}^a)^2}{2} \frac{\partial^2 d(F_{ij}^a, \phi)}{\partial \phi} \bigg|_{\phi = F_{ij}^a} + o((\phi - F_{ij}^a)^2) \\
&= \frac{(F_{ij}^a - \phi)^2}{2(\sigma_{ij}^a)^2} + o((\phi - F_{ij}^a)^2).
\end{aligned} \tag{106}$$

as $\triangle_{ij}^a \to 0$, we have $d(F_{ij}^a, \phi) \to \frac{(F_{ij}^a - \phi)^2}{2(\sigma_{ij}^a)^2}$. $\qquad \square$

### B.6 PROOF OF PROPOSITION 1

**Proposition 1.** *Define*

$$\mathcal{H} = \sum_{a \in [M]} \left( \sum_{j \in [S]} \frac{(\sigma_{i^*(a)j}^a)^2}{(\triangle_{i^*(a)j}^a)^2} + \sum_{i \in \mathcal{D}_1^a \cup \mathcal{M}_1^a \cup \{i^*(a)\}} \frac{(\sigma_i^a)^2}{(\triangle_i^a)^2} + \sum_{i \in \mathcal{D}_2^a \cup \mathcal{M}_2^a} \frac{(\sigma_{ij_h}^a)^2}{(\triangle_{ij_h}^a)^2} \right). \tag{107}$$

*For the problem instance $\mathcal{P}$ with $\triangle_i^a \to 0, \forall a \in [M], i \in \mathcal{D}_1^a \cup \mathcal{M}_1^a$, and $\triangle_{ij}^a \to 0, \forall a \in [M], i \in [K], j \in [S]$, we have $2\mathcal{H} \leq \mathcal{H}^*(\mathcal{P}) \leq 4\mathcal{H}$.*

*Proof.* Combining the results of Theorem 1 and 2, we have

$$\mathcal{H}^*(\mathcal{P})^{-1} = \max_{\boldsymbol{\omega} \in \Omega} \min_{a \in [M]} \min \left( V_1^a(\boldsymbol{\omega}), V_2^a(\boldsymbol{\omega}), V_3^a(\boldsymbol{\omega}) \right), \tag{108}$$

with

$$V_1^a(\boldsymbol{\omega}) = \min_{j \in [S]} \omega_{i^*(a)j}^a d(F_{i^*(a)j}^a, \phi),$$

$$V_2^a(\boldsymbol{\omega}) = \min_{i \in \mathcal{D}_1^a \cup \mathcal{M}_1^a} \inf_{\tilde{\mu}_i^a > \tilde{\mu}_{i^*(a)}^a} \omega_{i(S+1)}^a d(\mu_i^a, \tilde{\mu}_i^a) + \omega_{i^*(a)(S+1)}^a d(\mu_{i^*(a)}^a, \tilde{\mu}_{i^*(a)}^a), \tag{109}$$

$$V_3^a(\boldsymbol{\omega}) = \min_{i \in \mathcal{D}_2^a \cup \mathcal{M}_2^a} \omega_{ij_h}^a d(F_{ij_h}^a, \phi).$$

According to Lemma 2, we have as $\triangle_{ij}^a \to 0, \forall a \in [M], i \in [K], j \in [S]$,

$$d(F_{i^*(a)j}^a, \phi) \to \frac{(F_{i^*(a)j}^a - \phi)^2}{2(\sigma_{i^*(a)j}^a)^2}, \ d(F_{ij_h}^a, \phi) \to \frac{(F_{ij_h}^a - \phi)^2}{2(\sigma_{ij_h}^a)^2}. \tag{110}$$

According to Lemma 1, we have as $\triangle_i^a \to 0, \forall a \in [M], i \in [K]$,

$$\inf_{\tilde{\mu}_i^a > \tilde{\mu}_{i^*(a)}^a} \omega_{i(S+1)}^a d(\mu_i^a, \tilde{\mu}_i^a) + \omega_{i^*(a)(S+1)}^a d(\mu_{i^*(a)}^a, \tilde{\mu}_{i^*(a)}^a) \to \frac{(\mu_i^a - \mu_{i^*(a)}^a)^2}{2\left( \frac{(\sigma_i^a)^2}{\omega_{i(S+1)}^a} + \frac{(\sigma_{i^*(a)}^a)^2}{\omega_{i^*(a)(S+1)}^a} \right)}. \tag{111}$$

Define

$$\tilde{V}_1^a(\boldsymbol{\omega}) = \min_{j \in [S]} \omega_{i^*(a)j}^a \frac{(F_{i^*(a)j}^a - \phi)^2}{2(\sigma_{i^*(a)j}^a)^2},$$

$$\tilde{V}_2^a(\boldsymbol{\omega}) = \min_{i \in \mathcal{D}_1^a \cup \mathcal{M}_1^a} \frac{(\mu_i^a - \mu_{i^*(a)}^a)^2}{2\left( \frac{(\sigma_i^a)^2}{\omega_{i(S+1)}^a} + \frac{(\sigma_{i^*(a)}^a)^2}{\omega_{i^*(a)(S+1)}^a} \right)},$$

$$\tilde{V}_3^a(\boldsymbol{\omega}) = \min_{i \in \mathcal{D}_2^a \cup \mathcal{M}_2^a} \frac{(F_{ij_h}^a - \phi)^2}{2(\sigma_{ij_h}^a)^2}, \tag{112}$$

$$\tilde{\mathcal{H}}(\mathcal{P})^{-1} = \max_{\boldsymbol{\omega} \in \Omega} \tilde{\mathcal{H}}(\mathcal{P}, \boldsymbol{\omega})^{-1} = \max_{\boldsymbol{\omega} \in \Omega} \min_{a \in [M]} \min \left( \tilde{V}_1^a(\boldsymbol{\omega}), \tilde{V}_2^a(\boldsymbol{\omega}), \tilde{V}_3^a(\boldsymbol{\omega}) \right),$$

$$\mathcal{H}^*(\mathcal{P})^{-1} = \max_{\boldsymbol{\omega} \in \Omega} \mathcal{H}(\mathcal{P}, \boldsymbol{\omega})^{-1} = \max_{\boldsymbol{\omega} \in \Omega} \min_{a \in [M]} \min \left( V_1^a(\boldsymbol{\omega}), V_2^a(\boldsymbol{\omega}), V_3^a(\boldsymbol{\omega}) \right).$$

Then, it is straightforward to check that as $\triangle_i^a \to 0, \forall a \in [M], i \in \mathcal{D}_1^a \cup \mathcal{M}_1^a$, and $\triangle_{ij}^a \to 0, \forall a \in [M], i \in [K], j \in [S]$,

$$\mathcal{H}(\mathcal{P}, \boldsymbol{\omega})^{-1} \to \tilde{\mathcal{H}}(\mathcal{P}, \boldsymbol{\omega}). \tag{113}$$

Let

$$\boldsymbol{\omega}^* = \arg\max_{\boldsymbol{\omega} \in \Omega} \min_{a \in [M]} \min \left( V_1^a(\boldsymbol{\omega}), V_2^a(\boldsymbol{\omega}), V_3^a(\boldsymbol{\omega}) \right),$$

$$\tilde{\boldsymbol{\omega}} = \arg\max_{\boldsymbol{\omega} \in \Omega} \min_{a \in [M]} \min \left( \tilde{V}_1^a(\boldsymbol{\omega}), \tilde{V}_2^a(\boldsymbol{\omega}), \tilde{V}_3^a(\boldsymbol{\omega}) \right), \tag{114}$$

then we have $\mathcal{H}^*(\mathcal{P})^{-1} = \mathcal{H}(\mathcal{P}, \boldsymbol{\omega}^*)^{-1}$ and $\tilde{\mathcal{H}}(\mathcal{P})^{-1} = \tilde{\mathcal{H}}(\mathcal{P}, \tilde{\boldsymbol{\omega}})^{-1}$.

Furthermore, as $\triangle_i^a \to 0, \forall a \in [M], i \in \mathcal{D}_1^a \cup \mathcal{M}_1^a$, and $\triangle_{ij}^a \to 0, \forall a \in [M], i \in [K], j \in [S]$, we have

$$
\begin{aligned}
|\tilde{\mathcal{H}}(\mathcal{P})^{-1} - \mathcal{H}^*(\mathcal{P})^{-1}| &= |\tilde{\mathcal{H}}(\mathcal{P}, \tilde{\boldsymbol{\omega}})^{-1} - \mathcal{H}(\mathcal{P}, \boldsymbol{\omega}^*)^{-1}| \\
&\leq |\tilde{\mathcal{H}}(\mathcal{P}, \tilde{\boldsymbol{\omega}})^{-1} - \mathcal{H}(\mathcal{P}, \tilde{\boldsymbol{\omega}})^{-1}| + |\mathcal{H}(\mathcal{P}, \tilde{\boldsymbol{\omega}})^{-1} - \mathcal{H}(\mathcal{P}, \boldsymbol{\omega}^*)^{-1}| \quad (115) \\
&\to 0.
\end{aligned}
$$

where the third line can be verified using (113).

Next, we focus on analyzing the $\tilde{\mathcal{H}}(\mathcal{P})^{-1}$. We first provide a lower bound on $\tilde{\mathcal{H}}(\mathcal{P})^{-1}$. Consider a feasible sampling ratio:

$$
\begin{aligned}
\omega_{i^*(a)j}^a &= \frac{(\sigma_{i^*(a)j}^a)^2}{(\triangle_{i^*(a)j}^a)^2 \mathcal{H}}, \quad \forall a \in [M] \\
\omega_i^a &= \frac{(\sigma_i^a)^2}{(\triangle_i^a)^2 \mathcal{H}}, \quad \forall a \in [M], i \in \mathcal{D}_1^a \cup \mathcal{M}_1^a \cup \{i^*(a)\} \quad (116) \\
\omega_{ij_h}^a &= \frac{(\sigma_{ij_h}^a)^2}{(\triangle_{ij_h}^a)^2 \mathcal{H}}, \quad \forall a \in [M], i \in \mathcal{D}_2^a \cup \mathcal{M}_2^a
\end{aligned}
$$

and $\omega_{ij}^a = 0$, otherwise. Then, we can obtain that

$$
\tilde{\mathcal{H}}(\mathcal{P})^{-1} = \max_{\boldsymbol{\omega} \in \Omega} \tilde{\mathcal{H}}(\mathcal{P}, \boldsymbol{\omega})^{-1} \geq \frac{1}{2} \min(\frac{1}{\mathcal{H}}, \frac{1}{2\mathcal{H}}, \frac{1}{\mathcal{H}}) = \frac{1}{4\mathcal{H}}. \quad (117)
$$

Let $l^* = \arg\min_{l \in \mathcal{D}_1^a \cup \mathcal{M}_1^a}(\mu_{i^*(a)}^a - \mu_l^a)$, then $\triangle_{i^*(a)}^a = \mu_{i^*(a)}^a - \mu_{l^*}^a$. We then prove an upper bound on $\tilde{\mathcal{H}}(\mathcal{P})^{-1}$.

$$
\begin{aligned}
\tilde{\mathcal{H}}(\mathcal{P})^{-1} &= \max_{\boldsymbol{\omega} \in \Omega} \tilde{\mathcal{H}}(\mathcal{P}, \boldsymbol{\omega})^{-1} \\
&= \max_{\boldsymbol{\omega} \in \Omega} \min_{a \in [M]} \min\left(\tilde{V}_1^a(\boldsymbol{\omega}), \tilde{V}_2^a(\boldsymbol{\omega}), \tilde{V}_3^a(\boldsymbol{\omega})\right) \\
&= \frac{1}{2} \max_{\boldsymbol{\omega} \in \Omega} \min_{a \in [M]} \min\left(\min_{j \in [S]} \omega_{i^*(a)j}^a \frac{\triangle_{i^*(a)j}^a}{(\sigma_{i^*(a)j}^a)^2},\right. \\
&\quad \min\left(\min_{i \in \mathcal{D}_1^a \cup \mathcal{M}_1^a} \frac{(\triangle_i^a)^2}{\frac{(\sigma_i^a)^2}{\omega_{i(S+1)}^a} + \frac{(\sigma_{i^*(a)}^a)^2}{\omega_{i^*(a)(S+1)}^a}}, \frac{(\triangle_{i^*(a)}^a)^2}{\frac{(\sigma_i^a)^2}{\omega_{i(S+1)}^a} + \frac{(\sigma_{i^*(a)}^a)^2}{\omega_{i^*(a)(S+1)}^a}}\right), \\
&\quad \left.\min_{i \in \mathcal{D}_2^a \cup \mathcal{M}_2^a} \omega_{ij_h}^a \frac{(\triangle_{ij_h}^a)^2}{(\sigma_{ij_h}^a)^2}\right) \\
&\leq \frac{1}{2} \max_{\boldsymbol{\omega} \in \Omega} \min_{a \in [M]} \min\left(\min_{j \in [S]} \omega_{i^*(a)j}^a \frac{\triangle_{i^*(a)j}^a}{(\sigma_{i^*(a)j}^a)^2},\right. \\
&\quad \min\left(\min_{i \in \mathcal{D}_1^a \cup \mathcal{M}_1^a} \omega_{i(S+1)}^a \frac{(\triangle_i^a)^2}{(\sigma_i^a)^2}, \omega_{i^*(a)(S+1)}^a \frac{(\triangle_{i^*(a)}^a)^2}{(\sigma_{i^*(a)}^a)^2}\right), \min_{i \in \mathcal{D}_2^a \cup \mathcal{M}_2^a} \omega_{ij_h}^a \frac{(\triangle_{ij_h}^a)^2}{(\sigma_{ij_h}^a)^2}\right) \\
&= \frac{1}{2} \max_{\boldsymbol{\omega} \in \Omega} \min_{a \in [M]} \min\left(\min_{j \in [S]} \omega_{i^*(a)j}^a \frac{\triangle_{i^*(a)j}^a}{(\sigma_{i^*(a)j}^a)^2},\right. \\
&\quad \left.\min_{i \in \mathcal{D}_1^a \cup \mathcal{M}_1^a \cup \{i^*(a)\}} \omega_{i(S+1)}^a \frac{(\triangle_i^a)^2}{(\sigma_i^a)^2}, \min_{i \in \mathcal{D}_2^a \cup \mathcal{M}_2^a} \omega_{ij_h}^a\right) \\
&= \frac{1}{2\mathcal{H}},
\end{aligned}
$$

$$(118)$$

where the last equation is obtained by solving the corresponding optimization problem.

Therefore, we conclude that $2\mathcal{H} \leq \tilde{\mathcal{H}}(\mathcal{P}) \leq 4\mathcal{H}$, which also implies $2\mathcal{H} \leq \mathcal{H}^*(\mathcal{P}) \leq 4\mathcal{H}$ as $\triangle_i^a \to 0, \forall a \in [M], i \in \mathcal{D}_1^a \cup \mathcal{M}_1^a$, and $\triangle_{ij}^a \to 0, \forall a \in [M], i \in [K], j \in [S]$. □

### B.7 EXTENSION TO OTHER FORMULATIONS

*In this section, we extend the results of Theorem 1 to a new formulation, in which the agent observes a vector of all metrics after pulling an arm. Such performance setting was also considered in the literature (Hunter & Pasupathy, 2013; Camilleri et al., 2022). However, we address a complex multi-task BAI problem with risk constraint under the fixed-confidence setting. Let $\omega_i^a$ denote the sampling ratio of arm $i$ under task $a$ in this formulation. Theorem 5 provides the lower bound on the sample complexity $\mathbb{E}[\tau]$ and Lemma 7 outlines the optimality conditions satisfied by the static sampling rule $\boldsymbol{\omega}$.*

**Theorem 5.** *Given a fixed confidence level $\delta \in (0,1)$. For any problem instance $\mathcal{P} \in \mathcal{S}$ and any strategy satisfying $\mathbb{P}\left(\forall a \in [M], i^*(a) = \hat{i}_\tau(a)\right) \geq 1 - \delta$,*

$$\mathbb{E}[\tau] \geq \mathcal{H}^*(\mathcal{P})kl(\delta, 1 - \delta), \tag{119}$$

*as $\delta \to 0$, we have*

$$\liminf_{\delta \to 0} \frac{\mathbb{E}[\tau]}{\log(1/\delta)} \geq \mathcal{H}^*(\mathcal{P}), \tag{120}$$

*where $\mathcal{H}^*(\mathcal{P})^{-1} = \max_{\boldsymbol{\omega} \in \Omega} \min_{a \in [M]} \min\left(V_1^a(\boldsymbol{\omega}), V_2^a(\boldsymbol{\omega}), V_3^a(\boldsymbol{\omega}), V_4^a(\boldsymbol{\omega})\right)$, with*

$$V_1^a(\boldsymbol{\omega}) = \min_{j \in [S]} \omega_{i^*(a)}^a d(F_{i^*(a)j}^a, \phi),$$

$$V_2^a(\boldsymbol{\omega}) = \min_{i \in \mathcal{D}_1^a} \omega_i^a d(\mu_i^a, \mu_{i,i^*(a)}^a) + \omega_{i^*(a)}^a d(\mu_{i^*(a)}^a, \mu_{i,i^*(a)}^a),$$

$$V_3^a(\boldsymbol{\omega}) = \min_{i \in \mathcal{D}_2^a} \omega_i^a \sum_{j \in \mathcal{B}_2^a(i)} d(F_{ij}^a, \phi), \tag{121}$$

$$V_4^a(\boldsymbol{\omega}) = \min_{i \in \mathcal{D}_3^a} \inf_{\tilde{\mu}_i^a > \tilde{\mu}_{i^*(a)}^a} \omega_i^a d(\mu_i^a, \tilde{\mu}_i^a) + \omega_{i^*(a)}^a d(\mu_{i^*(a)}^a, \tilde{\mu}_{i^*(a)}^a) + \omega_i^a \sum_{j \in \mathcal{B}_2^a(i)} d(F_{ij}^a, \phi).$$

**Lemma 7.** *Define $\mathcal{A}_1 = \{a \in [M] : V_1^a(\boldsymbol{\omega}) \geq z^*\}$, and $\mathcal{A}_2 = \{a \in [M] : V_1^a(\boldsymbol{\omega}) = z^*\}$. The static optimal sampling ratio $\boldsymbol{\omega}^*$ satisfies:*

$$V_1^a(\boldsymbol{\omega}^*) \geq z^*, \forall a \in \mathcal{A}_1$$

$$V_1^{\tilde{a}}(\boldsymbol{\omega}^*) = V_2^a(\boldsymbol{\omega}^*) = V_3^a(\boldsymbol{\omega}^*) = V_4^a(\boldsymbol{\omega}^*) = z^*, \forall a \in [M], \tilde{a} \in \mathcal{A}_2$$

$$(\boldsymbol{\omega}_i^a)^* d(\mu_i^a, \mu_{i,i^*(a)}^a) + (\boldsymbol{\omega}_{i^*(a)}^a)^* d(\mu_{i^*(a)}^a, \mu_{i,i^*(a)}^a) = z^*, \forall a \in [M], i \in \mathcal{D}_1^a$$

$$(\boldsymbol{\omega}_i^a)^* \sum_{j \in \mathcal{B}_2^a(i)} d(F_{ij}^a, \phi) = z^*, \forall a \in [M], i \in \mathcal{D}_2^a$$

$$(\boldsymbol{\omega}_i^a)^*(d(\mu_i^a, \mu_{i,i^*(a)}^a) + \sum_{j \in \mathcal{B}_2^a(i)} d(F_{ij}^a, \phi)) + (\boldsymbol{\omega}_{i^*(a)}^a)^* d(\mu_{i^*(a)}^a, \mu_{i,i^*(a)}^a) = z^*, \forall a \in [M], i \in \mathcal{D}_3^a$$

$$\sum_{i \in \mathcal{D}_1^a} \frac{d(\mu_{i^*(a)}^a, \mu_{i,i^*(a)}^a)}{d(\mu_i^a, \mu_{i,i^*(a)}^a)} + \sum_{i \in \mathcal{D}_3^a} \frac{d(\mu_{i^*(a)}^a, \mu_{i,i^*(a)}^a)}{d(\mu_i^a, \mu_{i,i^*(a)}^a) + \sum_{j \in \mathcal{B}_2^a(i)} d(F_{ij}^a, \phi)} = 1, \forall a \in \mathcal{A}_1$$

$$\sum_{a \in [M]} \sum_{i \in [K]} (\boldsymbol{\omega}_i^a)^* = 1,$$

$$(\boldsymbol{\omega}_i^a)^* \geq 0, \forall a \in [M], i \in [K]. \tag{122}$$

## C ASYMPTOTICALLY OPTIMAL STRATEGY

### C.1 PROOF OF LEMMA 4

**Lemma 4.** *Given a fixed confidence level $\delta \in (0,1)$ and $\alpha > 1$. There exists a constant $C(\alpha, M, K, S)$ such that for any sampling rule $\{\pi^t\}_t$, using the stopping rule with $\beta(t,\delta) = \log\left(\frac{Ct^\alpha}{\delta}\right)$, then for all problem instance $\mathcal{P} \in \mathcal{S}$, $\mathbb{P}\left(\exists a \in [M], \hat{i}_\tau(a) \neq i^*(a)\right) \leq \delta$.*

*Proof.* By definition of the stopping rule $\tau = \inf\{t \in \mathbb{N} : \inf_{\tilde{\mathcal{P}} \in \mathcal{A}(\hat{\mathcal{P}}(t))} f(\hat{\mathcal{P}}(t), \tilde{\mathcal{P}}) > \beta(t,\delta)\}$, we have

$$
\begin{aligned}
&\mathbb{P}\left(\exists a \in [M], \hat{i}_\tau(a) \neq i^*(a)\right) \\
&\leq \mathbb{P}\left(\exists t \in \mathbb{N}, \exists a \in [M], \exists i \in [K] \setminus \{i^*(a, \mathcal{P})\}, i^*(a, \hat{\mathcal{P}}(t)) = i, \inf_{\tilde{\mathcal{P}} \in \mathcal{A}(\hat{\mathcal{P}}(t))} f(\hat{\mathcal{P}}(t), \tilde{\mathcal{P}}) > \beta(t,\delta)\right) \\
&\leq \sum_{t=1}^\infty \mathbb{P}\left(\inf_{\tilde{\mathcal{P}} \in \mathcal{A}(\hat{\mathcal{P}}(t))} \sum_{a \in [M], i \in [K]} \left(N_{i(S+1)}^a(t) d(\hat{\mu}_i^a(t), \tilde{\mu}_i^a(t)) + \sum_{j \in [S]} N_{ij}^a(t) d(\hat{F}_{ij}^a(b,t), \tilde{F}_{ij}^a)\right) > \beta(t,\delta)\right) \\
&\leq \sum_{t=1}^\infty \mathbb{P}\left(\sum_{a \in [M], i \in [K]} \left(N_{i(S+1)}^a(t) d(\hat{\mu}_i^a(t), \mu_i^a(t)) + \sum_{j \in [S]} N_{ij}^a(t) d(\hat{F}_{ij}^a(b,t), F_{ij}^a)\right) > \beta(t,\delta)\right) \\
&\leq \sum_{t=1}^\infty e^{1+MK(S+1)} \left(\frac{\beta(t,\delta)^2 \log t}{MK(S+1)}\right)^{MK(S+1)} e^{-\beta(t,\delta)},
\end{aligned}
\tag{123}
$$

where the last line is obtained by extending Theorem 2 of Magureanu et al. (2014).

Since we have $\beta(t,\delta) = \log\left(\frac{Ct^\alpha}{\delta}\right)$, it can be verified that if we choose the constant $C$ satisfy

$$
\sum_{t=1}^\infty \frac{e^{MK(S+1)}}{MK(S+1)^{MK(S+1)}} \frac{([\log(Ct^\alpha) + \log(1/\delta)]^2 \log t)^{MK(S+1)}}{t^\alpha} \leq C, \tag{124}
$$

then we can obtain $\mathbb{P}\left(\exists a \in [M], \hat{i}_\tau(a) \neq i^*(a)\right) \leq \delta$, which concludes the proof. $\square$

### C.2 PROOF OF THEOREM 3

*In this subsection, we prove the sample complexity results in Theorem 3. The proof of Theorem 3 is more or less identical to that of Garivier & Kaufmann (2016), we include it here for completeness.*

**Lemma 8.** *(Lemma 17 in Garivier & Kaufmann (2016)) Using the tracking rule in (13), we have $N_{ij}^a(t) \geq (\sqrt{t} - MK(S+1))^+ - 1$, For any $\epsilon > 0$ and $t_0 > 0$ such that $\sup_{t \geq t_0} \max_{(a,i,j) \in [M] \times [K] \times [S+1]} |(\omega_{ij}^a)^*(\hat{\mathcal{P}}(t)) - (\omega_{ij}^a)^*(\mathcal{P})| \leq \epsilon$, then*

$$
\sup_{t \geq t_\epsilon} \max_{(a,i,j) \in [M] \times [K] \times [S+1]} \left|\frac{N_{ij}^a(t)}{t} - (\omega_{ij}^a)^*(\mathcal{P})\right| \leq 3(MK(S+1) - 1)\epsilon. \tag{125}
$$

*Let $\mathcal{B}_\infty^r(c)$ denote a ball of radius $r$ centered at $c$ under the infinity norm. For problem instance $\mathcal{P} = (\boldsymbol{\mu}^a, \boldsymbol{F}^a)_{a \in [M]}$, since $\boldsymbol{\omega}^*(\mathcal{P})$ is a continuous function of $\mathcal{P}$, we have $\forall \epsilon > 0, \exists \xi(\epsilon) > 0$, such that $\forall \tilde{\mathcal{P}} \in \mathcal{B}_\infty^{\xi(\epsilon)}(\mathcal{P})$, we have $\boldsymbol{\omega}^*(\tilde{\mathcal{P}}) \in \mathcal{B}_\infty^\epsilon(\boldsymbol{\omega}^*(\mathcal{P}))$.*

**Lemma 9.** *For problem instance $\mathcal{P} \in \mathcal{S}$, define $\mathcal{E}_T(\epsilon) = \bigcap_{t \in T^{1/4}}^T \{\hat{\mathcal{P}}(t) \in \mathcal{B}_\infty^{\xi(\epsilon)}(\mathcal{P})\}$, we have*

$$
\mathbb{P}(\mathcal{E}_T^c(\epsilon)) \leq B(\epsilon, \mathcal{P}) \exp\left(-C(\epsilon, \mathcal{P}) T^{1/8}\right), \tag{126}
$$

*where*

$$B(\epsilon, \mathcal{P}) = \sum_{a \in [M], i \in [K]} \left( \frac{\exp(MK(S+1)d(\mu_i^a - \xi(\epsilon), \mu_i^a))}{1 - \exp(d(\mu_i^a - \xi(\epsilon), \mu_i^a))} + \frac{\exp(MK(S+1)d(\mu_i^a + \xi(\epsilon), \mu_i^a))}{1 - \exp(d(\mu_i^a + \xi(\epsilon), \mu_i^a))} \right.$$

$$\left. + \sum_{j \in [S]} \left( \frac{\exp(MK(S+1)d(F_{ij}^a - \xi(\epsilon), F_{ij}^a))}{1 - \exp(d(F_{ij}^a - \xi(\epsilon), F_{ij}^a))} + \frac{\exp(MK(S+1)d(F_{ij}^a + \xi(\epsilon), F_{ij}^a))}{1 - \exp(d(F_{ij}^a + \xi(\epsilon), F_{ij}^a))} \right) \right),$$

$$C(\epsilon, \mathcal{P}) = \min_{a \in [M]} \left( \min_{i \in [K]} \left( d(\mu_i^a - \xi(\epsilon), \mu_i^a) \bigwedge d(\mu_i^a + \xi(\epsilon), \mu_i^a) \right) \bigwedge \right.$$

$$\left. \min_{i \in [K], j \in [S]} \left( d(F_{ij}^a - \xi(\epsilon), F_{ij}^a) \bigwedge d(F_{ij}^a + \xi(\epsilon), F_{ij}^a) \right) \right).$$

$$(127)$$

*Proof.* By definition of $\mathcal{E}_T(\epsilon)$, we have that

$$\mathbb{P}\left(\mathcal{E}_T^c(\epsilon)\right) = \mathbb{P}\left( \bigcup_{t \in T^{1/4}}^{T} \{\hat{\mathcal{P}}(t) \notin \mathcal{B}_\infty^{\xi(\epsilon)}(\mathcal{P})\} \right)$$

$$= \mathbb{P}\left( \bigcup_{t \in T^{1/4}}^{T} \left( \bigcup_{a \in [M], i \in [K]} \left( \hat{\mu}_i^a(t) \notin \mathcal{B}_\infty^{\xi(\epsilon)}(\mu_i^a) \bigcup \bigcup_{j \in [S]} \left( \hat{F}_{ij}^a(b, t) \notin \mathcal{B}_\infty^{\xi(\epsilon)}(F_{ij}^a) \right) \right) \right) \right)$$

$$\leq \sum_{t = T^{1/4}}^{T} \sum_{a \in [M], i \in [K]} \left( \mathbb{P}\left( \hat{\mu}_i^a(t) \notin \mathcal{B}_\infty^{\xi(\epsilon)}(\mu_i^a) \right) + \sum_{j \in [S]} \mathbb{P}\left( \hat{F}_{ij}^a(b, t) \notin \mathcal{B}_\infty^{\xi(\epsilon)}(F_{ij}^a) \right) \right).$$

$$(128)$$

$\square$

*Let $T$ is large enough to satisfy that $N_{ij}^a(t) \geq \sqrt{t} - MK(S+1) \geq 0$, then we have*

$$\mathbb{P}\left( \hat{F}_{ij}^a(b, t) \notin \mathcal{B}_\infty^{\xi(\epsilon)}(F_{ij}^a) \right)$$

$$= \mathbb{P}\left( \left( \hat{F}_{ij}^a(b, t) \leq F_{ij}^a - \xi(\epsilon) \right) \bigcup \left( \hat{F}_{ij}^a(b, t) \geq F_{ij}^a + \xi(\epsilon) \right) \right) \qquad (129)$$

$$= \mathbb{P}\left( \hat{F}_{ij}^a(b, t) \leq F_{ij}^a - \xi(\epsilon) \right) + \mathbb{P}\left( \hat{F}_{ij}^a(b, t) \leq F_{ij}^a + \xi(\epsilon) \right),$$

*Denote $\hat{F}_{ij,l}^a(b)$ as the estimate of $F_{ij}^a$ based on first $l$ observations from $Y_{ij}^a$, then we have*

$$\mathbb{P}\left( \hat{F}_{ij}^a(b, t) \leq F_{ij}^a - \xi(\epsilon) \right)$$

$$= \mathbb{P}\left( \hat{F}_{ij}^a(b, t) \leq F_{ij}^a - \xi(\epsilon), N_{ij}^a(t) \geq \sqrt{t} - MK(S+1) \right)$$

$$\leq \sum_{l = \sqrt{t} - MK(S+1)}^{t} \mathbb{P}\left( \hat{F}_{ij,l}^a(b) \leq F_{ij}^a - \xi(\epsilon) \right) \qquad (130)$$

$$\leq \sum_{l = \sqrt{t} - MK(S+1)}^{t} \exp\left( -ld(F_{ij}^a - \xi(\epsilon), F_{ij}^a) \right)$$

$$\leq \frac{\exp\left( -(\sqrt{t} - MK(S+1))d(F_{ij}^a - \xi(\epsilon), F_{ij}^a) \right)}{1 - \exp\left( -d(F_{ij}^a - \xi(\epsilon), F_{ij}^a) \right)},$$

*where the second equation is from the Chernoff inequality. Using similar argument, we can obtain that*

$$\mathbb{P}\left(\hat{F}_{ij}^a(t) \le F_{ij}^a + \xi(\epsilon)\right) \le \frac{\exp\left(-(\sqrt{t} - MK(S+1))d(F_{ij}^a + \xi(\epsilon), F_{ij}^a)\right)}{1 - \exp\left(-d(F_{ij}^a + \xi(\epsilon), F_{ij}^a)\right)},$$

$$\mathbb{P}\left(\hat{\mu}_i^a(t) \le \mu_i^a - \xi(\epsilon)\right) \le \frac{\exp\left(-(\sqrt{t} - MK(S+1))d(\mu_i^a - \xi(\epsilon), \mu_i^a)\right)}{1 - \exp\left(-d(\mu_i^a - \xi(\epsilon), \mu_i^a)\right)}, \tag{131}$$

$$\mathbb{P}\left(\hat{\mu}_i^a(t) \le \mu_i^a + \xi(\epsilon)\right) \le \frac{\exp\left(-(\sqrt{t} - MK(S+1))d(\mu_i^a + \xi(\epsilon), \mu_i^a)\right)}{1 - \exp\left(-d(\mu_i^a + \xi(\epsilon), \mu_i^a)\right)}.$$

*Therefore, we conclude that*

$$\mathbb{P}\left(\mathcal{E}_T^c(\epsilon)\right) \le \sum_{t=T^{1/4}}^{T} B(\epsilon, \mathcal{P}) \exp(-\sqrt{t}C(\epsilon, \mathcal{P})) \le B(\epsilon, \mathcal{P}) \exp\left(-C(\epsilon, \mathcal{P})T^{1/8}\right). \tag{132}$$

**Theorem 3.** *For problem instance $\mathcal{P} \in \mathcal{S}$, the Algorithm 1 satisfies that*

$$\limsup_{\delta \to 0} \frac{\mathbb{E}[\tau]}{\log(1/\delta)} \le \mathcal{H}^*(\mathcal{P}). \tag{133}$$

*Proof.* Let $\mathcal{Q} = (\boldsymbol{\nu}^a, \boldsymbol{\Phi}^a)_{a \in [M]}$ denote an problem instance. Define

$$g(\tilde{\mathcal{P}}, \tilde{\boldsymbol{\omega}}) = \inf_{\mathcal{Q} \in \mathcal{A}(\tilde{P})} \sum_{a \in [M], i \in [K]} \left( \tilde{\omega}_{i(S+1)d(\tilde{\mu}_i^a)}^a d(\tilde{\mu}_i^a, \nu_i^a) + \sum_{j \in [S]} \tilde{\omega}_{ij}^a d(\tilde{F}_{ij}^a, \Phi_{ij}^a(b)) \right).$$

$$\mathcal{Z}_\epsilon^*(\mathcal{P}) = \inf_{\substack{\tilde{\mathcal{P}} \in \mathcal{B}_\infty^{\xi(\epsilon)}(\mathcal{P}) \\ \tilde{\boldsymbol{\omega}} \in \mathcal{B}_\infty^{3(MK(S+1)-1)\epsilon}(\boldsymbol{\omega}^*(\mathcal{P}))}} g(\tilde{\mathcal{P}}, \tilde{\boldsymbol{\omega}}). \tag{134}$$

According to Lemma 8, $\forall \epsilon > 0, \exists T_\epsilon > 0$, such that $\forall T \ge T_\epsilon$, on event $\mathcal{E}_T(\epsilon)$, we have

$$\sup_{t \ge \sqrt{T}} \max_{(a,i,j) \in [M] \times [K] \times [S+1]} \left| \frac{N_{ij}^a(t)}{t} - (\omega_{ij}^a)^*(\mathcal{P}) \right| \le 3(MK(S+1)-1)\epsilon. \tag{135}$$

Note that, the stopping rule is

$$\begin{aligned}
\tau &= \inf\{t \in \mathbb{N} : \inf_{\tilde{\mathcal{P}} \in \mathcal{A}(\hat{\mathcal{P}}(t))} f(\hat{\mathcal{P}}(t), \tilde{\mathcal{P}}) > \beta(t, \delta)\} \\
&= \inf\{t \in \mathbb{N} : \inf_{\tilde{\mathcal{P}} \in \mathcal{A}(\hat{\mathcal{P}}(t))} tg(\hat{\mathcal{P}}(t), \boldsymbol{\omega}^*(\hat{\mathcal{P}}(t))) > \beta(t, \delta)\} \\
&= \inf\{t \in \mathbb{N} : g(\hat{\mathcal{P}}(t), \boldsymbol{\omega}^*(\hat{\mathcal{P}}(t))) > \frac{\beta(t, \delta)}{t}\}.
\end{aligned} \tag{136}$$

Therefore, $\forall T > T_\epsilon$ and on $\mathcal{E}_T(\epsilon)$, we have

$$\begin{aligned}
\min(\tau, T) &\le \sqrt{T} + \sum_{t=\sqrt{T}}^{T} \mathbb{I}\left(\tau > t\right) \le \sqrt{T} + \sum_{t=\sqrt{T}}^{T} \mathbb{I}\left(g(\hat{\mathcal{P}}(t), \boldsymbol{\omega}^*(\hat{\mathcal{P}}(t))) \le \frac{\beta(t, \delta)}{t}\right) \\
&\le \sqrt{T} + \sum_{t=\sqrt{T}}^{T} \mathbb{I}\left(\mathcal{Z}_\epsilon^*(\mathcal{P}) \le \frac{\beta(t, \delta)}{t}\right) \le \sqrt{T} + \frac{\beta(t, \delta)}{t}.
\end{aligned} \tag{137}$$

Following Garivier & Kaufmann (2016), we have

$$T_0(\delta) = \inf\{T \in \mathbb{N} : \sqrt{T} + \frac{\beta(t, \delta)}{t} < T\} = \frac{1}{\mathcal{Z}_\epsilon^*(\mathcal{P})} \left(\mathcal{O}(\log(1/\delta)) + \mathcal{O}(\log\log(1/\delta))\right),$$

which means $\forall T > \max(T_0(\delta), T_\epsilon)$, on $\mathcal{E}_T(\epsilon)$, we have $\tau < T$.

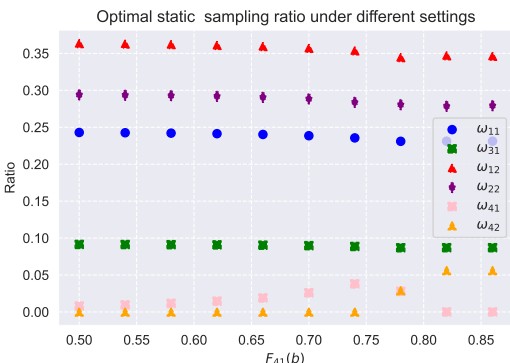

Figure 3: Optimal static sampling ratio $\boldsymbol{\omega}^*$ under different $F_{41}(b)$

Finally, we conclude that

$$
\mathbb{E}[\tau] = \sum_{T=1}^{\infty} \mathbb{P}(\tau \geq t)
$$

$$
= \sum_{T=1}^{\max(T_0(\delta), T_\epsilon)} \mathbb{P}(\tau \geq t) + \sum_{T=\max(T_0(\delta), T_\epsilon)}^{\infty} \mathbb{P}(\mathcal{E}_T(\epsilon))\mathbb{P}(\tau \geq t | \mathcal{E}_T(\epsilon)) + \mathbb{P}(\mathcal{E}_T^c(\epsilon))\mathbb{P}(\tau \geq t | \mathcal{E}_T^c(\epsilon))
$$

$$
\leq T_0(\delta) + T_\epsilon + \sum_{T=1}^{\infty} \mathbb{P}(\mathcal{E}_T^c(\epsilon))
$$

$$
\leq \frac{1}{\mathcal{Z}_\epsilon^*(\mathcal{P})} \left( \mathcal{O}(\log(1/\delta)) + \mathcal{O}(\log\log(1/\delta)) \right) + T_\epsilon + \sum_{T=1}^{\infty} B(\epsilon, \mathcal{P}) \exp\left( -C(\epsilon, \mathcal{P}) T^{1/8} \right).
$$

$$(138)$$

Furthermore, we have

$$
\limsup_{\delta \to 0} \frac{\mathbb{E}[\tau]}{\log(1/\delta)} \leq \frac{1}{\mathcal{Z}_\epsilon^*(\mathcal{P})}. \tag{139}
$$

Let $\epsilon \to 0$, we have

$$
\limsup_{\delta \to 0} \frac{\mathbb{E}[\tau]}{\log(1/\delta)} \leq \mathcal{H}^*(\mathcal{P}). \tag{140}
$$

$\square$

# D NUMERICAL EXPERIMENT

## D.1 ILLUSTRATIVE EXAMPLES

*In this subsection, we provide illustrative examples to demonstrate how the static optimal sampling ratio balances identification difficulty across different tasks and constraints, managing the trade-off between optimality and feasibility.*

**Single task example.** *In this example, there is $1$ task, $4$ arms and $1$ constraint. The $X_i$ are Gaussian random variables with means $\boldsymbol{\mu} = (1.0, 0.8, 1.2, 0.6)$ and standard deviations $\boldsymbol{\sigma} = (1.0, 0.8, 1.0, 1.0)$. The $Y_{ij}$ are Gaussian random variables with $\boldsymbol{F} = (0.95, 0.95, 0.80, 0.82)$. The threshold parameter is $\phi = 0.9$. Arm 1 is optimal, arm 2 is suboptimal, arm 3 is infeasible but has a better objective value, and arm 4 is infeasible with a worse objective value.*

*Figure 3 illustrates the change in the static optimal sampling ratio $\boldsymbol{\omega}^*$ for different values of $F_{41}(b)$. When $F_{41}(b)$ is much less than $\phi$, identifying arm 4 as infeasible is easier than identifying it as suboptimal, therefore, $\omega_{42} = 0$. In contrast, when $F_{41}(b)$ is close to $\phi$, identifying arm 4 as suboptimal becomes easier, then $\omega_{41} = 0$. When both cases are equally difficult, there are multiple solutions, and Figure 3 shows one of them, verifying the results in Theorem 2.*

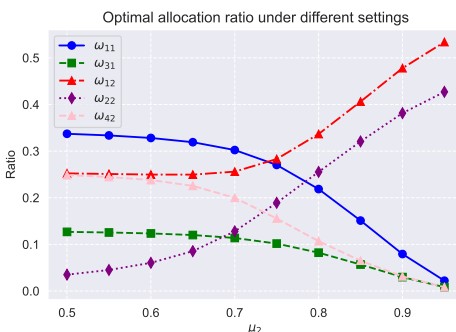

Figure 4: Optimal static sampling ratio $\boldsymbol{\omega}^*$ under different $\mu_2$

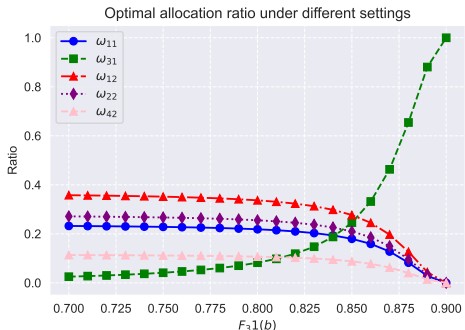

Figure 5: Optimal static sampling ratio $\boldsymbol{\omega}^*$ under different $F_{31}(b)$

Figures 4 and 5 illustrate how the $\boldsymbol{\omega}^*$ changes with $\mu_2$ and $F_{31}(b)$. As $\mu_2$ approaches $\mu_1$, distinguishing between arms 1 and 2 becomes more difficult, and the $\boldsymbol{\omega}^*$ shifts focus toward verifying the optimality of these arms, causing $\omega_{12}$ and $\omega_{22}$ to increase gradually. However, as $F_{31}(b)$ approaches $\phi$, the $\boldsymbol{\omega}^*$ shifts to focus on the feasibility of arm 3, leading to a gradual increase in $\omega_{31}$.

**Multiple tasks example.** This example involves 2 tasks, 3 arms, and 1 constraint. The $X_i^a$ are Gaussian random variables with means $\boldsymbol{\mu}^1 = (1.00, 0.90, 1.10)$, $\boldsymbol{\mu}^2 = (1.00, 0.99, 1.10)$, and standard deviations $\boldsymbol{\sigma}^1 = (1.0, 0.6, 1.0)$, $\boldsymbol{\sigma}^2 = (1.0, 0.6, 1.0)$. The $Y_{ij}^a$ are Gaussian random variables with $\boldsymbol{F}^1 = (0.91, 0.91, 0.89)$ and $\boldsymbol{F}^2 = (0.91, 0.91, 0.89)$. The threshold parameter is $\phi = 0.9$. Arm 1 is optimal, arm 2 is suboptimal, and arm 3 is infeasible with a better objective value.

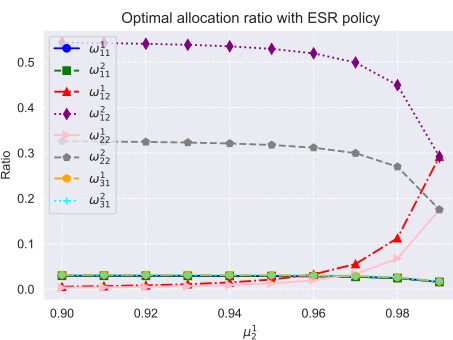

Figure 6: Optimal static sampling ratio $\boldsymbol{\omega}^*$ under different $\mu_2^1$

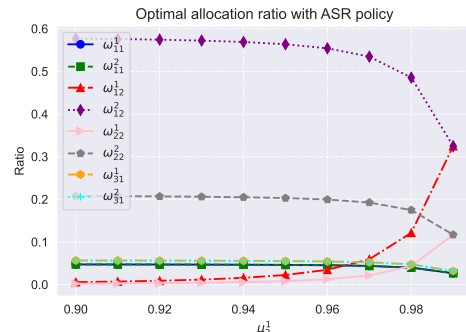

Figure 7: Approximate sampling ratio in Lemma 3 under different $\mu_2^1$

Figures 6 and 7 illustrate how the $\boldsymbol{\omega}^*$ and $\tilde{\boldsymbol{\omega}}$ in Lemma 3 changes with $\mu_2^1$. As $\mu_2^1$ approaches $\mu_1^1$, both sampling rules focus on assessing the optimality of arms 1 and 2 in task 1, causing $\omega_{12}^1$ and $\omega_{22}^1$ to gradually increase. When $\mu_2^1$ reaches 0.99, both tasks become equally difficult, resulting in an equal sampling ratio across the two tasks.

### D.2 EXPERIMENT CONFIGURATIONS

In Experiment 1, all random variables are Gaussian, with their mean and variance parameters summarized in Table 2 and 3. Other parameters are set as $b = 2.04$ and $\phi = 0.9$.

In Experiment 2, the objective-related random variables are Bernoulli, with their mean and variance parameters summarized in Table 4. The constraint-related random variables are Gaussian, with their mean and variance parameters summarized in Table 5. Other parameters are set as $b = 2.0$ and $\phi = 0.9$.

Table 2: Objective-related random variable parameters in experiment 1

|  | TASK1 | | TASK2 | |
| --- | --- | --- | --- | --- |
|  | Mean | Variance | Mean | Variance |
| $X_1$ | 1.000 | 1.000 | 1.000 | 1.000 |
| $X_2$ | 0.800 | 1.000 | 0.800 | 1.000 |
| $X_3$ | 1.200 | 1.000 | 1.200 | 1.000 |
| $X_4$ | 0.900 | 1.000 | 0.900 | 1.000 |

Table 3: Constraint-related random variable parameters in experiment 1

|  | TASK1 | | | TASK2 | | |
| --- | --- | --- | --- | --- | --- | --- |
|  | Mean | Variance | Quantile | Mean | Variance | Quantile |
| $Y_{11}$ | 0.400 | 1.000 | 1.682 | 0.400 | 1.000 | 1.682 |
| $Y_{21}$ | 0.400 | 1.000 | 1.682 | 0.400 | 1.000 | 1.682 |
| $Y_{31}$ | 1.203 | 1.000 | 2.485 | 1.203 | 1.000 | 2.485 |
| $Y_{41}$ | 2.045 | 1.000 | 3.326 | 2.045 | 1.000 | 3.326 |

## D.3 PSEUDO-CODE OF ALL STRATEGIES

*This subsection summarizes the pseudo-code for all strategies used in numerical experiments.*

**USR.** *The USR strategy in Algorithm 2 samples each task, arm, and metric pair equally.*

---

**Algorithm 2** USR Algorithm

---

1: **Initialization.** Pull each $(a, i, j) \in [M] \times [K] \times [S + 1]$ $n_0$ times.
2: Set $t \leftarrow n_0 M K(S + 1)$, and update $\hat{\mathcal{P}}(t)$, $\boldsymbol{\omega}^*(\hat{\mathcal{P}}(t))$, $N_{i(S+1)}^a(t)$, $N_{ij}^a(t)$.
3: **while** $\inf_{\tilde{\mathcal{P}} \in \mathcal{A}(\hat{\mathcal{P}}(t))} f(\hat{\mathcal{P}}(t), \tilde{\mathcal{P}}) < \beta(t, \delta)$ **do**
4:
$$\pi^{t+1} = \underset{(a,i,j) \in [M] \times [K] \times [S+1]}{\arg\min} N_{ij}^a(t)$$
5:     Sample the $\pi^{t+1}$ and obtain one observation $Z_{t+1}$.
6:     Set $t \leftarrow t + 1$, and update $\hat{\mathcal{P}}(t)$, $\boldsymbol{\omega}^*(\hat{\mathcal{P}}(t))$, $N_{i(S+1)}^a(t)$, $N_{ij}^a(t)$.
7: **end while**
8: **Output.** Select $\hat{i}_\tau(a) = \arg\max_{i \in [K]} \hat{\mu}_i^a(\tau)$ s.t. $\hat{F}_{ij}^a(\tau) \geq \phi$, $\forall j \in [S]$, as the best arm for each task $a \in [M]$.

---

**ASR.** *The ASR strategy in Algorithm 3 utilizes the static sampling rule in (10) along with the empirical estimate $\hat{\mathcal{P}}(t)$ and updates the sampling rule adaptively. Note that, for simplicity, we set $\mathcal{D}_3^a = \mathcal{M}_2^a$ in implementation to avoid solving the optimization problem required to differentiate between $\mathcal{M}_1^a$ and $\mathcal{M}_2^a$. We then use the approximate sampling rule, which does not significantly impact the performance of the algorithm in our example.*

---

**Algorithm 3** ASR Algorithm

---

1: **Initialization.** Pull each $(a, i, j) \in [M] \times [K] \times [S+1]$ $n_0$ times.

2: Set $t \leftarrow n_0 M K(S+1)$, and update $\hat{\mathcal{P}}(t), \tilde{\boldsymbol{\omega}}(\hat{\mathcal{P}}(t))$ (according to (10)), $U_t, N^a_{i(S+1)}(t), N^a_{ij}(t)$.

3: **while** $\inf_{\tilde{\mathcal{P}} \in \mathcal{A}(\hat{\mathcal{P}}(t))} f(\hat{\mathcal{P}}(t), \tilde{\mathcal{P}}) < \beta(t, \delta)$ **do**

4:
$$\pi^{t+1} = \begin{cases} \arg\min N^a_{ij}(t) \text{ if } U_t \neq \emptyset \text{ or } \exists a \in [M], \{i^*(a, \hat{\mathcal{P}}(t))\} = \emptyset \text{ or } \hat{\mathcal{D}}^a_1(t) = \emptyset \\ \arg\max t\tilde{\omega}^a_{ij}(\hat{\mathcal{P}}(t)) - N^a_{ij}(t) \end{cases}$$

5:      Sample the $\pi^{t+1}$ and obtain one observation $Z_{t+1}$.

6:      Set $t \leftarrow t + 1$, and update $\hat{\mathcal{P}}(t), \tilde{\boldsymbol{\omega}}(\hat{\mathcal{P}}(t)), U_t, N^a_{i(S+1)}(t), N^a_{ij}(t)$.

7: **end while**

8: **Output.** Select $\hat{i}_\tau(a) = \arg\max_{i \in [K]} \hat{\mu}^a_i(\tau)$ s.t. $\hat{F}^a_{ij}(\tau) \geq \phi$, $\forall j \in [S]$, as the best arm for each task $a \in [M]$.

---

**Algorithm 4** SEQSR Algorithm

---

1: **Initialization.** Pull each $(a, i, j) \in [M] \times [K] \times [S+1]$ $n_0$ times.

2: Set $t \leftarrow n_0 M K(S+1)$, and update $\hat{\mathcal{P}}(t), \boldsymbol{\omega}(t), U_t, N^a_{i(S+1)}(t), N^a_{ij}(t)$.

3: **while** $\inf_{\tilde{\mathcal{P}} \in \mathcal{A}(\hat{\mathcal{P}}(t))} f(\hat{\mathcal{P}}(t), \tilde{\mathcal{P}}) < \beta(t, \delta)$ **do**

4:      $(a^t, i^t, j^t) = \arg\min_{(a,i,j) \in [M] \times [K] \times [S+1]} s_{a,i,j}(\hat{\mathcal{P}}(t), \boldsymbol{\omega}(t))$

5:
$$\pi^{t+1} = \begin{cases} \arg\min N^a_{ij}(t) \text{ if } U_t \neq \emptyset \text{ or } \exists a \in [M], \{i^*(a, \hat{\mathcal{P}}(t))\} = \emptyset \text{ or } \hat{\mathcal{D}}^a_1(t) = \emptyset \\ (a^t, i^t, j^t) \text{ if Condition 1} \\ (a^t, i^*(a, \hat{\mathcal{P}}(t)), j^t) \text{ if Condition 2} \end{cases}$$

6:      Sample the $\pi^{t+1}$ and obtain one observation $Z_{t+1}$.

7:      Set $t \leftarrow t + 1$, and update $\hat{\mathcal{P}}(t), \tilde{\boldsymbol{\omega}}(\hat{\mathcal{P}}(t)), U_t, N^a_{i(S+1)}(t), N^a_{ij}(t)$.

8: **end while**

9: **Output.** Select $\hat{i}_\tau(a) = \arg\max_{i \in [K]} \hat{\mu}^a_i(\tau)$ s.t. $\hat{F}^a_{ij}(\tau) \geq \phi$, $\forall j \in [S]$, as the best arm for each task $a \in [M]$.

---

Table 4: Objective-related random variable parameters in experiment 2

| | | TASK1 | | TASK2 | |
| --- | --- | --- | --- | --- | --- |
| | Mean | Variance | | Mean | Variance |
| $X_1$ | 0.800 | 0.160 | | 0.800 | 0.160 |
| $X_2$ | 0.600 | 0.240 | | 0.600 | 0.240 |
| $X_3$ | 0.900 | 0.090 | | 0.900 | 0.090 |
| $X_4$ | 0.400 | 0.240 | | 0.400 | 0.240 |

Table 5: Constraint-related random variable parameters in experiment 2

| | | TASK1 | | | TASK2 | |
| --- | --- | --- | --- | --- | --- | --- |
| | Mean | Variance | Quantile | Mean | Variance | Quantile |
| $Y_{11}$ | 0.350 | 1.000 | 1.632 | 0.350 | 1.000 | 1.632 |
| $Y_{21}$ | 0.450 | 1.000 | 1.732 | 0.450 | 1.000 | 1.732 |
| $Y_{31}$ | 1.500 | 1.000 | 2.782 | 1.500 | 1.000 | 2.782 |
| $Y_{41}$ | 1.800 | 1.000 | 3.082 | 1.800 | 1.000 | 3.082 |

**SEQSR.** *The SEQSR strategy in Algorithm 4 utilizes the sampling rule in (16)-(18) along with the empirical estimate $\hat{\mathcal{P}}(t)$ and update the sampling rule adaptively. In the Algorithm 4, condition 1 is:*

$$
\left( i^t \in \hat{\mathcal{D}}_2^a(t) \cup \hat{\mathcal{M}}_2^a(t) \cup \{i^*(a, \hat{\mathcal{P}}(t))\} \right) \bigcup
$$

$$
\left( i^t \in \hat{\mathcal{D}}_1^a(t) \cup \hat{\mathcal{M}}_1^a(t), \frac{N_{i^*(a,\hat{\mathcal{P}}(t))j^t}^{a^t}(t)}{N_{i^*(a,\hat{\mathcal{P}}(t))j^t}^{a^t}(t) + N_{i^t j^t}^{a^t}(t)} > \frac{(\omega_{i^*(a,\hat{\mathcal{P}}(t))j^t}^{a^t})^*(\hat{\mathcal{P}}(t))}{(\omega_{i^*(a,\hat{\mathcal{P}}(t))j^t}^{a^t})^*(\hat{\mathcal{P}}(t)) + (\omega_{i^t j^t}^{a^t})^*(\hat{\mathcal{P}}(t))} \right),
$$

*the condition 2 is:*

$$
\left( i^t \in \hat{\mathcal{D}}_1^a(t) \cup \hat{\mathcal{M}}_1^a(t), \frac{N_{i^*(a,\hat{\mathcal{P}}(t))j^t}^{a^t}(t)}{N_{i^*(a,\hat{\mathcal{P}}(t))j^t}^{a^t}(t) + N_{i^t j^t}^{a^t}(t)} \leq \frac{(\omega_{i^*(a,\hat{\mathcal{P}}(t))j^t}^{a^t})^*(\hat{\mathcal{P}}(t))}{(\omega_{i^*(a,\hat{\mathcal{P}}(t))j^t}^{a^t})^*(\hat{\mathcal{P}}(t)) + (\omega_{i^t j^t}^{a^t})^*(\hat{\mathcal{P}}(t))} \right).
$$

*Note that, for simplicity, we set $\mathcal{D}_3^a = \mathcal{M}_2^a$ in implementation to avoid solving the optimization problem required to differentiate between $\mathcal{M}_1^a$ and $\mathcal{M}_2^a$. We then use the approximate sampling rule, which does not significantly impact the performance of the algorithm in our example.*

**FWSR.** *The FWSR strategy in Algorithm 5 extends the algorithm in Wang et al. (2021) for the standard BAI problem.*

*According to Lemma 6, we know that*

$$
\mathcal{G}(\boldsymbol{\omega}, \mathcal{P}) = \min_{a \in [M]} \min \left( \min_{j \in [S]} f_{i^*(a,\mathcal{P})j}^a(\boldsymbol{\omega}, \mathcal{P}), \min_{i \in [K] \setminus \{i^*(a,\mathcal{P})\}} f_i^a(\boldsymbol{\omega}, \mathcal{P}) \right), \tag{141}
$$

*is a non-smooth concave function and*

$$f^a_{i^*(a,\mathcal{P})j}(\boldsymbol{\omega},\mathcal{P}) = \omega^a_{i^*(a,\mathcal{P})j}\, d(F^a_{i^*(a,\mathcal{P})j},\phi),$$

$$g^a_i(\boldsymbol{\omega},\mathcal{P}) = \begin{cases} \inf_{\tilde{\mu}^a_i > \tilde{\mu}^a_{i^*(a,\mathcal{P})}} \left( \omega^a_{i(S+1)}\, d(\mu^a_i,\tilde{\mu}^a_i) + \omega^a_{i^*(a,\mathcal{P})(S+1)}\, d(\mu^a_{i^*(a)},\tilde{\mu}^a_{i^*(a)}) \right), & \text{if } i \in \mathcal{D}^a_1 \\ \sum_{j \in \mathcal{B}^a_2(i)} \omega^a_{ij}\, d(F^a_{ij},\phi), & \text{if } i \in \mathcal{D}^a_2 \\ \inf_{\tilde{\mu}^a_i > \tilde{\mu}^a_{i^*(a,\mathcal{P})}} \left( \omega^a_{i(S+1)}\, d(\mu^a_i,\tilde{\mu}^a_i) + \omega^a_{i^*(a,\mathcal{P})(S+1)}\, d(\mu^a_{i^*(a,\mathcal{P})},\tilde{\mu}^a_{i^*(a)}) \right) \\ \quad + \sum_{j \in \mathcal{B}^a_2(i)} \omega^a_{ij}\, d(F^a_{ij},\phi), & \text{if } i \in \mathcal{D}^a_3 \end{cases}$$

$$f^a_i(\boldsymbol{\omega},\mathcal{P}) = \min\left( \min_{i\in\mathcal{D}^a_1} g^a_1(\boldsymbol{\omega},\mathcal{P}), \min_{i\in\mathcal{D}^a_2} g^a_2(\boldsymbol{\omega},\mathcal{P}), \min_{i\in\mathcal{D}^a_3} g^a_3(\boldsymbol{\omega},\mathcal{P}) \right).$$

$$(142)$$

*Then, we can compute that*

$$\frac{\partial f^a_{i^*(a,\mathcal{P})j}(\boldsymbol{\omega},\mathcal{P})}{\omega^a_{ij}} = \begin{cases} d(F^a_{i^*(a,\mathcal{P})j},\phi) & \text{if } i = i^*(a,\mathcal{P}), j \in [S] \\ 0 & \text{otherwise} \end{cases}$$

$$\frac{\partial g^a_i(\boldsymbol{\omega},\mathcal{P})}{\omega^a_{ij}} = \begin{cases} d(\mu^a_i,\mu^a_{i,i^*(a,\mathcal{P})}) & \text{if } i \in \mathcal{D}^a_1 \cup \mathcal{D}^a_3, j = S+1 \\ d(\mu^a_{i^*(a,\mathcal{P})},\mu^a_{i',i^*(a,\mathcal{P})}) & \text{if } i = i^*(a,\mathcal{P}), i' \in \mathcal{D}^a_1 \cup \mathcal{D}^a_3, j = S+1 \\ d(F^a_{ij},\phi) & \text{if } i \in \mathcal{D}^a_2 \cup \mathcal{D}^a_3, j \in \mathcal{B}^a_2(i) \\ 0 & \text{otherwise} \end{cases}$$

$$(143)$$

*Using (143), we can compute* $\nabla f^a_{i^*(a,\mathcal{P})j}(\boldsymbol{\omega},\mathcal{P}), \nabla g^a_i(\boldsymbol{\omega},\mathcal{P})$ *and define the* $r$*-sub-differential subspace in Wang et al. (2021) with* $r \in (0,1)$:

$$H_{\mathcal{G}(\boldsymbol{\omega},\mathcal{P})}(\boldsymbol{\omega},r) = cov\Bigg\{ \nabla f^a_{i^*(a,\mathcal{P})j}(\boldsymbol{\omega},\mathcal{P}), \nabla g^a_i(\boldsymbol{\omega},\mathcal{P}) \mid a \in [M], i \in [K], j \in [S], \qquad (144)$$

$$f^a_{i^*(a,\mathcal{P})j}(\boldsymbol{\omega},\mathcal{P}) < \mathcal{G}(\boldsymbol{\omega},\mathcal{P}) + r, g^a_i(\boldsymbol{\omega},\mathcal{P}) < \mathcal{G}(\boldsymbol{\omega},\mathcal{P}) + r \Bigg\},$$

*where* $cov\{\mathcal{I}\}$ *denotes the convex hull of the set* $\mathcal{I}$. *Next, we apply the Frank-Wolfe algorithm to adaptively update the sampling rule, as outlined in Algorithm 5. In the numerical experiment, we set* $r_t = t^{-0.9}/MK(S+1)$, *consistent with the setting used in (Wang et al., 2021).*

### D.4 Motivation for the Multi-task Setting

*For the multi-task BAI problem we consider, another natural approach is to solve each task individually. In this subsection, we discuss the motivation behind and the benefits of solving the tasks simultaneously.*

*First, we compare the numerical performance of the two methods across different numbers of tasks* $M$. *Each task consists of 4 arms with a single constraint. All tasks are homogeneous, sharing the same parameters as outlined in Tables 2-3.*

*We represent the methods that use SEQSR strategy and solves each task individually as Single-SEQSR, to achieve the performance guarantee* $\mathbb{P}(\forall a \in [M], i^*(a) = \hat{i}_\tau(a)) \geq 1-\delta$, *we control the error probability for each task to be less than* $\delta/M$.

*We refer to our methods that use the SEQSR strategy and solve all tasks simultaneously as Multi-SEQSR. To estimate the sample complexity, we conduct 1000 independent replications, with the results summarized in Table 6.*

*The results show that Multi-SEQSR requires fewer samples to achieve the same statistical guarantees. For example, when* $M = 8$, *Single-SEQSR incurs an additional sample cost of 18.94%, which may be attributed to the statistical conservatism inherent in the stopping rule.*

*Another advantage of the multi-task setting is the generality of its mathematical models, which can include the single-task problem as a special case. More importantly, the formulation and analysis*

---

**Algorithm 5** FWSR Algorithm

---

1: **Initialization.** Pull each $(a, i, j) \in [M] \times [K] \times [S+1]$ $n_0$ times.

2: Set $t \leftarrow n_0 M K(S+1)$, and update $\hat{\mathcal{P}}(t)$, $\boldsymbol{\omega}(t)$, $\boldsymbol{x}(t) \leftarrow (1/MK(S+1), \ldots, 1/MK(S+1))$, $N^a_{i(S+1)}(t)$, $N^a_{ij}(t)$.

3: **while** $\inf_{\tilde{\mathcal{P}} \in \mathcal{A}(\hat{\mathcal{P}}(t))} f(\hat{\mathcal{P}}(t), \tilde{\mathcal{P}}) < \beta(t, \delta)$ **do**

4:     **if** $\sqrt{\lfloor t/MK(S+1) \rfloor} \in \mathbb{N}$ or $\exists a \in [M], \left( \{i^*(a, \hat{\mathcal{P}}(t))\} = \emptyset \cup \hat{\mathcal{D}}^a_1(t) = \emptyset \right)$ **then**

5:         $\boldsymbol{z}(t+1) \leftarrow (1/MK(S+1), \ldots, 1/MK(S+1))$

6:     **else**

7:         $\boldsymbol{z}(t+1) \leftarrow \arg\max_{\boldsymbol{z} \in \Omega} \min_{h \in H_{\mathcal{G}(\boldsymbol{\omega}(t), \hat{\mathcal{P}}(t))}(\boldsymbol{x}(t), r_t)} < \boldsymbol{z} - \boldsymbol{x}(t), h >$

8:

9:     $\boldsymbol{x}(t+1) \leftarrow \frac{t}{t+1}\boldsymbol{x}(t) + \frac{1}{t+1}\boldsymbol{z}(t+1)$

10:

11:

$$\pi^{t+1} = \arg\max_{(a,i,j) \in [M] \times [K] \times [S+1]} \frac{\boldsymbol{x}(t+1)}{\boldsymbol{\omega}(t)}$$

12:     **end if**

13:     Sample the selected arm $\pi^{t+1}$ and obtain observation $Z_{t+1}$.

14:     Increment $t \leftarrow t+1$, and update $\hat{\mathcal{P}}(t)$, $\boldsymbol{\omega}(t)$, $N^a_{i(S+1)}(t)$, $N^a_{ij}(t)$.

15: **end while**

16: **Output:** For each task $a \in [M]$, select the best arm:

$$\hat{i}_\tau(a) = \arg\max_{i \in [K]} \hat{\mu}^a_i(\tau) \quad \text{s.t.} \quad \hat{F}^a_{ij}(\tau) \geq \phi, \ \forall j \in [S]$$

---

Table 6: Comparison of Single-SEQSR and Multi-SEQSR under different numbers of tasks

| $M$ | $\delta$ | Single-SEQSR | Multi-SEQSR | Ratio | Additional Cost |
|---|---|---|---|---|---|
| 2 | 0.1 | 2984.01 | 2745.81 | 1.09 | 8.67% |
| 4 | 0.2 | 5943.42 | 5292.85 | 1.12 | 12.29% |
| 6 | 0.24 | 9411.51 | 8053.55 | 1.17 | 16.86% |
| 8 | 0.24 | 13428.68 | 11290.69 | 1.19 | 18.94% |

*methods can be extended to scenarios where there is linear structure across tasks, as demonstrated in Section E.2.*

## D.5 EFFECT OF HYPER-PARAMETERS $n_0$

*Figure 8 and Table D.5 summarize the empirical sample complexity for 1000 independent runs of various strategies across different $n_0$ values for the Gaussian bandit. The results are consistent with those in Figure 1. ESR and SEQSR outperform the other benchmarks.*

Table 7: Empirical sample complexity for 1000 runs times with $\delta = 0.1$ and different $n_0$ for Gaussian bandit

| $n_0$ | USR | ASR | FWSR | SEQSR | ESR |
|---|---|---|---|---|---|
| 10 | 10179.68 | 4807.64 | 5692.85 | 2745.81 | 3792.85 |
| 20 | 10216.50 | 4751.39 | 5893.63 | 2806.00 | 3814.50 |
| 30 | 10293.07 | 4853.74 | 5956.22 | 2816.83 | 3848.68 |
| 40 | 10262.33 | 4794.75 | 5887.34 | 2943.61 | 3829.84 |
| 50 | 10114.14 | 4912.75 | 6024.18 | 3437.72 | 4100.23 |

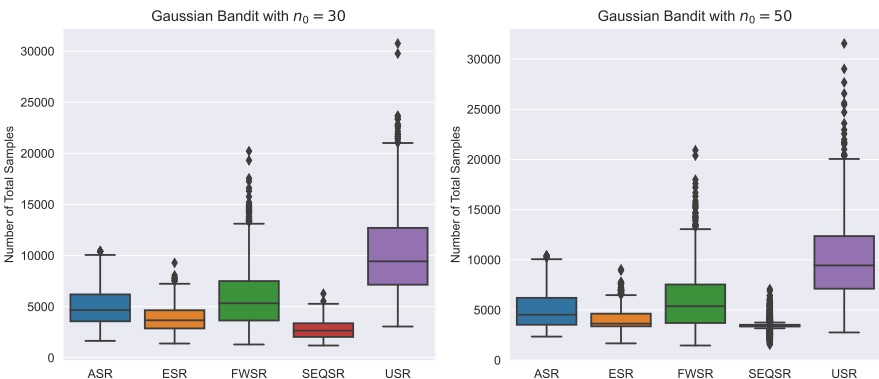

Figure 8: Empirical sample complexity for 1000 runs times with $\delta = 0.1$ and $n_0 = 30$ (left) and $n_0 = 50$ (right) for Gaussian bandit.

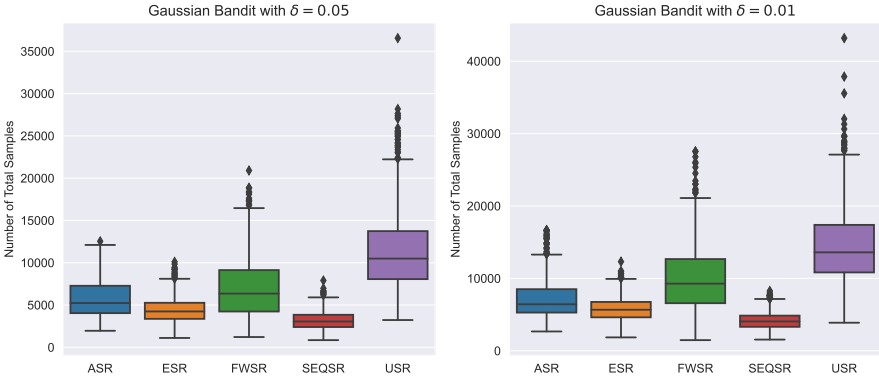

Figure 9: Empirical sample complexity for 1000 runs times with $n_0 = 10$ and $\delta = 0.05$ (left) and $\delta = 0.01$ (right) for Gaussian bandit.

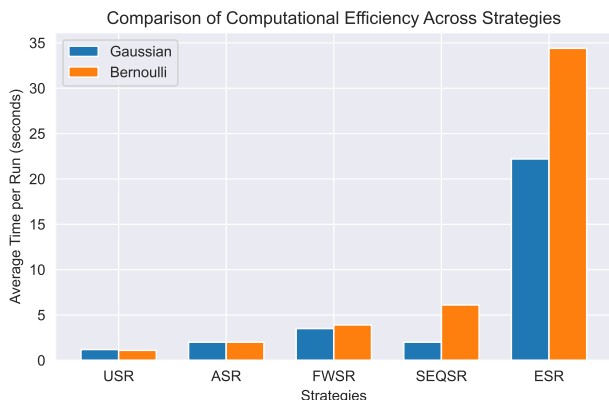

Figure 10: Average time per run (seconds) for different strategies in Gaussian and Bernoulli Bandit with $n_0 = 10$.

### D.6 EFFECT OF CONFIDENCE LEVEL $\delta$

*Figure 9 presents the empirical sample complexity for 1000 independent runs of various strategies across different values of $\delta$ for the Gaussian bandit. The results align with those in Figure 1. As $\delta$ decreases, all algorithms require more samples, with ESR and SEQSR outperforming the other benchmarks.*

### D.7 COMPUTATIONAL EFFICIENCY

*Figure 10 shows the average time per run (in seconds) for various strategies in both Gaussian and Bernoulli bandits, each using the same fixed sample size: 3000. The computational efficiency of USR, ASR, FWSR, and SEQSR is comparable and significantly higher than that of the ESR strategy.*

*In the Gaussian bandit example, SEQSR is much more efficient than in the Bernoulli bandit. This is because, in Bernoulli bandit, SEQSR uses the sampling rules in (16)-(18), which require solving an optimization problem to obtain $\boldsymbol{\omega}^*(\hat{\mathcal{P}}(t))$ in some iterations. However, in Gaussian bandit case, the optimal condition*

$$\sum_{i \in \mathcal{D}_1^a \cup \mathcal{M}_1^a} \frac{d(\mu_{i^*(a,\mathcal{P})}^a, \mu_{i,i^*(a,\mathcal{P})}^a)}{d(\mu_i^a, \mu_{i,i^*(a,\mathcal{P})}^a)} = 1, \forall a \in [M] \tag{145}$$

*is equivalent to*

$$(\omega_{i^*(a,\mathcal{P})(S+1)}^a)^*(\mathcal{P}) = \sigma_{i^*(a,\mathcal{P})(S+1)}^a \sqrt{\sum_{i \in \mathcal{D}_1^a \cup \mathcal{M}_1^a} \frac{((\omega_{i(S+1)}^a)^*(\mathcal{P}))^2}{(\sigma_{i(S+1)}^a)^2}}, \tag{146}$$

*Define*

$$(a^t, i^t, j^t) = \underset{(a,i,j) \in [M] \times [K] \times [S+1]}{\arg\min} s_{a,i,j}(\mathcal{P}, \boldsymbol{\omega}(t)). \tag{147}$$

*A more efficient sampling rule is to choose $(a^t, i^t, j^t)$ if $i^t \in \mathcal{D}_2^a \cup \mathcal{M}_2^a \cup \{i^*(a,\mathcal{P})\}$ or $i^t \in \mathcal{D}_1^a \cup \mathcal{M}_1^a$ with*

$$\omega_{i^*(a,\mathcal{P})(S+1)}^a(t) > \sigma_{i^*(a,\mathcal{P})(S+1)}^a \sqrt{\sum_{i \in \mathcal{D}_1^a \cup \mathcal{M}_1^a} \frac{(\omega_{i(S+1)}^a(t))^2}{(\sigma_{i(S+1)}^a)^2}}. \tag{148}$$

*and choose $(a^t, i^*(a,\mathcal{P}), j^t)$ otherwise. Since this sampling rule does not require solving an optimization problem, it is much more efficient.*

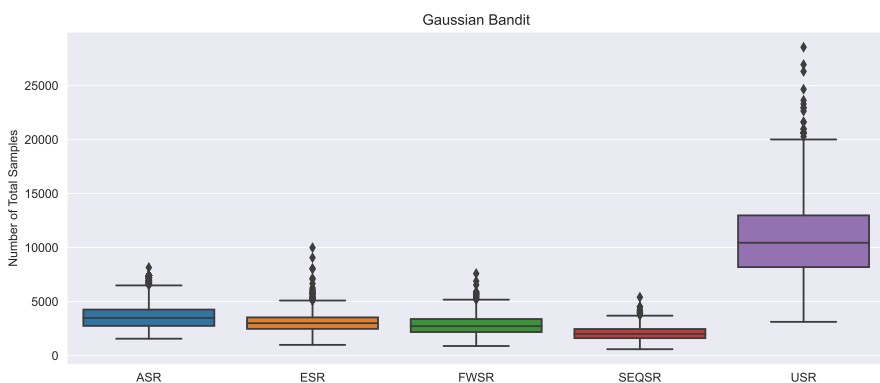

Figure 11: Empirical sample complexity for 1000 runs time with $n_0 = 10$ and $\delta = 0.1$ for Gaussian bandit with more arms.

Table 8: Objective-related random variable parameters in Gaussian bandit example with more arms

| | TASK1 | | TASK2 | |
| --- | --- | --- | --- | --- |
| | Mean | Variance | Mean | Variance |
| $X_1$ | 1.000 | 1.000 | 1.000 | 1.000 |
| $X_2$ | 0.700 | 1.000 | 0.700 | 1.000 |
| $X_3$ | 0.600 | 1.000 | 0.600 | 1.000 |
| $X_4$ | 1.300 | 1.000 | 1.300 | 1.000 |
| $X_5$ | 1.400 | 1.000 | 1.400 | 1.000 |
| $X_6$ | 0.700 | 1.000 | 0.700 | 1.000 |
| $X_7$ | 0.800 | 1.000 | 0.800 | 1.000 |

### D.8 GAUSSIAN BANDIT EXAMPLE WITH MORE ARMS

*In this subsection, we consider a larger Gaussian bandit example with 2 tasks, 7 arms, and 1 constraint. The mean and variance parameters of the corresponding random variables are summarized in Table 8 and 9. Other parameters are set as $b = 2.0$ and $\phi = 0.9$.*

*Figure 11 presents the empirical sample complexity for 1,000 runs of this example. The results indicate that SEQSR, with an average sample complexity of 2074.71, performs relatively better than the other strategies. FWSR, with an average sample complexity of 2862.06, and ESR, with ,3103.25 show comparable performance.*

## E EXTENSION TO LINEAR BANDIT

### E.1 PROOF OF THEOREM 4

**Theorem 4.** *Given a fixed confidence level $\delta \in (0,1)$. Under Assumption 3, for any linear BAI problem instance $\mathcal{P} \in \mathcal{S}$ and any strategy satisfying $\mathbb{P}\left(\forall a \in [M], i^*(a) = \hat{i}_\tau(a)\right) \geq 1 - \delta$,*

$$\mathbb{E}[\tau] \geq \mathcal{H}^*(\mathcal{P})kl(\delta, 1 - \delta), \qquad (149)$$

Table 9: Constraint-related random variable parameters in Gaussian bandit with more arms

| | | TASK1 | | | TASK2 | |
|---|---|---|---|---|---|---|
| | Mean | Variance | Quantile | Mean | Variance | Quantile |
| $Y_{11}$ | 0.350 | 1.000 | 1.632 | 0.350 | 1.000 | 1.632 |
| $Y_{21}$ | 0.450 | 1.000 | 1.732 | 0.450 | 1.000 | 1.732 |
| $Y_{31}$ | 0.450 | 1.000 | 1.732 | 0.450 | 1.000 | 1.732 |
| $Y_{41}$ | 1.650 | 1.000 | 2.932 | 1.650 | 1.000 | 2.932 |
| $Y_{51}$ | 1.650 | 1.000 | 2.932 | 1.650 | 1.000 | 2.932 |
| $Y_{61}$ | 1.750 | 1.000 | 3.032 | 1.750 | 1.000 | 3.032 |
| $Y_{71}$ | 1.750 | 1.000 | 3.032 | 1.750 | 1.000 | 3.032 |

*where* $\Lambda_j^a = \sum_{i \in [K]} \omega_{ij}^a \frac{x_i x_i^T}{(\sigma_{ij}^a)^2}, \Lambda_{S+1}^a = \sum_{i \in [K]} \omega_{i(S+1)}^a \frac{x_i x_i^T}{(\sigma_i^a)^2}$,

$$
\begin{aligned}
\mathcal{H}^*(\mathcal{P})^{-1} = \frac{1}{2} \sup_{\boldsymbol{\omega} \in \Omega} \min_{a \in [M]} \min \Bigg( & \min_{j \in [S]} \frac{(b - x_{i^*(a)}^T \theta_j^a)^2}{x_{i^*(a)}^T (\Lambda_j^a)^{-1} x_{i^*(a)}}, \min_{i \in \mathcal{D}_1^a} \frac{((x_i - x_{i^*(a)})^T \theta_{S+1}^a)^2}{\|x_i - x_{i^*(a)}\|_{(\Lambda_{S+1}^a)^{-1}}^2}, \\
& \min_{i \in \mathcal{D}_2^a} \sum_{j \in \mathcal{B}_2^a(x_i)} \frac{(b - x_i^T \theta_j^a)^2}{x_i^T (\Lambda_j^a)^{-1} x_i}, \min_{i \in \mathcal{D}_3^a} \left[ \frac{((x_i - x_{i^*(a)})^T \theta_{S+1}^a)^2}{\|x_i - x_{i^*(a)}\|_{(\Lambda_{S+1}^a)^{-1}}^2} + \sum_{j \in \mathcal{B}_2^a(x_i)} \frac{(b - x_i^T \theta_j^a)^2}{x_i^T (\Lambda_j^a)^{-1} x_i} \right] \Bigg).
\end{aligned}
\tag{150}
$$

*Proof.* The analysis follows the same approach as in the Theorem 1 and we initiate the analysis for the linear bandit starting from (35)

$$
\mathbb{E}[\tau] \geq \mathcal{H}^*(\mathcal{P}) \text{kl}(\delta, 1 - \delta),
\tag{151}
$$

where

$$
\mathcal{H}^*(\mathcal{P})^{-1} = \sup_{\boldsymbol{\omega} \in \Omega} \inf_{\tilde{\mathcal{P}} \in \mathcal{A}(\mathcal{P})} \sum_{a \in [M]} \sum_{i \in [K]} \left( \omega_{i(S+1)}^a d(x_i^T \theta_{S+1}^a, x_i^T \tilde{\theta}_{S+1}^a) + \sum_{j \in [S]} \omega_{ij}^a d(x_i^T \theta_j^a, x_i^T \tilde{\theta}_j^a) \right).
\tag{152}
$$

For each task $a \in [M]$ and constraint $j \in [S]$, define

$$
\mathcal{C}_{i^*(a, \mathcal{P})j}^a = \{\tilde{\boldsymbol{\theta}} : x_{i^*(a, \mathcal{P})}^T \tilde{\theta}_j^a > b\}.
\tag{153}
$$

For each task $a \in [M]$ and arm $i \in [K] \setminus \{i^*(a, \mathcal{P})\}$, define

$$
\mathcal{C}_i^a = \{\tilde{\boldsymbol{\theta}} : (x_i - x_{i^*(a, \mathcal{P})})^T \tilde{\theta}_{S+1}^a \geq 0, x_i^T \tilde{\theta}_j^a \leq b, \forall j \in [S]\},
\tag{154}
$$

where $\tilde{\boldsymbol{\theta}} = (\tilde{\theta}_j^a)_{a \in [M], j \in [S+1]} = \tilde{\mathcal{P}}$ denote the problem instance $\tilde{\mathcal{P}}$.

According to (39), we have

$$
\mathcal{A}(\mathcal{P}) = \bigcup_{a \in [M]} \left( \left( \bigcup_{j \in [S]} \mathcal{C}_{i^*(a, \mathcal{P})j}^a \right) \bigcup \left( \bigcup_{i \in [K] \setminus \{i^*(a, \mathcal{P})\}} \mathcal{C}_i^a \right) \right),
\tag{155}
$$

Define

$$
f(\boldsymbol{\omega}, \mathcal{P}, \tilde{\mathcal{P}}) = \sum_{a \in [M]} \sum_{i \in [K]} \left( \omega_{i(S+1)}^a d(x_i^T \theta_{S+1}^a, x_i^T \tilde{\theta}_{S+1}^a) + \sum_{j \in [S]} \omega_{ij}^a d(x_i^T \theta_j^a, x_i^T \tilde{\theta}_j^a) \right),
\tag{156}
$$

and $\mathcal{G}(\boldsymbol{\omega}, \mathcal{P}) = \inf_{\tilde{\mathcal{P}} \in \mathcal{A}(\mathcal{P})} f(\boldsymbol{\omega}, \mathcal{P}, \tilde{\mathcal{P}})$. Then, according to (155), we have

$$
\mathcal{G}(\boldsymbol{\omega}, \mathcal{P}) = \min_{a \in [M]} \min \left( \min_{j \in [S]} f_{i^*(a, \mathcal{P})j}^a(\boldsymbol{\omega}, \mathcal{P}), \min_{i \in [K] \setminus \{i^*(a, \mathcal{P})\}} f_i^a(\boldsymbol{\omega}, \mathcal{P}) \right),
\tag{157}
$$

where $f_{i^*(a,\mathcal{P})j}^a(\boldsymbol{\omega}, \mathcal{P}) = \inf_{\tilde{\mathcal{P}} \in \mathcal{C}_{i^*(a,\mathcal{P})j}^a} f(\boldsymbol{\omega}, \mathcal{P}, \tilde{\mathcal{P}})$, $f_i^a(\boldsymbol{\omega}, \mathcal{P}) = \inf_{\tilde{\mathcal{P}} \in \mathcal{C}_i^a} f(\boldsymbol{\omega}, \mathcal{P}, \tilde{\mathcal{P}})$.

We first consider $f_{i^*(a,\mathcal{P})j}^a(\boldsymbol{\omega}, \mathcal{P})$,

$$
\begin{aligned}
f_{i^*(a,\mathcal{P})j}^a(\boldsymbol{\omega}, \mathcal{P}) &= \inf_{\tilde{\mathcal{P}} \in \mathcal{C}_{i^*(a,\mathcal{P})j}^a} f(\boldsymbol{\omega}, \mathcal{P}, \tilde{\mathcal{P}}) \\
&= \inf_{x_{i^*(a,\mathcal{P})}^T \tilde{\theta}_j^a > b} \sum_{a \in [M]} \sum_{i \in [K]} \left( \omega_{i(S+1)}^a d(x_i^T \theta_{S+1}^a, x_i^T \tilde{\theta}_{S+1}^a) + \sum_{j \in [S]} \omega_{ij}^a d(x_i^T \theta_j^a, x_i^T \tilde{\theta}_j^a) \right) \\
&= \inf_{x_{i^*(a,\mathcal{P})}^T \tilde{\theta}_j^a > b} \sum_{i \in [K]} \sum_{j \in [S]} \omega_{ij}^a d(x_i^T \theta_j^a, x_i^T \tilde{\theta}_j^a) \\
&= \min_{j \in [S]} \inf_{x_{i^*(a,\mathcal{P})}^T \tilde{\theta}_j^a > b} \sum_{i \in [K]} \omega_{ij}^a d(x_i^T \theta_j^a, x_i^T \tilde{\theta}_j^a).
\end{aligned}
\tag{158}
$$

According to Assumption 3, we have

$$
\begin{aligned}
\sum_{i \in [K]} \omega_{ij}^a d(x_i^T \theta_j^a, x_i^T \tilde{\theta}_j^a) &= \sum_{i \in [K]} \omega_{ij}^a \frac{(\theta_j^a - \tilde{\theta}_j^a)^T x_i x_i^T (\theta_j^a - \tilde{\theta}_j^a)}{2(\sigma_{ij}^a)^2} \\
&= (\theta_j^a - \tilde{\theta}_j^a)^T \sum_{i \in [K]} \omega_{ij}^a \frac{x_i x_i^T}{2(\sigma_{ij}^a)^2} (\theta_j^a - \tilde{\theta}_j^a) \\
&= \frac{1}{2} (\theta_j^a - \tilde{\theta}_j^a)^T \Lambda_j^a (\theta_j^a - \tilde{\theta}_j^a),
\end{aligned}
\tag{159}
$$

where $\Lambda_j^a = \sum_{i \in [K]} \omega_{ij}^a \frac{x_i x_i^T}{(\sigma_{ij}^a)^2}$.

Consider the optimization problem

$$
\inf_{x_{i^*(a,\mathcal{P})}^T \tilde{\theta}_j^a > b} \frac{1}{2} (\theta_j^a - \tilde{\theta}_j^a)^T \Lambda_j^a (\theta_j^a - \tilde{\theta}_j^a).
\tag{160}
$$

The KKT condition can be derived as

$$
\Lambda_j^a (\theta_j^a - \tilde{\theta}_j^a) + \lambda x_{i^*(a,\mathcal{P})} = 0,
\tag{161}
$$

$$
x_{i^*(a,\mathcal{P})}^T \tilde{\theta}_j^a = b.
\tag{162}
$$

According to (161), we have

$$
\tilde{\theta}_j^a = \theta + \lambda (\Lambda_j^a)^{-1} x_{i^*(a,\mathcal{P})}.
\tag{163}
$$

Plug (163) into (162), we have

$$
\lambda = \frac{b - x_{i^*(a,\mathcal{P})}^T \theta_j^a}{x_{i^*(a,\mathcal{P})}^T (\Lambda_j^a)^{-1} x_{i^*(a,\mathcal{P})}}.
\tag{164}
$$

Then we have

$$
\tilde{\theta}_j^a = \theta_j^a + \frac{b - x_{i^*(a,\mathcal{P})}^T \theta_j^a}{x_{i^*(a,\mathcal{P})}^T (\Lambda_j^a)^{-1} x_{i^*(a,\mathcal{P})}} (\Lambda_j^a)^{-1} x_{i^*(a,\mathcal{P})},
\tag{165}
$$

with the optimal value as

$$
\frac{1}{2} \frac{(b - x_{i^*(a,\mathcal{P})}^T \theta_j^a)^2}{x_{i^*(a,\mathcal{P})}^T (\Lambda_j^a)^{-1} x_{i^*(a,\mathcal{P})}}.
\tag{166}
$$

Next, we analyze $\min_{i \in [K] \setminus \{i^*(a,\mathcal{P})\}} f_i^a(\boldsymbol{\omega}, \mathcal{P})$, notice that

$$
\min_{i \in [K] \setminus \{i^*(a,\mathcal{P})\}} f_i^a(\boldsymbol{\omega}, \mathcal{P}) = \min \left( \min_{i \in \mathcal{D}_1^a} f_i^a(\boldsymbol{\omega}, \mathcal{P}), \min_{i \in \mathcal{D}_2^a} f_i^a(\boldsymbol{\omega}, \mathcal{P}), \min_{i \in \mathcal{D}_3^a} f_i^a(\boldsymbol{\omega}, \mathcal{P}) \right),
\tag{167}
$$

and we have that

$$
\min_{i \in \mathcal{D}_1^a} f_i^a(\boldsymbol{\omega}, \mathcal{P}) = \min_{i \in \mathcal{D}_1^a} \inf_{\tilde{\mathcal{P}} \in \mathcal{C}_i^a} f(\boldsymbol{\omega}, \mathcal{P}, \tilde{\mathcal{P}})
$$

$$
= \min_{i \in \mathcal{D}_1^a} \inf_{\tilde{\mathcal{P}} \in \mathcal{C}_i^a} \sum_{a \in [M]} \sum_{i \in [K]} \left( \omega_{i(S+1)}^a d(x_i^T \theta_{S+1}^a, x_i^T \tilde{\theta}_{S+1}^a) + \sum_{j \in [S]} \omega_{ij}^a d(x_i^T \theta_j^a, x_i^T \tilde{\theta}_j^a) \right)
$$

$$
= \min_{i \in \mathcal{D}_1^a} \inf_{(x_i - x_{i^*(a,\mathcal{P})})^T \tilde{\theta}_{S+1}^a \geq 0} \sum_{i \in [K]} \omega_{i(S+1)}^a d(x_i^T \theta_{S+1}^a, x_i^T \tilde{\theta}_{S+1}^a)
$$

$$
= \min_{i \in \mathcal{D}_1^a} \inf_{(x_i - x_{i^*(a,\mathcal{P})})^T \tilde{\theta}_{S+1}^a \geq 0} (\theta_{S+1}^a - \tilde{\theta}_{S+1}^a)^T \sum_{i \in [K]} \omega_{i(S+1)}^a \frac{x_i x_i^T}{2(\sigma_i^a)^2} (\theta_{S+1}^a - \tilde{\theta}_{S+1}^a)
$$

$$
= \frac{1}{2} \min_{i \in \mathcal{D}_1^a} \inf_{(x_i - x_{i^*(a,\mathcal{P})})^T \tilde{\theta}_{S+1}^a \geq 0} (\theta_{S+1}^a - \tilde{\theta}_{S+1}^a)^T \Lambda_{S+1}^a (\theta_{S+1}^a - \tilde{\theta}_{S+1}^a),
$$

$$(168)$$

where $\Lambda_{S+1}^a = \sum_{i \in [K]} \omega_{i(S+1)}^a \frac{x_i x_i^T}{(\sigma_i^a)^2}$.

By solving the optimization problem

$$
\inf_{(x_i - x_{i^*(a,\mathcal{P})})^T \tilde{\theta}_{S+1}^a \geq 0} (\theta_{S+1}^a - \tilde{\theta}_{S+1}^a)^T \Lambda_{S+1}^a (\theta_{S+1}^a - \tilde{\theta}_{S+1}^a), \tag{169}
$$

we have the optimal value as

$$
\min_{i \in \mathcal{D}_1^a} \frac{((x_i - x_{i^*(a,\mathcal{P})})^T \theta_j^a)^2}{\|x_i - x_{i^*(a,\mathcal{P})}\|_{(\Lambda_{S+1}^a)^{-1}}^2}. \tag{170}
$$

The analysis for $\min_{i \in \mathcal{D}_2^a} f_i^a(\boldsymbol{\omega}, \mathcal{P})$ and $\min_{i \in \mathcal{D}_3^a} f_i^a(\boldsymbol{\omega}, \mathcal{P})$ are similar.

Finally, we conclude that

$$
\mathcal{H}^*(\mathcal{P})^{-1} = \frac{1}{2} \sup_{\boldsymbol{\omega} \in \Omega} \min_{a \in [M]} \min \left( \min_{j \in [S]} \frac{(b - x_{i^*(a)}^T \theta_j^a)^2}{x_{i^*(a)}^T (\Lambda_j^a)^{-1} x_{i^*(a)}}, \min_{i \in \mathcal{D}_1^a} \frac{((x_i - x_{i^*(a)})^T \theta_{S+1}^a)^2}{\|x_i - x_{i^*(a)}\|_{(\Lambda_{S+1}^a)^{-1}}^2}, \right.
$$
$$
\left. \min_{i \in \mathcal{D}_2^a} \sum_{j \in \mathcal{B}_2^a(x_i)} \frac{(b - x_i^T \theta_j^a)^2}{x_i^T (\Lambda_j^a)^{-1} x_i}, \min_{i \in \mathcal{D}_3^a} \left[ \frac{((x_i - x_{i^*(a)})^T \theta_{S+1}^a)^2}{\|x_i - x_{i^*(a)}\|_{(\Lambda_{S+1}^a)^{-1}}^2} + \sum_{j \in \mathcal{B}_2^a(x_i)} \frac{(b - x_i^T \theta_j^a)^2}{x_i^T (\Lambda_j^a)^{-1} x_i} \right] \right). \tag{171}
$$

$\square$

### E.2 LINEAR STRUCTURES ACROSS TASKS

*In this subsection, we introduce a new multi-task BAI problem with constraints, incorporating linear structural information among tasks. In personalized medicine, the agent aims to identify the best drug for each patient, represented by a feature vector capturing demographics and physical conditions. Patients with similar features are likely to experience similar efficacy and side effects. Thus, observations from one patient can inform others, enabling the agent to improve sampling efficiency by utilizing structural information across patients.*

*Consider that each task $a$ is associated with a feature vector $c^a \in \mathbb{R}^d$. For simplicity, we assume that each arm $i$ has two performance metrics $(X_i^a, Y_i^a) \in \mathbb{R}^2$ ($S = 2$), represented as linear functions of the features $c^a, a \in [M]$. The results can be extended to settings with multiple constraints using our previous method.*

**Assumption 4.** *There exist unknown parameter $\beta_i, \gamma_i, i \in [K]$ such that $X_i^a = \beta_i^T c^a + \epsilon$ and $Y_i^a = \gamma_i^T c^a + \epsilon$, where $\epsilon \sim N(0, \sigma^2)$, $\|c^a\|_2 \leq 1, \forall a \in [M]$, and $\|\beta_i^T\|_2 \leq 1, \|\gamma_i\|_2 \leq 1, \forall i \in [K]$.*

*The agent needs to solve the following optimization problem for each task*

$$
\max_{i \in [K]} \beta_i^T c^a \ \ s.t. \ \ \gamma_i^T c^a \leq b. \tag{172}
$$

**Theorem 6.** *Given a fixed confidence level $\delta \in (0,1)$. Under Assumption 4, for any linear BAI problem instance $\mathcal{P} \in \mathcal{S}$ and any strategy satisfying $\mathbb{P}\left(\forall a \in [M], i^*(a) = \hat{i}_\tau(a)\right) \geq 1 - \delta$,*

$$\mathbb{E}[\tau] \geq \mathcal{H}^*(\mathcal{P})kl(\delta, 1 - \delta), \tag{173}$$

*where $\Lambda_i = \sum_{a \in [M]} \frac{\omega_{i1}^a}{\sigma^2} c^a (c^a)^T, \Theta_i = \sum_{a \in [M]} \frac{\omega_{i2}^a}{\sigma^2} c^a (c^a)^T,$*

$$\mathcal{H}^*(\mathcal{P})^{-1} = \frac{1}{2} \sup_{\boldsymbol{\omega} \in \Omega} \min_{a \in [M]} \min \left( \frac{(b - \gamma_{i^*(a)}^T c^a)^2}{(c^a)^T \Lambda_{i^*(a)}^{-1} c^a}, \min_{i \in \mathcal{D}_1^a} \frac{((\beta_{i^*(a)} - \beta_i)^T c^a)^2}{(c^a)^T (\Theta_i^{-1} + \Theta_{i^*(a)}^{-1}) c^a}, \right.$$

$$\left. \min_{i \in \mathcal{D}_2^a} \frac{(b - \gamma_i^T c^a)^2}{(c^a)^T \Lambda_i^{-1} c^a}, \min_{i \in \mathcal{D}_3^a} \left[ \frac{((\beta_{i^*(a)} - \beta_i)^T c^a)^2}{(c^a)^T (\Theta_i^{-1} + \Theta_{i^*(a)}^{-1}) c^a} + \frac{(b - \gamma_i^T c^a)^2}{(c^a)^T \Lambda_i^{-1} c^a} \right] \right). \tag{174}$$

*Proof.* The proof follows the same approach as that of Theorem 4 and is therefore omitted. $\square$

