# OpenReview forum: "Multi-Task Best Arm Identification with Risk Constraint"
_ICLR.cc/2025/Conference — Submitted to ICLR 2025_

### Official Review · Reviewer_XQSZ · 2024-10-28

**Soundness:** 4
**Presentation:** 4
**Contribution:** 4
**Rating:** 8
**Confidence:** 3

**Summary:**

This paper presents a novel approach to the multi-task best arm identification (BAI) problem with risk constraints in a fixed-confidence setting. The authors derive a lower bound on sample complexity, establish optimality conditions for sampling ratios, and extend the Track-and-Stop strategy. They provide numerical experiments demonstrating the performance of their proposed algorithm.

**Strengths:**

1. The introduction of a multi-task BAI problem with risk constraints is a significant contribution to the literature. The problem is well-motivated with real-world applications.

2. The derivation of tight, instance-dependent lower bounds on sample complexity and optimality conditions provides a solid theoretical foundation for the proposed approach.

3. The experiments validate the algorithm's performance, showcasing its effectiveness compared to existing benchmarks.

**Weaknesses:**

(1) While the paper claims to address a multi-task best arm identification (BAI) problem with risk constraints by extending the work of Garivier & Kaufmann (2016), the contributions seem to overlap with existing works in BAI with safety constraints, such as the study by Wang et al. (2022). It would be beneficial for the authors to more explicitly differentiate their approach from previous studies, providing a clearer rationale for the novelty of their contributions.

(2) There are missing periods at the end of formulas, such as in equations (8), (9), and (16). Please ensure all equations are properly punctuated for consistency and clarity.


Reference:

[1]Wang, Z., Wagenmaker, A. J., & Jamieson, K. (2022, May). Best arm identification with safety constraints. In International Conference on Artificial Intelligence and Statistics (pp. 9114-9146). PMLR.

[2]Kaufmann, E., Cappé, O., & Garivier, A. (2016). On the complexity of best-arm identification in multi-armed bandit models. The Journal of Machine Learning Research, 17(1), 1-42.

**Questions:**

1.  How do you envision incorporating additional constraints, such as fairness, into your model? Are there specific challenges you anticipate?

2. Can you provide insights into the robustness of your results under varying conditions, such as experiments with different configurations?

---

> ### Author Response · Authors · 2024-11-21
>
> Thank you for your thoughtful and constructive review. We greatly appreciate your positive feedback on our paper.
>
> **Response to Weakness:**
>
> Our formulation is related to that of Wang et al. (2022), as both consider multiple performance metrics and incorporate certain metrics as constraints. However, our study addresses a different setting and makes distinct contributions.
>
> **Focus on Multi-Task and Risk Constraints.** Unlike Wang et al. (2022), which deals with a single-task BAI problem, our formulation is designed for a multi-task setting with risk constraints. Our method can simultaneously handle multiple tasks and identify the best arm that satisfies these risk constraints.
>
> **Fundamental Difference in Constraint Handling.** In Wang et al. (2022), the agent is required to select arms $i_t$ and values $a_t$ (as defined in their paper) that satisfy predefined safety conditions at each step $t$. In contrast, our setting involves pure exploration, where the agent evaluates all tasks, arms, and constraints without such immediate restrictions.
>
> **Estimating Feasibility and Optimality.** Our approach emphasizes simultaneously estimating both feasibility and optimality, which requires balancing the exploration efforts across different tasks, arms, and constraints. This is a key differentiator and adds to the theoretical novelty of our work compared to existing studies, e.g., Wang et al. (2022).
>
> We will further elaborate on these distinctions in comparison to other works on BAI with safety constraints and clarify our contributions in the revised version of the paper.
>
> We have ensured that all formulas, including those in equations (8), (9), and (16), are properly punctuated with periods for consistency and clarity in the revised version.
>
> **Response to Questions:**
>
> Our formulation and analysis method can be extended to incorporate additional constraints, such as fairness. For instance, consider a fairness constraint similar to that in [1]. In this case, the sampling rule needs to satisfy the pre-specified constraint:
>     \begin{equation}
>         \frac{\mathbb{E}[N^a_{ij}(t)]}{\mathbb{E}[\tau]} \geq p^a_{ij},\forall a\in[M], i\in[K], j\in [S+1],
>     \end{equation}
>     where $\mathbb{E}[N^a_{ij}(t)]$  represents the number of samples of corresponding metrics up to time $t$, and $p^a_{ij}\geq 0$ is the minimum proportion required for fairness.
>     Given that this constraint is imposed on the sampling rule, we can derive similar sample complexity results as in Theorem 1 using the same analysis method. Specifically, the sample complexity
>     \begin{equation}
>         \mathcal{H}^*(\mathcal{P})^{-1} = \max_{\omega\in\Omega}\min_{a\in[M]}\min(V^a_1(\omega),V^a_2(\omega),V^a_3(\omega),V^a_4(\omega)),
>     \end{equation}
>     where
>     \begin{equation}
>         \Omega =\\{\omega \in \mathbb{R}^{MK(S+1)}:\sum_{a\in[M],i\in[K],j\in[S+1]}\omega^a_{ij}=1,\omega^a_{ij}\geq p^a_{ij},\forall a\in[M],i\in[K],j\in[S+1]\\}.
>     \end{equation}
>     The main challenge in incorporating such constraints is to derive the optimal conditions that the solution $\omega^*$ must satisfy and to quantify the price of fairness in this setting. This involves determining how much additional sample complexity is required to meet the fairness constraints while maintaining the overall performance.
>
> Our method is robust across different configurations because, for different distributions (like Gaussian and Bernoulli), we can compute the optimal sampling rule $\omega^*(\hat{\mathcal{P}}(t))$ using the empirical estimates $\hat{\mathcal{P}}(t)$. The optimal rule $\omega^*(\hat{\mathcal{P}}(t))$ dynamically balances different tasks and constraints, effectively managing the trade-off between optimality and feasibility. This balance enhances sampling efficiency. As more samples are collected, the empirical estimates $\hat{\mathcal{P}}(t)$ are updated adaptively. Consequently, $\omega^* (\hat{\mathcal{P}}(t))$ gradually converges to the true optimal sampling rule, leading to a lower sample complexity compared to non-optimal sampling rules.

---

> > ### Comment · Reviewer_XQSZ · 2024-11-26
> >
> > I appreciate the authors for the detailed explanation. I decided to maintain my score.

---

> > > ### Author Response · Authors · 2024-11-26
> > > **Thank you for your response**
> > >
> > > Thank you once again for your detailed and valuable review. We truly appreciate your time and insightful suggestions!

---

### Official Review · Reviewer_uMef · 2024-10-29

**Soundness:** 3
**Presentation:** 1
**Contribution:** 2
**Rating:** 5
**Confidence:** 4

**Summary:**

This paper studies the problem of best arm identification with fixed confidence in a multi-task multi-objective constrained optimization setting. For each task and an arm, there are multiple metrics to consider with one primary metric that needs to be optimized while satisfying the quantile constraint for other metrics. At each time, the agent can choose a (task, metric, arm) tuple to sample, and receives the reward feedback of this specific chosen tuple, and no feedback from other metrics can be observed. The goal is to find for each task the best arm a^* which maximizes the primary metric while satisfying the constraint for all other metrics. The paper first provide and study a lower bound on the sample complexity from which the oracle static sampling strategy can be computed which requires the knowledge of the ground-truth reward of all arms, tasks and metrics. Then, the paper uses the same idea as [1] to convert the lower bound into a track-and-stop algorithm with asymptotic optimality guarantee. Computationally efficient algorithm and extension to linear bandits are also studied.

[1] Garivier, Aurélien, and Emilie Kaufmann. "Optimal best arm identification with fixed confidence." Conference on Learning Theory. PMLR, 2016.

**Strengths:**

This paper formulates the multi-task multi-metric best arm identification problem which models a wide range of applications. The study of quantile constraints as risk measure is of novelty in the best arm identification literature which generalizes the traditional constrained bandit problems which focus purely on expectation. The theoretical lower bound and a track-and-stop based matching algorithm are provided with provable guarantees. The paper also provides some insights towards instance-dependent hardness of the problem through analysis of the lower bound and the sample complexity. Lemma 1, Lemma2, and Proposition 1 provides some understanding on how the sub-optimality gaps and variances affect the problem hardness, and how the sample complexity scales with the number of tasks, arms, and metrics. Computationally efficient algorithms are also proposed in the paper which shows good performance in simulation results.

**Weaknesses:**

1. Presentation: The main body presentation of this paper is somewhat not satisfactory due to 1) the studied problem is unnecessarily complicated (also see 2. Formulation) which has too many variables and quantities, and 2) the notation system is not well established and the abuse of notation is quite frequent without declaration in the main body. The notations and equations in the main body are much too involved for general readers and the authors should reconsider to simplify the paper presentation structure and avoid unnecessary notations in the main body.

2. Formulation: The multi-task problem seems unnecessarily complicated. Even though the authors does not make very clear whether there is correlation between tasks and metrics in section 2, it seems in the paper's formulation, each task is independent of one another (even in the linear bandit formulation), and therefore the agent can solve the best arm identification problem of each task independently with certain confidence \delta' one-by-one, and combine the tasks together in the end with the help of independence among tasks. It also seems in the lower bound Theorem. 1, the inverse of complexity constant H^* is defined by equalizing the complexity of each task a, which essentially equalizes the probability of error for each individual task in asymptotic regime, which makes calculating the probability of error for each individual task possible.

3. Algorithm and Analysis: The paper follows a standard track-and-stop algorithm development and analysis machinery and naturally inherits the downside of track-and-stop type algorithms [1], i.e., the proposed Algorithm 1 needs to exactly solve a min-max problem at each step, which is computationally heavy and cannot be used in practice, and the provable guarantee Theorem 3 only holds for asymptotic regimes where \delta -> 0 which does not imply finite-time guarantees (which is also why ESR is worse than SEQSR in simulation). Based on the vast amount of track-and-stop literature in various bandits[1][2][3][4] and RL problems[5][6], the proposed algorithm and performance guarantees are somewhat expected. Even though computationally effecient algorithm is provided, it does not have provable sample complexity guarantee even in simplified models.

[1] Garivier, Aurélien, and Emilie Kaufmann. "Optimal best arm identification with fixed confidence." Conference on Learning Theory. PMLR, 2016.

[2] Jedra, Yassir, and Alexandre Proutiere. "Optimal best-arm identification in linear bandits." Advances in Neural Information Processing Systems 33 (2020): 10007-10017.

[3] Barrier, Antoine, Aurélien Garivier, and Tomáš Kocák. "A non-asymptotic approach to best-arm identification for gaussian bandits." International Conference on Artificial Intelligence and Statistics. PMLR, 2022.

[4] Russac, Yoan, et al. "A/B/n Testing with Control in the Presence of Subpopulations." Advances in Neural Information Processing Systems 34 (2021): 25100-25110.

[5] Al Marjani, Aymen, and Alexandre Proutiere. "Adaptive sampling for best policy identification in markov decision processes." International Conference on Machine Learning. PMLR, 2021.

[6] Taupin, Jerome, Yassir Jedra, and Alexandre Proutiere. "Best policy identification in linear mdps." 2023 59th Annual Allerton Conference on Communication, Control, and Computing (Allerton). IEEE, 2023.

**Questions:**

1. Is there any specific reason that the best arm identification problem of all tasks need to be considered and solved all-together?
2. Is it possible to derive performance guarantees (asymptotic or finite-time) for the SEQSR algorithm which can be quantitatively compared with H^* or H in proposition 1?
3. The formulation that the agent needs to choose a metric at each time-step and only observes the reward of this metric lacks motivation. Even though this formulation is statistically harder, it also produces enough variables \omega in Theorem 1, so that one could convert the lower bound to the optimality condition in Theorem 2. The reviewer is curious on whether the algorithm development and analysis framework will still work for the setting where the reward of all metrics can be observed once an arm is pulled, i.e., Theorem 5. Is the algorithm development framework in the main body still applicable to Theorem 5? is it possible to derive the optimality condition for Theorem 5 analogous to Theorem 2?

Typo: line 409 ESR -> USR (seems this is correct according to the figure)

---

> ### Author Response · Authors · 2024-11-21
>
> Thank you for your detailed and thoughtful review. We greatly appreciate your suggestions regarding the presentation, formulation, algorithm, and analysis sections of our paper, which have been invaluable in improving its quality.
>
> **Response to Weakness:**
>
> **Presentation** We will simplify the problem formulation by reducing the quantile levels $\phi^a_j$ and constraint threshold parameters $b^a_j$ to decrease complexity and streamline the notation system. Additionally, we will ensure that all notations are clearly defined and eliminate unnecessary symbols to enhance readability. We will address your suggestions in our revisions.
>
> **Formulation** The observations from different tasks, arms, and performance metrics are independent; we will clarify this in the problem formulation section. The motivation for considering the multi-task setting is detailed in the response to the questions section. Motivated by your feedback, we have considered a new extension to the problem formulation that imposes a linear structural relationship across tasks as shown in the public comments.
>
> **Algorithm and Analysis** See the response to the questions section.
>
> **Response to Questions:**
>
> The rationale for considering multi-task settings can be categorized into application, theoretical, and algorithmic aspects.
>
> **Application side:** The multi-task problem is commonly encountered in practical applications and has also been explored in the literature under different settings[1-2]. For example, in drug discovery, the experimenter must identify the most suitable drug for each disease while ensuring a robust performance guarantee, i.e. $\mathbb{P}(\forall a\in[M],i^* (a)=\hat{i}_{\tau}(a))\geq 1-\delta$, as considered.
>
> **Theoretical side:** The performance measure we consider in this paper is
> \begin{equation}
> \\mathbb{P}(\forall a\in[M],i^* (a)=\hat{i}_{\tau}(a)) \geq 1-\delta
> \end{equation}
>
> , which quantifies the probability of correct identification across all tasks. The optimal sampling rule under this measure must balance the difficulty of identification across tasks. Assume that we solve each task one by one, according to Bonferroni’s inequality:
>
> \begin{equation}
> \\mathbb{P}(\exists a\in[M],i^*(a)\neq \hat{i}_{\tau}(a)) \leq \sum_a \\mathbb{P}(i^* (a)\neq \hat i_t(a))
> \end{equation}
>
> , we need to ensure $\mathbb{P}(i^*(a)\neq \hat{i}_{\tau} (a))\leq\delta/M$ for each task $a\in[M]$ to achieve the same performance guarantee. However, solving tasks individually using this approach increases the sample complexity due to the conservative nature of the approximation.
>
> **Algorithm and Analysis:** In the implementation of the algorithm, our method automatically balances the difficulty of different tasks and adjusts the sampling rule accordingly. It stops when the stopping rule is satisfied, thereby achieving the performance guarantee. This approach is more natural than predefining a robust probability of error for each task without knowledge of the difficulty information, even though the latter may be near-optimal in the asymptotic regime.

---

> ### Author Response · Authors · 2024-11-21
>
> **Response to Questions:**
>
> **Sample complexity of SEQSR:**  To establish the asymptotic sample complexity results, such as Theorem 3 for the SEQSR algorithm, we need to demonstrate the following:
>
> Each random variable is sampled infinity often:
>
> \begin{equation}
>             \lim_{t\rightarrow\infty} N^a_{ij}(t)\rightarrow \infty,
> \end{equation}
>
> , which is straightforward due to the forced exploration procedure.
>
> The empirical sampling rule converges to the static optimal sampling rule:
>
> \begin{equation}
>             \sup_{t\geq t_\epsilon}\max_{(a,i,j)\in[M]\times[K]\times[S+1]}\left|\frac{N^a_{ij}(t)}{t}-(\omega^a_{ij})^*(\mathcal{P})\right|\leq \epsilon.
>         \end{equation}
>
> However, the SEQSR algorithm employs a heuristic approach to sequentially solve the optimality conditions in Theorem 2. Analyzing the relationship between the actual sampling rule and the static optimal rule remains non-trivial and is an ongoing research problem for us.
>
> **One-metric-at-a-time observation setting:**
> The one-metric-at-a-time observation setting can be justified from the application perspective, and the analysis method is also applicable to scenarios where all metrics are observed simultaneously.
>
> **Application Side:** The one-metric-at-a-time observation constraint reflects practical limitations in real-world environments where measuring each performance metric incurs additional costs. This setup allows the agent to strategically decide whether to collect specific information based on its sampling strategy and objectives.
>
> For example, in drug discovery, evaluating different performance metrics of a new drug (e.g., effectiveness, side effects) often requires distinct experiments or interventions. As a result, the agent must decide when and which metric to observe based on an optimal sampling rule. This decision balances the need to gather sufficient information to identify the best drug candidate while minimizing the overall cost and resources needed in the experimentation process.
>
> **Theoretical Side:** The algorithm development and analysis framework remains applicable even in a setting where the rewards for all metrics can be observed simultaneously when an arm is pulled. In this case, we can still derive the optimality conditions for Theorem 5.
>
>
>  **Theorem** Define $\\mathcal{A}_1=\\{a\in[M]:V^a_1(\\omega)\\geq z\\}$, and $\mathcal{A}_2=\\{a\in[M]:V^a_1(\\omega)= z\\}$. The static optimal sampling ratio $\omega$ satisfies:
>
> $$
> V_{1}^{a}(\omega)\geq z,\forall a\in \mathcal{A}_1
> $$
>
> $$
> V_{1}^{\tilde{a}}(\omega) = V_{2}^{a}(\omega) = V_{3}^{a}(\omega) = V_{4}^{a}(\omega) = z, \forall a\in [M], \tilde{a}\in \mathcal{A}_2
> $$
>
> $$
> \omega^a_i d(\mu^a_i,\mu^a_{i,i^*(a)})+\omega^a_{i^*(a)}d(\mu^a_{i^*(a)},\mu^a_{i,i^*(a)}) = z,\forall a\in[M],i\in \mathcal{D}^a_1
> $$
>
> $$\\omega^a_i \sum_{j\in\mathcal{B}^a_2(i)}d(F_{ij}^{a}(b_{j}^{a}) ,\phi_{j}) = z, \forall a\in[M], i\in \mathcal{D}_{2}^{a}$$
>
> $$\\omega^a_i (d(\mu_i^a,\mu^a_{i,i^*}(a))+\sum_{j\in\mathcal{B}^a_2(i)}d(F_{ij}^{a}(b_{j}^{a}) ,\phi_{j}) ) + \\omega^a_{i^*(a)}d(\mu^a_{i^*(a)},\mu^a_{i,i^*(a)}) = z.\forall a\in[M], i\in \mathcal{D}^a_3$$
>
> $$\sum_{i\in \mathcal{D}^a_1} \frac{d(\mu^a_{i^*(a)},\mu^a_{i,i^*(a)})}{d(\mu^a_i,\mu^a_{i,i^*(a)})} + \sum_{i\in \mathcal{D}^a_3}\frac{d(\mu^a_{i^*(a)},\mu^a_{i,i^*(a)})}{d(\mu^a_i,\mu^a_{i,i^*(a)})+\sum_{j\in\mathcal{B}^a_2(i)}d(F^a_{ij}(b^a_j),\phi_j)} = 1,\forall a\in \mathcal{A}_1$$
>
> $$\sum_{a\in[M]}\sum_{i\in[K]}\omega^{a}_{i} = 1,$$
>
> $$\omega^{a}_{i} \geq 0, \forall a\in[M], i\in[K].$$
>
> **Reperence**
>
> [1] Du Y, Huang L, Sun W. Multi-task representation learning for pure exploration in linear bandits[C]//International Conference on Machine Learning. PMLR, 2023: 8511-8564.
>
> [2] Du Y, Chen W, Kuroki Y, et al. Collaborative pure exploration in kernel bandit[J]. arXiv preprint arXiv:2110.15771, 2021.

---

> ### Comment · Reviewer_uMef · 2024-11-23
>
> I appreciate the authors for the detailed response. I've carefully read them and decided to maintain my score. I acknowledge the contribution in risk transformation, and designing low-complexity algorithms. However, given the vast number of track-and-stop papers, the standard, mature, and powerful machinery shown in TAS literature, and the lack of solid motivation to study such complex models, the lack of intuition and discussion of behind results except for heavy and hard-to-understand equations, this paper's contribution is minimal. I don't feel this paper, in its current form, has reached the high standard of ICLR. Specifically, I'm not convinced by the author's argument that all tasks should be solved together if they are independent. It seems the result of balancing the difficulty across tasks will eventually result in a similar, if not the same, probability of error for all tasks, which favors solving each task one by one, considering how much simpler it would be. This puts the motivation of the model in this paper in question.

---

> ### Author Response · Authors · 2024-11-25
>
> Thank you for your thoughtful comments and for taking the time to review our paper. Here are some clarifications we would like to provide, which we hope will address your concerns.
>
> **1.Technical Novelty on the Algorithm**
>
> We acknowledge the existing track-and-stop literature [1-3]. However, the main novelty of our work is extending this framework to a more complex setting. Specifically, the framework consists of three key components: the sampling rule, the stopping rule, and the recommendation rule. For the multi-task BAI problem with constraints, the sampling rule is derived from a new sample complexity lower bound (Theorem 1) and optimality conditions (Theorem 2). The new stopping rule (Lemma 4) and upper bound analysis (Theorem3) incorporate the effects of multiple tasks and risk constraints. The recommendation rule is the same for most algorithms, which involves recommending the arm with the maximum estimate. Without these results, the track-and-stop framework cannot be applied directly to this problem.
>
> **2.Advantages of the Multi-Task Setting**
>
> **Better numerical performance.** Compared to solving each task individually, the multi-task methods we derived demonstrate better numerical performance. We present a simple numerical result to support this claim. The numerical setting mirrors our Gaussian bandit case in Section 5, involving 2 tasks, 4 arms, and 1 constraint. To ensure a performance guarantee, we control the error probability for each task to be lower than $\delta/2$, (with $\delta=0.1$). We compare the SEQSR algorithm and run 1000 macro replications to estimate the sample complexity. The results show that the average number of samples required under the multi-task setting is 2745.814, whereas solving each task individually requires an average of 2984.009 samples. We will conduct additional numerical experiments with larger values of $M$ to further compare these two methods in the coming days.
>
> The additional sample cost stems from two factors. First, there is statistical conservatism due to the confidence level. To achieve the performance guarantee $\mathbb{P}(\forall a\in[M], i^*(a)=\hat{i}(a))\geq 1-\delta$, we apply Bonferroni’s inequality: $\mathbb{P}(\forall a\in[M], i^*(a)=\hat{i}(a)) \leq \sum_a \mathbb{P}(i^*(a)\neq \hat{i}(a))$ which requires controlling the error probability for each task to be less than $\delta/M$. This approximation introduces more samples as $M$ increases due to the approximation, which in turn degrades the algorithm’s performance. The second source of conservatism comes from the stopping rule, which also increases the sample size when tasks are solved individually.
>
> **Asymptotically Optimality.** The multi-task methods, such as ESR, are asymptotically optimal (Theorem 3). In contrast, solving each task individually leads to a sampling rule that is not asymptotically optimal. While we know that the asymptotically optimal rule aims to equalize the error probability across tasks, the exact equalized probability or $z^*$ in Theorem 2 remains unknown. Therefore, we cannot use pre-defined confidence levels and solve each task individually to achieve both statistical guarantees and asymptotic optimality simultaneously.
>
> **Generality of Mathematical Models.** The multi-task setting is more general from a mathematical perspective, as the corresponding theoretical results can encompass the single-task problem as a special case. The independent tasks setting is also well-studied in the literature [4-5]. More importantly, the formulation and analysis methods can be generalized to settings where there is linear structural information across tasks, as demonstrated in the public comments.
>
>
> **3.Insights and discussions of the Theoretical Results**
>
> We acknowledge that the theoretical results are complex due to dense notations. In the revised version, we have removed unnecessary notations, such as multiple quantile levels and constraint thresholds, for clarity. Another contribution is that we provide some insights and discussions on our theoretical results. Specifically, we describe the complexity magnitude H through challenging instances (Proposition 1), using intuitive terms like variance, optimality, and feasibility gaps. We also offer a detailed explanation (lines 243-251) of the optimality conditions in Theorem 2 and provide numerical examples in Appendix D.1 to illustrate the trade-offs between optimality, feasibility, task and constraints difficulty. In the revised version, we will make these insights clearer and provide more intuitions on Theorem 3 and the algorithm design.
>
> We value your feedback and will make every effort to improve the clarity and motivation of the paper to better align with the high standards expected by ICLR. Thank you again for your constructive criticism.

---

> ### Author Response · Authors · 2024-11-25
>
> [1] Juneja S, Krishnasamy S. Sample complexity of partition identification using multi-armed bandits[C]//Conference on Learning Theory. PMLR, 2019: 1824-1852.
>
> [2] Elahi M Q, Wei L, Kocaoglu M, et al. Adaptive Online Experimental Design for Causal Discovery[J]. arXiv preprint arXiv:2405.11548, 2024.
>
> [3] Jedra Y, Proutiere A. Optimal best-arm identification in linear bandits[J]. Advances in Neural Information Processing Systems, 2020, 33: 10007-10017.
>
> [4] Du J, Gao S, Chen C H. Rate-optimal contextual ranking and selection[J]. arXiv e-prints, 2022: arXiv: 2206.12640.
>
> [5] Li H, Lam H, Liang Z, et al. Context-dependent ranking and selection under a bayesian framework[C]//2020 winter simulation conference (WSC). IEEE, 2020: 2060-2070.

---

> > ### Author Response · Authors · 2024-11-26
> > **Additional Numerical Results**
> >
> > Dear Reviewer,
> >
> > We have added additional numerical results to further support the claim that the multi-task algorithm demonstrates superior numerical performance compared to solving each task individually. Notably, the performance gap widens as the number of tasks $M$ increases.
> >
> > The table below summarizes these results. Here, $M$ represents the number of tasks, Single-SEQSR refers to solving each task individually using the SEQSR strategy, and Multi-SEQSR refers to the proposed SEQSR method. The results indicate that Multi-SEQSR requires fewer samples to achieve the same statistical guarantees. For instance, when $M=8$, Single-SEQSR incurs an additional $18.94%$ in sample cost.
> >
> > | M  | Single-SEQSR | Multi-SEQSR | Ratio     | Additional Cost |
> > |----|--------------|-------------|-----------|-----------------|
> > | 2  | 2984.009     | 2745.814    | 1.086748  | 8.67%          |
> > | 4  | 5943.42      | 5292.847    | 1.122916  | 12.29%         |
> > | 6  | 9411.51      | 8053.553    | 1.168616  | 16.86%         |
> > | 8  | 13428.68     | 11290.694   | 1.189358  | 18.94%         |
> >
> > We sincerely hope that these numerical results and our clarifications address your concerns. Thank you for your time and thoughtful review!

---

> ### Comment · Reviewer_uMef · 2024-11-26
>
> I thank the reviewer for the additional experiment and discussion trying to address my concerns. I have the following questions regarding the new experiment:
>
> (1) $\delta = 0.1$ seems to be a very large error rate which is far from the asymptotic results studied in the paper. I'm not sure how convincing it is given the difference between a low-confidence regime and an asymptotic regime in BAI. It may be more convincing to test on a much lower $\delta$, e.g., $\delta = 10^{-5}$ in this paper:
>
> [3] Barrier, Antoine, Aurélien Garivier, and Tomáš Kocák. "A non-asymptotic approach to best-arm identification for Gaussian bandits." International Conference on Artificial Intelligence and Statistics. PMLR, 2022.
>
> (2) If the tasks are independent, why need union bound? It should be possible to exactly calculate the error probability for each task if they are the same. I feel as the number of tasks increases, the union bound becomes looser, and that partly explains the increasing sub-optimality.
>
> (3) I also don't feel comparing SEQSR is not a good example, since it is not the asymptotic optimal algorithm this paper proposes. My concern is on the approach of solving the multi-task problem itself, which is algorithm-independent. It may make more sense if a comparison is conducted in a close-to-asymptotic regime with algorithms that are close to the information limit of the problem.
>
> Generally, my concern is about the problem itself, so I feel it is best if the authors could address my concern from a theoretical perspective, since experiment results may be limited to instance setup or algorithm specifications. Let me formulate my question this way, suppose we look at the complexity measure in Eq. (9), Prop. 1 (which I conjure is the tight lower bound of an appropriate model under Gaussian noise), it can be written as:
>
> $$H = \sum_a H_a$$
>
> where $H_a$ are the terms inside the first summation in Eq.(9). So the information-theoretic limit sample complexity for all tasks is approximately:
>
> $$ \sum_a H_a log(\frac{1}{\delta})$$
>
> Now, if I separate the tasks, for any task $a$, it seems possible to design an algorithm with complexity $H_a log(\frac{M}{\delta})$ (use the TAS in the paper with a single task and error rate $\delta / M$), so the total sample complexity of solving all tasks separately would be around:
>
> $$ \sum_a H_a log(\frac{M}{\delta}) = \sum_a H_a log(\frac{1}{\delta}) + \sum_a H_a log(M)$$
>
> If we keep $M$ as constant and divide the term by $log(\frac{1}{\delta})$, and let $\delta$ approach $0$, the second term would disappear. So the complexity measure of solving each task separately would become:
>
> $$ \sum_a H_a $$
>
> which is exactly the same as in Eq.(9). So this argument seems to support that solving the tasks separately is asymptotically the same as the TAS algorithm for multi-tasks. It feels to me that in your experiment, $log(\frac{1}{\delta})$ is almost the same or even smaller than $log(M)$ which caused the second term to dominate, but this difference will be sub-order if the confidence $\delta$ is much smaller.
>
> I'm very confused by the author's argument that **"While we know that the asymptotically optimal rule aims to equalize the error probability across tasks, the exact equalized probability in Theorem 2 remains unknown. "** Is the optimal rule equalizing the error probability or not? If so, in the asymptotic regime, they should be the same since you have infinite adaptivity. If not, in the asymptotic regime, which task should have a higher error probability? and how much higher? how is this difference reflected in the complexity measure $H$ in Prop.1? I believe these questions are very important and need to be answered to address my confusion.
>
> The confusion on this problem setting is an example of my skepticism towards the paper's technical novelty, given the vast amount of TAS literature. From my perspective, the authors have not provided enough technical contribution beyond the classic and mature TAS machinery. I will maintain my score.
> ***
> Sorry for the notation mistake if any.

---

> > ### Author Response · Authors · 2024-11-26
> > **Response to the Reviewer**
> >
> > Thank you for your detailed comments regarding the additional experimental results. I would like to clarify the following points:
> >
> > **Asymptotic Optimality.** Our previous feedback mistakenly conflated the fixed-confidence setting (with a higher
> > $\delta$) with the asymptotic setting (where $\delta \rightarrow 0$). Apologies for any confusion caused, and I would like to clarify it here. In the asymptotic regime, the optimal rule indeed aims to equalize the error probability across tasks (as shown by the first equation of Theorem 2). Additionally, solving each task individually using TAS with a confidence level of $\delta/M$ is also asymptotically optimal.
> >
> > **Better Numerical Performance.** The numerical experiments were conducted within the fixed-confidence setting with a higher $\delta$. We did not implement the algorithm with $\delta = 10^{-5}$ due to its high computational cost. Unlike [3], which has only five random variables, our setting involves more random variables due to the multiple tasks and arms. Solving each task individually indeed requires more samples, and the performance gap widens as $M$ increases in the fixed-confidence setting. This may be due to the statistical conservatism of the stopping rule.
> >
> > **Multi-task Setting.** We retain the multi-task setting as it is a more general mathematical model. This model includes the single-task case as a special instance and can also be extended to scenarios involving linear relationships across tasks (as shown in Theorem 6 of the revised paper).
> >
> > **Technical Contribution.** The main contribution we wish to emphasize is the solution to the risk-constrained BAI problem. To address this, we perform a risk constraint transformation, classify arms into different categories based on optimality and feasibility, and derive the sample complexity for each category, which is non-trivial. To the best of our knowledge, our sample complexity results represent new contributions to the risk-constrained BAI problem. As seen in equation (9), different types of arms exhibit varying levels of difficulty in $\mathcal{H}$, and we incorporate the feasibility gap, optimality gap, and variance to characterize the hardness of each category. This approach differs significantly from existing results. The insights in Theorem 2 and the illustrative examples in D.1 provide valuable guidance on the sampling rule, which also incorporates existing BAI results as a special case. We hope the multi-task setting, for generality purpose, will not detract from the evaluation of the other contributions of our paper.
> >
> > Finally, thank you for your thorough review, detailed feedback, and constructive criticism. We greatly appreciate your time and patience. We will make every effort to improve the quality of our paper to better meet the high standards expected by ICLR.

---

> > > ### Comment · Reviewer_uMef · 2024-11-28
> > >
> > > I thank the authors for the detailed response to clarify my question and the summary of the highlight of their paper. I've taken them into consideration and intend to maintain my score. I don't have further questions.

---

> > > > ### Author Response · Authors · 2024-11-28
> > > >
> > > > Thank you once again for your thorough review and valuable feedback. We truly appreciate your thoughtful suggestions!

---

### Official Review · Reviewer_QDjz · 2024-10-29

**Soundness:** 3
**Presentation:** 3
**Contribution:** 3
**Rating:** 6
**Confidence:** 3

**Summary:**

This paper introduces a multi-task best arm identification problem with risk constraint in the fixed-confidence setting, where each arm has multiple performance metrics. The agent aims to optimize one metric, while ensuring that the quantiles of other metrics remain below specified thresholds for each task. The authors first build a tight, instance-dependent lower bound on sample complexity. Based on this bound, the authors establish optimality conditions for the static optimal sampling ratio, and illustrate how it balances among different tasks and constraints, while addressing the tradeoff between optimality and feasibility. In addition, the authors design a Track-and-Stop strategy with asymptotically optimal sample complexity, and a computationally efficient strategy that iteratively solves the optimality conditions. Finally, the authors extend their results to the linear bandit setting. Numerical experiments show that the proposed algorithm performs relatively well.

**Strengths:**

1.	The authors propose a multi-task BAI problem with risk constraint applicable to various real-world problems. The authors establish a tight, instance-dependent lower bound on the sample complexity required to guarantee a high probability identification of the feasible and optimal arm for each task.
2.	Based on the lower bound, the authors derive the optimality conditions for the static optimal sampling ratio, and discuss how this ratio balances the difficulty across different tasks and constraints, illustrating the trade-off between optimality and feasibility. In addition, the authors present a closed-form formula for the problem’s hardness by analyzing some challenging instances.
3.	The authors develop a Track-and-Stop strategy for the multi-task BAI problem with risk constraint, achieving asymptotically optimal sample complexity and a computationally efficient strategy that iteratively solves the optimality conditions. What’s more, the authors also extend their sample complexity results to the linear bandit setting. Numerical experiments show that the proposed algorithm performs well in comparison to several benchmarks.
4.	This paper is overall well-written and clearly organized.

**Weaknesses:**

1.	The motivation of the proposed problem formulation is not very clear. Can the authors describe 1-2 concrete application scenarios for the proposed problem formulation, in particular, connect the risk constraint formulation with real-world requirements on risk management.
2.	It is a little surprising that the sample complexity of Algorithm 1 (Theorem 3) perfectly matches the lower bound for the multi-task best arm identification problem with risk constraints, even for logarithmic factors. Can the authors give the intuition on why Algorithm 1 can avoid a union bound over the number of arms and tasks in the logarithmic factor?
3.	The formulas and notations in this paper are dense. The authors should add more intuition behind the algorithm design and theoretical results.

**Questions:**

Please see the weaknesses above.

---

> ### Author Response · Authors · 2024-11-21
>
> Thank you for your detailed and constructive review. We appreciate your feedback and the opportunity to clarify the motivation and theoretical results of our paper.
>
> Per your suggestion, we will provide the following three examples to illustrate the motivation behind the proposed problem formulation in the revised version.
>
> **Drug discovery example.**  In drug discovery, each candidate drug (arm) is evaluated based on multiple performance metrics, such as efficacy and side effects. The primary goal is to identify the drug with the highest average curative effect while ensuring that the risk (measured by a quantile in our paper) of severe side effects remains below a specified threshold for each targeted disease (task).
>
> **Supply chain management example.** In supply chain management, companies often evaluate multiple suppliers (arms) based on multiple performance metrics such as cost, delivery reliability, and product quality. The primary goal is to identify the supplier with the lowest average cost while ensuring the likelihood of significant delays or quality issues (quantile-based risk measure) remains below a specified threshold across different product categories (tasks).
>
> **Financial risk management example.** In financial risk management, an investor evaluates various investment strategies (arms) based on multiple performance metrics, such as expected return, volatility, and drawdown. The primary goal is to identify the strategy with the highest expected return while controlling risk metrics that remain below a specified threshold across different market conditions (tasks).
>
> The sample complexity lower bound in Theorem 1 is determined by two factors: the confidence level related term $kl(\delta,1-\delta)$, and the problem instance-dependent term $\mathcal{H}^* (\mathcal{P})$, which scales with the number of tasks, arms, and constraints. For the static optimal sampling rule, the sample complexity upper bound can be derived using a union bound, resulting in $\log(MK(S+1)/\delta)$ term. However, our analysis focuses on the Track-and-Stop algorithm, which employs an adaptive sampling rule and is significantly more challenging than the static case. Since we operate in the asymptotic regime where $\delta \rightarrow 0$, some constant terms in the sample complexity vanish asymptotically. Consequently, we conclude that the upper bound matches the lower bound asymptotically.
>
> In the revised version, we will provide more intuition and detailed explanations on the algorithm design and theoretical results.

---

> > ### Comment · Reviewer_QDjz · 2024-11-25
> > **Thank you for your response.**
> >
> > Thank you for your response. I tend to maintain my score.
> >
> > It would improve the paper if the authors can include these explanations in their revision.

---

> > > ### Author Response · Authors · 2024-11-26
> > >
> > > Thank you for your constructive review and valuable feedback! I will incorporate these explanations into the revised version of the paper.

---

### Official Review · Reviewer_5qqJ · 2024-10-31

**Soundness:** 3
**Presentation:** 3
**Contribution:** 2
**Rating:** 6
**Confidence:** 2

**Summary:**

This paper tackles the complex problem of Best Arm Identification in sequential decision-making, focusing on multi-task scenarios with risk constraints. The authors derive a tight lower bound on sample complexity and propose a Track-and-Stop strategy with optimal performance. Their approach balances tasks and constraints effectively, and results demonstrate strong performance in numerical experiments, extending findings to the linear bandit setting.

**Strengths:**

1. The methodology employed is robust, featuring a detailed analysis across several variants: the foundational algorithm is presented in Section 4.1, an optimized version in Section 4.2, and an adaptation to linear settings discussed in Section 5.

2. The numerical experiments conducted are thorough, employing ESR, ASR, and FWSR as benchmark algorithms.

**Weaknesses:**

The technical innovation presented in this paper appears to be limited. Algorithm 1 essentially replicates the Track-and-stop method referenced in [1]. Consequently, the paper's novelty should primarily be evaluated based on the efficient sampling rule introduced in Section 4.2. Nevertheless, this sampling rule closely resembles the BestChallenger method detailed in [1, Section 6], and Equation (17) directly corresponds to the definition of BestChallenger within this setting. Furthermore, the authors should have cited the BestChallenger method in Section 4.2 to acknowledge the previous researchers' efforts.

[1] Garivier, A., & Kaufmann, E. (2016, June). Optimal best arm identification with fixed confidence. In Conference on Learning Theory (pp. 998-1027). PMLR.

**Questions:**

see Weaknesses.

---

> ### Author Response · Authors · 2024-11-21
>
> Thank you for your detailed and thoughtful review. We appreciate your constructive feedback on the technical aspects of the paper.
>
> The proposed Algorithm 1 follows the Track-and-stop framework from [1], which has been extended in various formulations by several recent studies [2-4]. However, the primary novelty of our work lies in extending this framework to a more complex multi-task BAI setting with risk constraints. This extension required several key innovations:
>
>
> **Handling Multi-Task and Quantile Constraints.** Unlike standard BAI problems, our setting introduces multiple tasks with quantile-based risk constraints, which significantly increase the problem's complexity. To address this, we developed a new analysis to derive a tight sample complexity lower bound (Theorem 1) and characterize optimal sampling conditions (Theorem 2).
>
> **Insights on Problem Hardness and Optimality.** Through our analysis, we provided new insights into the problem's hardness (Proposition 1) and optimal sampling rule, which are essential for understanding the complexity of the multi-task BAI problem with risk constraints.
>
> **Extending Analysis to Multi-Task Linear Bandits.** We further demonstrated the generality of our method by extending it to a multi-task linear bandit scenario (Theorem 4), showcasing the broader applicability of our analysis framework.
>
> The efficient sampling rule is indeed an extension of the BestChallenger method in [1] to the multi-task BAI with risk constraints setting based on our theoretical results.  Per your suggestion, we will revise the paper to explicitly cite the BestChallenger method and the adaptation we need to make to handle the additional complexity of multi-task, risk-constrained scenarios.
>
> **Reference**
>
> [1] Garivier A, Kaufmann E. Optimal best arm identification with fixed confidence[C]//Conference on Learning Theory. PMLR, 2016: 998-1027.
>
> [2] Juneja S, Krishnasamy S. Sample complexity of partition identification using multi-armed bandits[C]//Conference on Learning Theory. PMLR, 2019: 1824-1852.
>
> [3] Elahi M Q, Wei L, Kocaoglu M, et al. Adaptive Online Experimental Design for Causal Discovery[J]. arXiv preprint arXiv:2405.11548, 2024.
>
> [4] Jedra Y, Proutiere A. Optimal best-arm identification in linear bandits[J]. Advances in Neural Information Processing Systems, 2020, 33: 10007-10017.

---

> > ### Comment · Reviewer_5qqJ · 2024-11-25
> >
> > Thanks for acknowledging that the design of Section 4.2 is indeed following BestChallenger of Garivier and Kaufmann. I will determine my final rating after the discussion with other reviewers as well as AC in the next stage. I don't have further questions.

---

> > > ### Author Response · Authors · 2024-11-25
> > >
> > > Thank you once again for your thorough review and valuable feedback. Please don't hesitate to reach out if you have any further questions!

---

### Author Response · Authors · 2024-11-21
**A new linear bandit formulation leveraging structural information across tasks.**

Motivated by the reviewers' feedback, we have introduced a new extension to the problem formulation that incorporates a linear structural relationship across tasks.

**Application example** In personalized medicine, the agent must identify the best drug (arm) for each patient (task). Each patient has specific features, such as demographics and physical conditions. The efficacy of a drug tends to be similar for patients with similar features, allowing us to model this structural information across tasks to improve sampling efficiency.

**Problem formulation** Consider that each task $a$ corresponds to a vector $c^a\in \mathbb{R}^d$. Each arm $i$ has two performance metrics $(X^a_i, Y^a_i)\in\mathbb{R}^2,$ which can be extended to a multiple-constraints setting.

**Assumption** There exist unknown parameters $\beta_i,\gamma_i,i\in[K]$ such that $X^a_i = \beta_i^Tc^a+\epsilon$ and $Y^a_i = \gamma_i^Tc^a+\epsilon$, where $\epsilon \sim N(0,\sigma^2)$.

**Decision problem:** The agent needs to solve the following optimization problem for each task
        \begin{equation}
            \max_{i\in[K]}\beta_i^Tc^a~~\text{s.t.}~\gamma_i^Tc^a\leq b.
        \end{equation}

**Theoretical result:** Using our analysis method, we can derive the sample complexity results for this setting:

**Theorem** Given a fixed confidence level $\delta \in (0,1)$. For any linear BAI problem instance $\mathcal{P}\in \mathcal{S}$ and any strategy satisfying $\mathbb{P}\left(\forall a\in[M], i^{*}(a) = \hat{i}_{\tau}(a)\right)\geq 1-\delta$,

\begin{equation}
    \mathbb{E}[\tau] \geq \mathcal{H}^{*}(\mathcal{P})\text{kl}(\delta,1-\delta),
\end{equation}

with

\begin{equation}
\mathcal{H}^*(\mathcal{P})^{-1} = \frac{1}{2}\sup_{\omega\in \Omega}\min_{a\in[M]}\min(V^a_1(\omega), V^a_2(\omega),V^a_3(\omega),V^a_4(\omega))
\end{equation}

$$
V^a_1(\omega) = \frac{(b-\gamma_{i^*(a)}^{T}c^{a})^2}{(c^a)^{T}\Lambda_{i^*(a)}^{-1}c^a}
$$

$$
V^a_2(\omega) = \min_{i\in \mathcal{D}^{a}_1} \frac{((\beta_i^*-\beta_i)^Tc^a)^2}{(c^a)^T(\Sigma(i)^{-1}+\Sigma(i^*(a))^{-1})c^a}
$$

$$
V^a_3(\omega) = \min_{i\in \mathcal{D}^{a}_2} \frac{(b - \gamma_i^{T}c^{a})^2}{(c^a)^{T}\Lambda_i^{-1}c^a}
$$

$$
V^a_4(\omega) =  \min_{i\in \mathcal{D}^{a}_3}\frac{((\beta_i^*-\beta_i)^Tc^a)^2}{(c^a)^T(\Sigma(i)^{-1}+\Sigma(i^*(a))^{-1})c^a} + \frac{(b - \gamma_i^{T}c^{a})^2}{(c^a)^{T}\Lambda_i^{-1}c^a}
$$

where $\Lambda_i=\sum_a\frac{\omega^a_{i}}{\sigma^2}c^a(c^a)^{T}$,  and $\Sigma(i) =  \sum_a \frac{\rho_i^a}{\sigma^2}c^a(c^a)^{T}$, $\omega^a_{i}, \rho_i^a$ represents the sampling ratio of  constraint and objective of arm $i$ under task $a$, respectively.

---

### Meta-Review · Area_Chair_FYnv · 2024-12-21

**Metareview:**

This paper considers a multi-task best arm identification problem with risk constraint in the fixed-confidence setting. The authors first prove a lower bound on the sample complexity for the problem studied in this paper, and then design a track-and-stop algorithm with asymptotically optimal sample complexity. The authors also give computationally efficient algorithms and show how to extend their results to linear bandits.

The first weakness of this paper is the lack of motivation. The setting studied in this paper is rather complicated, and could be unnecessarily complicated and therefore confusing. This has been observed by two out of the four reviewers.

The second weakness of this paper is the lack of technical novelty. The algorithm in this paper basically replicates the track-and-stop method by Garivier and Kaufmann and therefore, the technical innovation is rather limited. This has also been observed by two out of the four reviewers.

Given the high standards of ICLR and the weakness mentioned above, I would recommend rejecting this paper.

**Additional Comments On Reviewer Discussion:**

The reviewers raised concerns regarding the lack of motivation, the lack of technical novelty, as well as clarity of the problem formulation . Although the authors provided responses which addressed some of those concerns, concerns regarding the lack of motivation and the lack of technical novelty remain.

---

### Decision · Program_Chairs · 2025-01-22

Reject